

# An integrative taxonomic revision of slug-eating snakes (Squamata: Pareidae: Pareineae) reveals unprecedented diversity in Indochina

Nikolay A. Poyarkov[1,2], Tan Van Nguyen[3], Parinya Pawangkhanant[4], Platon V. Yushchenko[2], Peter Brakels[5], Linh Hoang Nguyen[6], Hung Ngoc Nguyen[6], Chatmongkon Suwannapoom[4], Nikolai Orlov[7] and Gernot Vogel[8]

[1] Laboratory of Tropical Ecology, Joint Russian-Vietnamese Tropical Research and Technological Center, Hanoi, Vietnam
[2] Faculty of Biology, Department of Vertebrate Zoology, Moscow State University, Moscow, Russia
[3] Department of Species Conservation, Save Vietnam's Wildlife, Ninh Binh, Vietnam
[4] Division of Fishery, School of Agriculture and Natural Resources, University of Phayao, Phayao, Thailand
[5] IUCN Laos PDR, Vientiane, Lao PDR
[6] Department of Zoology, Southern Institute of Ecology, Vietnam Academy of Science and Technology, Ho Chi Minh City, Vietnam
[7] Department of Herpetology, Zoological Institute, Russian Academy of Sciences, St. Petersburg, Russia
[8] Society for Southeast Asian Herpetology, Heidelberg, Germany

Corresponding author
Nikolay A. Poyarkov,
n.poyarkov@gmail.com

## ABSTRACT

Slug-eating snakes of the subfamily Pareinae are an insufficiently studied group of snakes specialized in feeding on terrestrial mollusks. Currently Pareinae encompass three genera with 34 species distributed across the Oriental biogeographic region. Despite the recent significant progress in understanding of Pareinae diversity, the subfamily remains taxonomically challenging. Here we present an updated phylogeny of the subfamily with a comprehensive taxon sampling including 30 currently recognized Pareinae species and several previously unknown candidate species and lineages. Phylogenetic analyses of mtDNA and nuDNA data supported the monophyly of the three genera *Asthenodipsas*, *Aplopeltura*, and *Pareas*. Within both *Asthenodipsas* and *Pareas* our analyses recovered deep differentiation with each genus being represented by two morphologically diagnosable clades, which we treat as subgenera. We further apply an integrative taxonomic approach, including analyses of molecular and morphological data, along with examination of available type materials, to address the longstanding taxonomic questions of the subgenus *Pareas*, and reveal the high level of hidden diversity of these snakes in Indochina. We restrict the distribution of *P. carinatus* to southern Southeast Asia, and recognize two subspecies within it, including one new subspecies proposed for the populations from Thailand and Myanmar. We further revalidate *P. berdmorei*, synonymize *P. menglaensis* with *P. berdmorei*, and recognize three subspecies within this taxon, including the new subspecies erected for the populations from Laos and Vietnam. Furthermore, we describe two new species of *Pareas* from Vietnam: one belonging to the *P. carinatus* group from southern Vietnam, and a new member

of the *P. nuchalis* group from the central Vietnam. We provide new data on
*P. temporalis*, and report on a significant range extension for *P. nuchalis*.
Our phylogeny, along with molecular clock and ancestral area analyses, reveal a
complex diversification pattern of Pareinae involving a high degree of sympatry of
widespread and endemic species. Our analyses support the "upstream" colonization
hypothesis and, thus, the Pareinae appears to have originated in Sundaland
during the middle Eocene and then colonized mainland Asia in early Oligocene.
Sundaland and Eastern Indochina appear to have played the key roles as the centers
of Pareinae diversification. Our results reveal that both vicariance and dispersal
are responsible for current distribution patterns of Pareinae, with tectonic
movements, orogeny and paleoclimatic shifts being the probable drivers of
diversification. Our study brings the total number of Pareidae species to 41 and
further highlights the importance of comprehensive taxonomic revisions not only for
the better understanding of biodiversity and its evolution, but also for the elaboration
of adequate conservation actions.

Biogeography, Southeast Asia, Sundaland, Cryptic species

## INTRODUCTION

The snakes of the family Pareidae Romer, 1956 (Squamata, Serpentes) currently (as of
December 1st, 2021) encompassing 39 species inhabiting the Oriental biogeographic
region are divided into two subfamilies: Pareinae Romer, 1956 in Southeast Asia and
Xylophiinae *Deepak, Ruane & Gower, 2019* in southern India (*Deepak, Ruane & Gower,
2019*; *Uetz, Freed & Hošek, 2021*). Slug-eating snakes (or snail-eating snakes) of the
subfamily Pareinae are widely distributed throughout the tropical and subtropical areas of
Southeast and East Asia. Its members are mainly small-sized, arboreal, nocturnal snakes,
and are regarded as dietary specialists of terrestrial pulmonates *i.e.*, slugs and snails
(*You, Poyarkov & Lin, 2015*; *Cundall & Greene, 2000*). Snail-eating species of Pareinae
are unique among terrestrial vertebrates in having asymmetric lower jaws, with more teeth
on the right mandible than on the left (*Hoso, Asami & Hori, 2007*; *Hoso et al., 2010*).
Due to the specialized feeding habit and foraging behaviour, the evolutionary biology of
*Pareas* has received much attention in recent years (*Götz, 2002*; *Hoso & Hori, 2006*, *2008*;
*Hoso, 2007*; *Hoso, Asami & Hori, 2007*; *Hoso et al., 2010*; *You, Poyarkov & Lin, 2015*;
*Danaisawadi et al., 2015*, *2016*; *Kojima et al., 2020*; *Chang et al., 2021*).

The subfamily Pareinae had a turbulent taxonomic history (*David & Vogel, 1996*; *Rao &
Yang, 1992*) with recent works (*Grossmann & Tillack, 2003*; *Guo et al., 2011*; *Ding et al.,
2020*; *Vogel et al., 2020*, *2021*) recognizing three genera: *Pareas Wagler, 1830* with 24
species (type species: *Pareas carinatus Wagler, 1830*); *Asthenodipsas Peters, 1864* with nine
species (type species: *Asthenodipsas malaccana* [*Peters, 1864*]), and a monotypic genus
*Aplopeltura Duméril, 1853* (type species: *Aplopeltura boa* [*Boie, 1828*]). Two genus-level
nomens, namely *Eberhardtia Angel, 1920* (type species: *Eberhardtia tonkinensis*

*Angel, 1920*, regarded as a synonym of *Pareas formosensis* [Van Denburgh, 1909] by *Ding et al., 2020*) and *Internatus* Yang & Rao, 1992 (type species: *Asthenodipsas leavis* [Boie, 1827]) are presently considered junior synonyms of the genera *Pareas* and *Asthenodipsas*, respectively (see *Grossmann & Tillack, 2003*; *Wallach, Williams & Boundy, 2014*; *Ding et al., 2020*; *Vogel et al., 2020, 2021*). Several recent phylogenetic studies suggested that the genus *Pareas* consists of two highly divergent major clades and is paraphyletic with respect to *Aplopeltura* or *Asthenodipsas* (*Guo et al., 2011*; *Pyron et al., 2011*; *Wang et al., 2020*). At the same time, other multilocus studies recovered *Pareas* as a monophyletic group though with moderate or low node support values, and suggested the genus *Aplopeltura* as its sister taxon (*Pyron, Burbrink & Wiens, 2013*; *You, Poyarkov & Lin, 2015*; *Figueroa et al., 2016*; *Deepak, Ruane & Gower, 2019*; *Zaher et al., 2019*). The genus *Asthenodipsas* was also shown to include two major lineages (*Loredo et al., 2013*; *Figueroa et al., 2016*; *Deepak, Ruane & Gower, 2019*; *Wang et al., 2020*), though its monophyly got only moderate support based on the concatenated analysis of mitochondrial and nuclear DNA markers (*Wang et al., 2020*; *Ding et al., 2020*; *Vogel et al., 2021*). Therefore, despite the recent significant progress in evolutionary studies on Pareinae, the phylogenetic relationship among the major genus-level lineages of the subfamily still remain debated and unclear.

Several recent taxonomic studies have demonstrated that the species diversity of Pareinae is still underestimated (*e.g.*, *Vogel, 2015*; *Hauser, 2017*; *Quah et al., 2019, 2020*; *Quah, Lim & Grismer, 2021*; *Le et al., 2021*). The high degree of morphological similarity among closely related taxa of Pareinae often makes species delineation in slug snakes quite challenging (*Guo & Deng, 2009*; *Vogel, 2015*; *Yang et al., 2021*), suggesting that the molecular data represent an effective tool to help untangle taxonomic controversies when morphological analyses yield inconsistent results (*You, Poyarkov & Lin, 2015*; *Loredo et al., 2013*; *Vogel et al., 2020, 2021*; *Bhosale et al., 2020*; *Wang et al., 2020*; *Ding et al., 2020*; *Liu & Rao, 2021*; *Yang et al., 2021*). Application of the integrative taxonomic approach combining evidence from morphological and molecular data resulted in the discovery of several previously unnoticed taxa and allowed to revise several species complexes, including the *Pareas hamptoni* complex (*You, Poyarkov & Lin, 2015*; *Bhosale et al., 2020*; *Ding et al., 2020*; *Liu & Rao, 2021*; *Yang et al., 2021*), the *P. margaritophorus* complex (*Vogel et al., 2020*; *Suntrarachun et al., 2020*), and the *P. monticola* complex (*Vogel et al., 2021*).

On the other hand, the Keeled slug snake, *Pareas carinatus*, has received comparatively little attention in most recent revisions. This species was originally described by *Wagler (1830)* from Java, Indonesia, and was later reported to be widely distributed throughout Southeast Asia, from southern China, southern Myanmar, Laos, south-western and eastern Cambodia, Vietnam, Thailand, southwards to Peninsular Malaysia, and islands of Borneo, Sumatra, Java and Bali (*Wallach, Williams & Boundy, 2014*). However, since geographic variation of this species has never been examined across the different regions, its taxonomic status remained controversial and a number of misidentifications were made in the past (*e.g.*, see discussion in *Das, 2012, 2018*). Recently, *Wang et al. (2020)* demonstrated *P. nuchalis* (*Boulenger, 1900*) to be closely related to *P. carinatus* complex,

and divided the latter by describing *P. menglaensis Wang et al., 2020*, as a sister species of *P. carinatus sensu stricto*. However, in this revision the authors did not examine type specimens of *P. carinatus*, and also have neglected to re-evaluate the status of two available species names currently considered as junior synonyms of *P. carinatus*: *Pareas berdmorei* Theobald, 1868, and *Amblycephalus carinatus unicolor* Bourret, 1934 (see *Nguyen, Ho & Nguyen, 2009*; *Wallach, Williams & Boundy, 2014*; *Uetz, Freed & Hošek, 2021*). The most recent addition to the taxonomy of the group is the discovery of a new species from southern Vietnam – *P. temporalis*, which was suggested as a sister species to *P. nuchalis* from Borneo (*Le et al., 2021*); the authors also provided the most complete phylogeny for the genus *Pareas* published up to date, generally concordant with the earlier results (*Wang et al., 2020*; *Vogel et al., 2021*). The taxonomic history of the *P. carinatus – P. nuchalis* complex is summarized in Table 1. Overall, the taxonomic status of *P. carinatus*, its synonyms, and *P. menglaensis* remains unclear pending an integrative study combining data on molecular and morphological variation of this group throughout its range.

In the present study, we provide an updated phylogeny for the subfamily Pareinae based on the analysis of mitochondrial and nuclear DNA markers, and re-assess the genus-level taxonomy of the group. Based on an extensive sampling we also report on a previously unrecognized diversity of the genus *Pareas* in Indochina. We examine name-bearing types and re-assess taxonomy of the *P. carinatus* complex using an integrative taxonomic approach, combining morphological and molecular data from the newly collected and older specimens preserved in herpetological collections. We als o provide an updated identification key for the members of the subfamily Pareinae and species of the *P. carinatus* complex. Finally, we conduct a divergence time estimation analysis for the subfamily Pareinae and discuss evolution and the historical biogeography of this peculiar group of snakes.

## MATERIALS AND METHODS

### Species concept

In the present study, we follow the General Lineage Concept (GLC: *De Queiroz, 2007*) which suggests that a species constitutes a population of organisms independently evolving from other such populations owing to a lack of gene flow (*Barraclough, Birky & Burt, 2003*; *De Queiroz, 2007*). Numerous recent integrative taxonomic studies, rather than relying solely on traditional taxonomic procedure increasingly, use a wide range of empirical data to delimit species boundaries (reviewed in *Coyne & Allen, 1998*; *Knowles & Carstens, 2007*; *Fontaneto et al., 2007*). We herein follow the framework of integrative taxonomy (*Padial et al., 2010*; *Vences et al., 2013*) which relies on independent multiple lines of evidence to assess the taxonomic status of the lineages. To infer species boundaries we use the DNA-based molecular phylogenies, while to describe those boundaries we rely on univariate (ANOVA) and multivariate (PCA) morphological analyses (*e.g. Okamiya et al., 2018*).

Table 1 Species-level scientific names erected for the members of the subgenus *Pareas*.

| No. | Authority | Original taxon name | Type locality | Previous taxonomy | New taxonomy |
|-----|-----------|---------------------|---------------|-------------------|--------------|
| 1 | *Wagler (1830)* | *Pareas carinata* | Java, Indonesia | *Pareas carinatus* | *Pareas carinatus* |
| 2 | Theobald (1868) | *Pareas berdmorei* | Mon State, Myanmar | synonym of *Pareas carinatus* | *Pareas berdmorei* |
| 3 | *Boulenger (1900)* | *Amblycephalus nuchalis* | Matang, Kidi District, Sarawak, Malaysia | *Pareas nuchalis* | *Pareas nuchalis* |
| 4 | *Bourret (1934)* | *Amblycephalus carinatus unicolor* | Kampong Speu Province, Cambodia | synonym of *Pareas carinatus* | *Pareas berdmorei unicolor* **comb. nov.** |
| 5 | *Wang et al. (2020)* | *Pareas menglaensis* | Mengla County, Yunnan Province, China | *Pareas menglaensis* | synonym of *Pareas berdmorei* |
| 6 | *Le et al. (2021)* | *Pareas temporalis* | Doan Ket Commune, Da Huoai District, Lam Dong Province, Vietnam | *Pareas temporalis* | *Pareas temporalis* |
| 7 | this paper | *Pareas carinatus tenasserimicus* | Suan Phueng District, Ratchaburi Province, Thailand | - | *Pareas carinatus tenasserimicus* **ssp. nov.** |
| 8 | this paper | *Pareas berdmorei truongsonicus* | Nahin District, Khammouan Province, Laos | - | *Pareas berdmorei truongsonicus* **ssp. nov.** |
| 9 | this paper | *Pareas kuznetsovorum* | Song Hinh District, Phu Yen Province, Vietnam | - | *Pareas kuznetsovorum* **sp. nov.** |
| 10 | this paper | *Pareas abros* | Song Thanh N.P., Quang Nam Province, Vietnam | - | *Pareas abros* **sp. nov.** |

## Nomenclatural acts

The electronic version of this article in Portable Document Format (PDF) will represent a published work according to the International Commission on Zoological Nomenclature (ICZN), and hence the new names contained in the electronic version are effectively published under that Code from the electronic edition alone (see Articles 8.5–8.6 of the Code). This published work and the nomenclatural acts it contains have been registered in ZooBank, the online registration system for the ICZN. The ZooBank Life Science Identifiers (LSIDs) can be resolved and the associated information can be viewed through any standard web browser by appending the LSID to the prefix http://zoobank.org/.

The LSID for this publication is as follows: urn:lsid:zoobank.org:pub:192CDD83-E08C-40B1-92EB-3DB2C3E63CFA. The online version of this work is archived and available from the following digital repositories: PeerJ, PubMed Central and CLOCKSS.

## Taxon sampling

We used tissues from the herpetological collections of Zoological Museum of Moscow University (ZMMU, Moscow, Russia); California Academy of Sciences Museum (CAS; San Francisco, CA, USA); Southern Institute of Ecology Zoological Collection (SIEZC, Ho Chi Minh City, Vietnam); School of Agriculture and Natural Resources, University of Phayao (AUP, Phayao, Thailand); and National Museum of Natural Science (NMNS, Taichung, Taiwan) (summarized in Table S1 and Appendix S1). For alcohol-preserved voucher specimens stored in museum collections, we removed a small sub-sample of muscle, preserved it in 96% ethanol, and stored samples at −70 °C. Altogether, we analyzed 48 tissue samples representing 20 nominal taxa of the genus *Pareas*. Geographic location of sampled populations of the members of the subgenus *Pareas* is presented in Fig. 1.

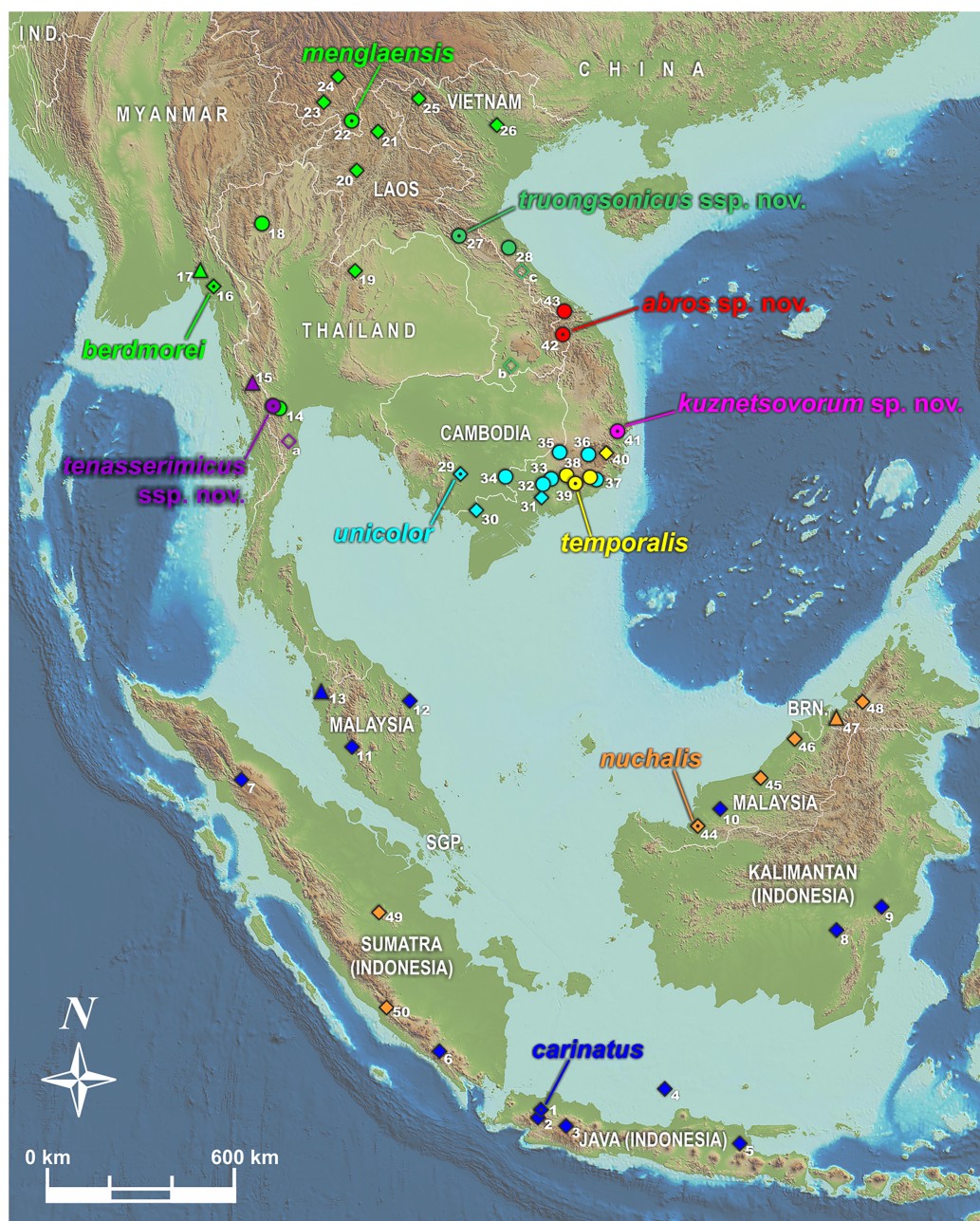

**Figure 1 Map showing distribution of the subgenus *Pareas* and location of studied populations.**
Circles denote localities for which both DNA and morphological data were examined; triangles denote
populations for which only DNA data were available; filled diamonds denote localities for which only
morphological data were available; empty diamonds denote localities for which only photo data was
available and therefore the species identification is tentative (a–c); dot in the center of an icon indicates
the type locality. **Confirmed localities:** (1) Indonesia, Java; (2) Indonesia, West Java, Bogor; (3) Indo-
nesia, West Java; (4) Indonesia, Central Java, Karimundjava Isl.; (5) Indonesia, East Java; (6) Indonesia,
Sumatra, Ranau Lake; (7) Indonesia, North Sumatra; (8) Indonesia, Borneo, Central Kalimantan, Moara
Terweh; (9) Indonesia, Borneo, East Kalimantan, Kutai N.P.; (10) Malaysia, Borneo, Sarawak; (11)
Malaysia, Pahang, Frazers Hills; (12) Malaysia, Terengganu; (13) Malaysia, Kedah, Sungai Sedim; (14)
Thailand, Ratchaburi, Suan Phueng; (15) Myanmar, Tanintharyi, Yaephyu; (16) Myanmar, Mon;
(17) Myanmar, Mon, Kyaikhto, Kinpon Chaung; (18) Thailand, Chiang Mai, Doi Inthanon N.P.;
(19) Thailand, Phitsanulok, Phu Hin Rong Kla N.P.; (20) Laos, Luangphrabang; (21) Laos, Phongsaly;

**Figure 1** (continued)

(22) China, Yunnan, Mengla; (23) China, Yunnan, Xishuangbannna; (24) China, Yunnan, Pu'er; (25) Vietnam, Dien Bien, Muong Nhe N.R.; (26) Vietnam, Vinh Phuc, Tam Dao N.P.; (27) Laos, Khammouan, Nahin; (28) Vietnam, Quang Binh, Thanh Thach; (29) Cambodia, Kampong Speu; (30) Vietnam, An Giang, Bay Nui Mt.; (31) Vietnam, Dong Nai, Trang Bom; (32) Vietnam, Dong Nai, Ma Da N. R. (Vinh Cuu); (33) Vietnam, Dong Nai, Cat Tien N.P.; (34) Vietnam, Tay Ninh, Lo Go - Xa Mat N.P.; (35) Vietnam, Binh Phuoc, Bu Gia Map N.P.; (36) Vietnam, Lam Dong, Loc Bao; (37) Vietnam, Lam Dong, Di Linh; (38) Vietnam, Lam Dong, Cat Loc; (39) Vietnam, Lam Dong, Da Huoai; (40) Vietnam, Lam Dong, Bidoup - Nui Ba N.P.; (41) Vietnam, Phu Yen, Song Hinh; (42) Vietnam, Quang Nam, Song Thanh N.P.; (43) Vietnam, Thua Thien-Hue, A Roang, Sao La N.R.; (44) Malaysia, Sarawak, Betong, Saribas; (45) Malaysia, Sarawak, Niah N.P.; (46) Malaysia, Sarawak, Bintulu; (47) Brunei, Brunei Darussalam; (48) Malaysia, Sabah, Tenom; (49) Indonesia, Sumatra, Riau, Indragiri; (50) Indonesia, Sumatra, Bengkulu, Kepahiang. **Unconfirmed localities:** (a) Thailand, Phetchaburi, Kaeng Krachan N. P.; (b) Laos, Champasak, Xe Pian N.P.A.; (c) Vietnam, Quang Binh, Phong Nha - Ke Bang N.P.. Base Map created using simplemappr.net.

Permissions to conduct fieldwork and collect specimens were granted by the Department of Forestry, Ministry of Agriculture and Rural Development of Vietnam (permit numbers #547/TCLN-BTTN; #432/TCLN-BTTN; #822/TCLN-BTTN; #142/SNgV-VP; #1539/TCLN-DDPH; #1700/UBND.VX); the Forest Protection Departments of the Peoples' Committees of Gia Lai Province (permit numbers #530/UBND-NC; #1951/UBND-NV), Phu Yen Province (permit number #05/UBND-KT); Phu Tho Province (permit number #2394/UBND-TH3); Thanh Hoa Province (permit number #3532/UBND-THKH); and Quang Nam Province (permit number #308/SNgV-LS), Vietnam; by the Biotechnology and Ecology Institute Ministry of Science and Technology, Lao PDR (permit no. 299); and by the Institute of Animals for Scientific Purpose Development (IAD), Bangkok, Thailand (permit numbers U1-01205-2558 and UP-AE59-01-04-0022). Specimen collection protocols and animal operations followed the Institutional Ethical Committee of Animal Experimentation of University of Phayao (permit number 610104022).

## DNA isolation, PCR, and sequencing

To infer the phylogenetic relationships among the Pareinae we obtained partial sequence data of cytochrome *b* (cyt *b*) and NADH dehydrogenase subunit 4 (*ND4*) mtDNA genes, as well as two nuclear genes: oocyte maturation factor mos (*c-mos*) and recombination activating gene 1 (*RAG1*). These genetic markers have been widely applied in studies of Pareidae diversity and phylogenetic relationships (*e.g.*, *Guo et al., 2011*; *You, Poyarkov & Lin, 2015*; *Deepak, Ruane & Gower, 2019*; *Wang et al., 2020*; *Vogel et al., 2020, 2021*; *Ding et al., 2020*). Total genomic DNA was extracted from muscle or liver tissue samples preserved in 95% ethanol using standard phenol-chloroform-proteinase K (final concentration 1 mg/ml) extraction procedures with consequent isopropanol precipitation (protocols followed *Russell & Sambrook, 2001*). DNA amplification was performed in 20 ml reactions using ca. 50 ng genomic DNA, 10 nmol of each primer, 15 nmol of each dNTP, 50 nmol of additional $MgCl_2$, Taq PCR buffer (10 mM of Tris–HCl, pH 8.3, 50 mM of KCl, 1.1 mM of $MgCl_2$, and 0.01% gelatine) and 1 U of Taq DNA polymerase. Primers used for PCR and sequencing are summarized in Table S2.

PCRs were run on a Bio-Rad T100TM Thermal Cycler. PCR protocols for cyt *b* and *ND4* gene fragments followed *De Queiroz, Lawson & Lemos-Espinal (2002)* and *Salvi et al. (2013)*, respectively; the cycling parameters for *c-mos* gene were identical to those described in *Slowinski & Lawson (2002)*, and for *RAG1* to those described in *Groth & Barrowclough (1999)* and *Chiari et al. (2004)*. Sequence data collection and visualization were performed on an ABI 3730xl automated sequencer (Applied Biosystems, Foster City, CA, USA). PCR purification and cycle sequencing were done commercially through Evrogen Inc. (Moscow, Russia).

## Phylogenetic analyses

Sequences were managed and edited manually using Seqman in Lasergene. v7.1 (DNASTAR Inc., Madison, WI, USA), MEGA 7 (*Kumar, Stecher & Tamura, 2016*), and BioEdit v7.0.5.2 (*Hall, 1999*). For individuals which were detected to be heterozygous in nuclear gene sequences, they were phased using the software program PHASE with default sets of iterations, burn-in, and threshold (*Stephens, Smith & Donnelly, 2001*), on the web-server interface SEQPHASE (*Flot, 2010*). One of the phased copies was selected at random to represent each individual in subsequent analyses. All sequences were deposited in GenBank (see Table S1 for accession numbers).

To reconstruct the phylogenetic relationships within the Pareinae, we aligned the newly obtained cyt *b*, *ND4*, *c-mos*, and *RAG1* sequences together with representative sequences from 32 specimens of approximately 16 nominal *Pareas* species and seven other Pareinae representatives, retrieved from GenBank (see Table S1). Two species of the genus *Xylophis* (Pareidae: Xylophinae) were added to the alignment and used as outgroups for rooting the phylogenetic tree following the phylogenetic data of *Deepak, Ruane & Gower (2019)* and *Deepak et al. (2020)*. In total, we obtained molecular genetic data for 81 samples representing 38 taxa of Pareinae, including all currently recognized species of the genus *Pareas*, five species of *Asthenodipsas*, and the single species of the genus *Aplopeltura* (*A. boa*). Details on taxonomy, localities, GenBank accession numbers, and associated references for all examined specimens are summarized in Table S1.

The nucleotide sequences were initially aligned in MAFFT v.6 (*Katoh et al., 2002*) with default parameters; the alignment was subsequently checked by eye in BioEdit 7.0.5.2 (*Hall, 1999*) and slightly adjusted. The mean uncorrected genetic *p*-distances between sequences were calculated with MEGA 7 (*Kumar, Stecher & Tamura, 2016*). Phylogenetic trees were estimated for the combined mitochondrial DNA fragments (cyt *b* and *ND4*) and nuclear gene (*c-mos* and *RAG1*) datasets. The total evidence analysis was performed as the approximately unbiased tree-selection test (AU-test; *Shimodaira, 2002*) conducted using Treefinder v.March 2011 (*Jobb, 2011*) did not reveal statistically significant differences between mtDNA and nuDNA topologies.

Phylogenetic relationships of Pareinae were inferred using Bayesian Inference (BI) and Maximum Likelihood (ML) approaches. The optimum partitioning schemes for alignments were identified with PartitionFinder 2.1.1 (*Lanfear et al., 2012*) using the greedy search algorithm under an AIC criterion, and are presented in Table S3. When the

same model was proposed to different codon positions of a given gene, they were treated as a single partition.

BI was performed in MrBayes v3.1.2 (*Ronquist & Huelsenbeck, 2003*) with two simultaneous runs, each with one cold chain and three heated chains for 200 million generations. Two independent Metropolis-coupled Markov chain Monte Carlo (MCMCMC) runs were performed and checked for the effective sample sizes (ESS) were all above 200 by exploring the likelihood plots using TRACER v1.6 (*Rambaut & Drummond, 2007*). We discarded the initial 10% of trees as burn-in. Confidence in tree topology was assessed by posterior probability for Bayesian analysis (BI PP) (*Huelsenbeck & Ronquist, 2001*). Nodes with BI PP values of 0.95 and above were considered strongly supported, nodes with values of 0.90–0.94 were considered as well-supported, and the BI PP values below 0.90 were regarded as no support (*Wilcox et al., 2002*).

A Maximum Likelihood (ML) analysis was implemented using the IQ-TREE webserver (*Nguyen et al., 2015*; *Trifnopoulos et al., 2016*). One-thousand bootstrap pseudoreplicates *via* the ultrafast bootstrap (ML UB; *Hoang et al., 2018*) approximation algorithm were employed, and nodes having ML UB values of 95% and above were considered strongly supported, while nodes with values of 90–94% we regarded as well-supported, and the ML UB node values below 90% were considered as no support (*Minh, Nguyen & von Haeseler, 2013*).

## Divergence times estimation

The time-calibrated Bayesian Inference analysis was implemented in the program Bayesian Evolutionary Analysis Utility (BEAUti) version 2.4.7 and run on BEAST v1.8.4 (*Drummond et al., 2012*), including the concatenated mtDNA + nuDNA dataset. We used hierarchical likelihood ratio tests in PAML v4.7 (*Yang, 2007*) to test molecular clock assumptions separately for mtDNA and nuDNA markers. Based on PAML results, which indicated that there was very little rate variation among the sites of mtDNA markers and so a strict clock model was used for the final analysis employing unlinked site and linked tree models for the nuDNA, and an uncorrelated lognormal relaxed clock for mtDNA genes. We also used these models and partitioning schemes from the ML analysis with empirical frequencies estimated so as to fix them to the proportions observed in the data. A coalescent exponential population prior was employed as the tree prior because intraspecific relationships among many individuals were being assessed and it was not known *a priori* which individuals would be grouped as species. Under the coalescent model, the default priors for population growth (Laplace Distribution) and size (1/X) were left unchanged because these parameters were not being estimated. We conducted two runs of 100 million generations each in BEAST v1.8.4. We also assumed parameter convergence in Tracer and discarded the first 10% of generations as burn-in. We used TreeAnnotator v1.8.0 (in BEAST) to create our maximum credibility clades. Since no paleontological data for the Pareidae are known to exist, we relied on four recently estimated calibration priors for this family obtained from recent large-scale phylogeny of the group (*Deepak, Ruane & Gower, 2019*) as primary calibration points. Calibration points and priors are summarized in Table S4.

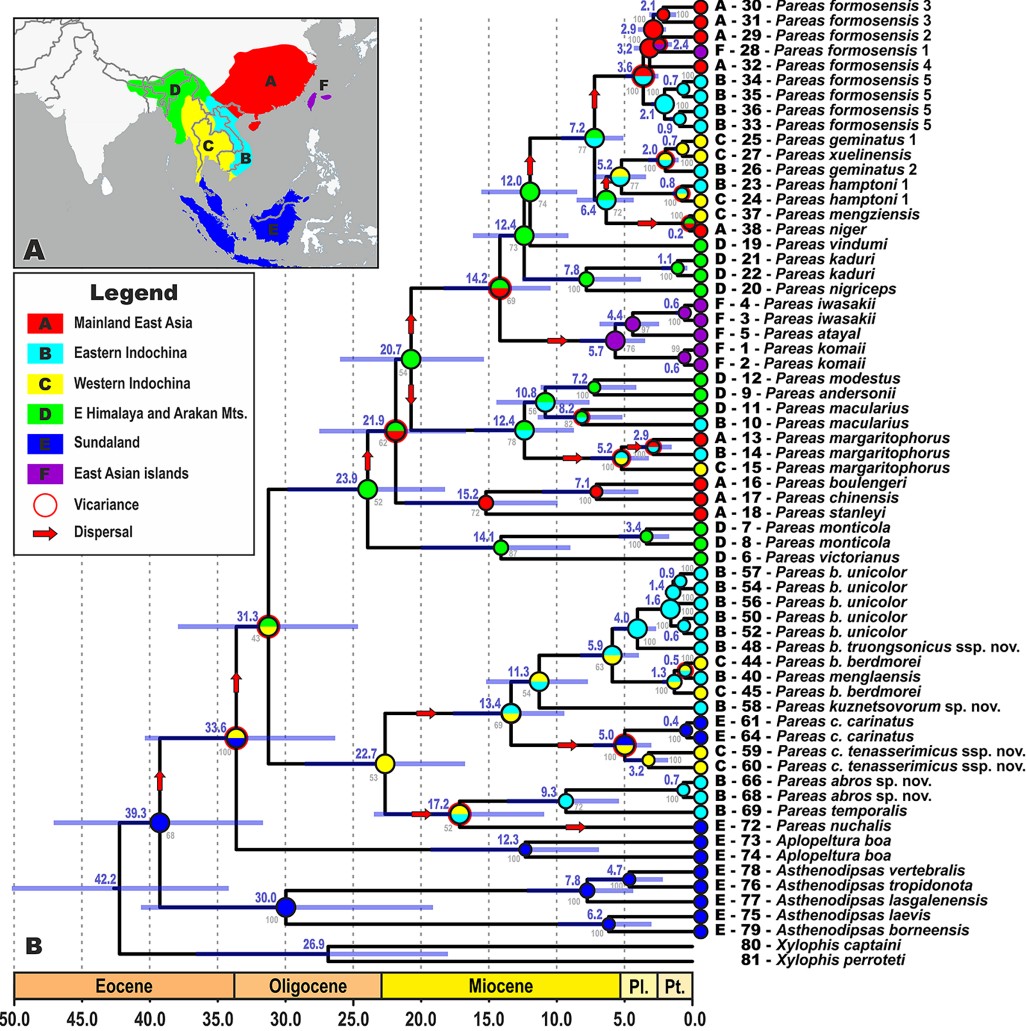

**Figure 2 Time tree and biogeographic history of the subfamily Pareinae.** (A) Biogeographic regions used in the present study; (B) BEAST chronogram on the base of 3588 bp–long mtDNA + nuDNA dataset with the results of ancestral area reconstruction using Langrange Dispersal-Extinction-Cladogenesis (DEC) model in RASP. For biogeographic areas definitions, species occurrence data and transition matrices see Tables S5 and S6. Information at tree tips corresponds to biogeographic area code (A), sample number (summarized in Table S1), and species name, respectively. Node colors correspond to the respective biogeographic areas; values inside node icons correspond to node numbers (see Table S9 and Fig. S1 for divergence time estimates); values in grey near nodes indicate marginal probabilities for ancestral ranges (S–DIVA analysis), values in blue near nodes correspond to median time of divergence (see Table S9); icons illustrate vicariant and dispersal events (see legend). Base Map created using simplemappr.net.

## Biogeographic analyses

The biogeographic range evolution history of Pareinae was reconstructed by a model-testing approach in a common ML framework to find the best statistical fit using AIC in RASP v3.2 (*Ree et al., 2005*; *Ree & Smith, 2008*; *Yu et al., 2015*). The models allow testing alternative biogeographic hypotheses, such as dispersal, vicariance, and extinction. Six areas were defined that are covered by our ingroup sample (see Fig. 2A): (A) Mainland East Asia; (B) Eastern Indochina; (C) Western Indochina; (D) Indo-Burma,

including eastern Himalaya and the Arakan Mountains of Myanmar; (E) Sundaland; and (F) East Asian islands (Taiwan + the Ryukyus) following *Gorin et al. (2020)*, *Chen et al. (2018)*, and *Nguyen et al. (2020a)*. This coding scheme reflects the complex palaeogeographic history of Southeast Asia, because Borneo, Java, Sumatra and the Thai-Malay Peninsula constituted the connected landmass of Sundaland until recently (*Hall, 2012*; *Morley, 2018*). Maximum areas per species were set to three, as no extant species occurs in more than three biogeographical regions. Matrices of modern distributions of species across the areas are presented in Table S5; transition matrices between biogeographic regions are given in Table S6. Discrete state transitions for ranges were estimated using ML framework on branches as functions of time, suggesting the best fit model for ancestral ranges at the times of cladogenesis using the Akaike Information Criterion (AIC) and Akaike weights (*Ree & Smith, 2008*; *Matzke, 2013*). Two models were compared: Langrange Dispersal-Extinction-Cladogenesis (DEC; *Ree & Smith, 2008*), and the ML version of Statistical Dispersal-Vicariance Analysis (S-DIVA; *Ronquist, 1997*).

## Morphological characteristics and analyses

For this study, a total of 270 preserved specimens of the subfamily Pareinae, including 82 specimens of the subgenus *Pareas*, were examined for their external morphological characters (Table S11, Appendix S2).

A total of 46 morphological and chromatic characters were recorded for each specimen (following *Vogel, 2015*). Morphological measurements (all in mm) included: snout-vent length (SVL); tail length (TaL); total length (TL); relative tail length (TaL/TL); horizontal eye diameter (ED); distance from the anterior edge of orbit to nostril (Eye-nos); minimal distance from the ventral edge of orbit to the edge of upper lip (Eye-mouth); head length from snout tip to jaw angles (HL); maximal head width (HW). Meristic characters evaluated were the number of dorsal scale rows counted at one head length behind head (ASR), at mid-body (MSR), namely at SVL/2, and at one head length before vent (PSR); number of enlarged vertebral scale rows (VSE); presence of keeled dorsal scale rows (DORkeel); number of keeled dorsal scale rows at midbody (KMD); number of ventral scales (VEN); number of preventral scales (preVEN); number of subcaudal scales (SC); number of cloacal (anal) plates (AN); number of supralabials (SL); number of supralabials touching the orbit (SL-eye); number of supralabials touching subocular (SL-suboc); number of infralabials (IL); numbers of infralabials touching each other (IL-touch); number of nasals (NAS); number of anterior temporals (At); number of posterior temporals (Pt); number of loreals (LOR); loreal touching the orbit or not (LOR-eye); number of preoculars (Preoc); number of presuboculars (Presuboc); prefrontal touching the orbit or not (Prefr-eye); number of suboculars (SoO); subocular fused with postocular or not (SoO-PoO); number of postoculars (PoO). Coloration and pattern characters evaluated were the background body dorsal coloration; presence or absence of ornamentation on neck; presence or absence of dark blotches or chevron on neck and nuchal areas; coloration of head dorsal surface; presence and number of postorbital stripes; presence or absence of a dark blotch on 7th supralabial; presence or absence of transverse

bands on body; number of transverse bands on body; number of discontinuous dorsal bands comprised of dark dots; presence or absence of body ornamentation others than bands; dorsal bands continue on belly or not; belly pattern (no pattern, banded, mottled or dotted).

We took the color notes from living specimens or their digital images prior to preservation following *Vogel (2015)*. We measured body and tail lengths with a measuring tape (to the nearest of 1 mm); all other measurements were taken using a Mitutoyo digital slide-caliper (to the nearest 0.1 mm). We counted the number of ventral scales following *Dowling (1951)*; we regarded the first enlarged shield anterior to the ventrals as a preventral, while half-ventrals were counted as one. We regarded the first scale under the tail contacting its opposite as the first subcaudal. We did not include the terminal tail scute in the number of subcaudals. Supralabial(s) touching the presubocular were included in the SL-suboc count. We regarded as infralabials those shields that were bordering the mouth gap and were placed ventrally than supralabials. Smaller shields located posteriorly than the last enlarged supralabial shield do not border the mouth gap and were excluded in the sublabial scales count. We defined the scale starting between the posterior chin shield and the infralabials and bordering the infralabials as the first sublabial shield. We recorded the values for paired head characters on both sides of the head (in a left/right order). We determined the sex of the specimens by the presence of everted hemipenes or by dissection of the ventral tail base. We described the morphology of hemipenial structures on specimens in which such structures were everted before preservation; description and terminology followed *Keogh (1999)*.

An analysis of variance (ANOVA) was performed to ascertain if statistically significant mean differences among meristic characters ($p < 0.05$) existed among the discrete populations delimited in the phylogenetic analyses. ANOVAs having a $p$-value less than 0.05 indicating that statistical differences existed were subjected to a Tukey HSD test to ascertain which population pairs differed significantly ($p < 0.05$) from each other. We used the Principal Component Analysis (PCA) to determine if populations from different localities occupied unique positions in morphospace, as well as the degree to which their variation coincided with potential species boundaries as predicted by the molecular phylogeny and univariate analyses. Juvenile specimens, as well as the specimens with incomplete or damaged tails were excluded from the PCA. Characters used in the PCA were continuous mensural data from SVL, TaL, TL, ED, Eye-nos, Eye-mouth, HL, and HW, and the discrete meristic data from the scale counts VSE, KMD, DORkeel, VEN, preVEN, SC, SL, SL-eye, IL, At, Pt, LOR, Preoc, Presuboc, Prefr-eye, SoO, SoO-PoO, and PoO. In order to normalize the PCA data distribution and to transform meristic and mensural data into comparable units for analysis, we natural log-transformed all PCA data prior to analysis and scaled it to their standard deviation. To exclude possible overweighting effects, when we found a high correlation between certain pairs of characters, we omitted one of them from the analyses. Statistical analyses were carried out using Statistica 8.0 (Version 8.0; StatSoft, Tulsa, OK, USA).

Morphological and coloration characters of the examined specimens were compared in detail to other species of the genus the *Pareas*. The examined comparative material is listed

in *Appendix S2*. For comparison with other taxa, we also relied on previously published data (*e.g.*, *Theobald, 1868b*; *Bourret, 1934*; *Pope, 1935*; *Smith, 1943*; *Taylor, 1965*; *Guo & Zhao, 2004*; *Guo & Deng, 2009*; *Stuebing, Inger & Lardner, 2014*; *You, Poyarkov & Lin, 2015*, *Vogel, 2015*; *Hauser, 2017*; *Wang et al., 2020*; *Vogel et al., 2020, 2021*; *Ding et al., 2020*; *Bhosale et al., 2020*; *Liu & Rao, 2021*; *Le et al., 2021*). Other abbreviations used: Prov.: Province; Mt.: Mountain; N.P.: National Park; N.R. Natural Reserve; Is.: Island; asl: above sea level.

## RESULTS

### Partitions, substitution models, and sequence characteristics

Our combined dataset was composed of 1,804 bp of cyt *b* and *ND4* mtDNA genes, 1,757 bp of nuDNA (including 734 bp of *c-mos*, and 1,023 bp of *RAG1*), and 3,561 bp (mtDNA + nuDNA), respectively. The concatenated mtDNA + nuDNA dataset included 81 samples, representing ca. 29 *Pareas* taxa, including all 24 currently recognized species of the genus (*Uetz, Freed & Hošek, 2021*), one species of the monotypic genus *Aplopeltura*, five species of the genus *Asthenodipsas* (of nine currently recognized species, 56%), and two outgroup taxa (see Table S1). Information on fragment lengths and variability is summarized in Table S3. PartitionFinder 2.1.1 proposed the partition schemes and substitution models which resulted in nine partitions in total (Table S3).

### Phylogenetic relationships and distribution

Phylogenetic trees obtained with ML and BI analyses of the three data partitions (mtDNA + nuDNA, mtDNA, nuDNA) are congruent apart from the generally lower resolution of nuDNA trees (see Figs. S2, S3). Overall, since the mtDNA + nuDNA phylogenetic tree was mostly better resolved and had greater node support than the mtDNA and nuDNA trees, we relied on the combined mtDNA + nuDNA topology for inferring phylogenetic relationships and biogeographic history of Pareinae. The BI tree resulted from the analysis of the concatenated mtDNA + nuDNA data (Fig. 3) inferred the following set of phylogenetic relationships:

1. The subfamily Pareinae was subdivided into five major strongly supported, deeply divergent groups, including two groups within the genus *Pareas sensu lato* (clades A and B, see Fig. 3), the genus *Aplopeltura* (clade C, see Fig. 3), and two groups corresponding to the genus *Asthenodipsas sensu lato* (clades D and E, see Fig. 3).

2. The monophyly of the genus *Asthenodipsas* got strong support in mtDNA + nuDNA analysis (1.0/100; hereafter node support values are given for BI PP/ML UB, respectively; see Fig. 3), while it was rendered paraphyletic in the analysis of the mtDNA dataset alone, though with no significant node support. The two clades within *Asthenodipsas* correspond to the *A. malaccana* species group (clade E, in our analysis represented by *A. laevis* and *A. borneensis*; 1.0/100), and to the *A. vertebralis* species group (clade D, in our analysis including *A. vertebralis*, *A. tropidonota*, and *A. lasgalenensis*; 1.0/100).

3. The monophyly of the clade joining *Pareas* + *Aplopeltura* was strongly supported (1.0/99). The monotypic genus *Aplopeltura* (1.0/100) in our analysis was represented

with two samples of *A. boa* from Peninsular Malaysia and Borneo (Sabah, Malaysia), which were assigned into two highly divergent lineages (see Fig. 3).

4. The monophyly of the genus *Pareas sensu lato* was strongly supported by all analyses (1.0/99); the genus comprised two reciprocally monophyletic highly supported groups: clade A, including the members of the *P. carinatus – P. nuchalis* complex (1.0/100); and clade B, including the remainder of *Pareas* species (1.0/100) (see Fig. 3).

5. Within the clade B, encompassing the majority of the genus *Pareas* diversity, four subclades were recovered corresponding to the following species groups:

 a) ***Pareas hamptoni* species group** (subclade B1; 1.0/100) including *P. formosensis*, *P. xuelinensis*, *P. geminatus*, *P. hamptoni*, *P. niger*, *P. mengziensis*, *P. iwasakii*, *P. atayal*, *P. komaii*, *P. vindumi*, *P. kaduri*, and *P. nigriceps*. *Pareas kaduri* and *P. nigriceps* from East Himalaya formed a well-supported monophylum (1.0/99). The three species of *Pareas* from the East Asian Islands also formed a well-supported clade (1.0/100); with *P. komaii* reconstructed as a sister species with respect to *P. atayal* + *P. iwasakii* though with a low nodal support (0.56/88). Phylogenetic position of *P. vindumi* from Myanmar within the subclade B1 remained essentially unresolved (Fig. 3). The remaining species of the subclade B1 formed a well-supported clade, corresponding to the *P. hamptoni* species complex (1.0/100). Within the latter, *P. niger* and *P. mengziensis* from Yunnan Province of China grouped together (1.0/100) and were represented with almost identical haplotypes. *Pareas hamptoni* from Myanmar and Northern Indochina was suggested as a sister taxon with respect to the clade joining *P. xuelinensis* from Yunnan and *P. geminatus* from Northern Indochina; the latter species was recovered as paraphyletic with respect to *P. xuelinensis* (1.0/100). *Pareas formosensis* was represented in our analysis with five major lineages from Taiwan and Hainan islands, southern mainland China and Eastern Indochina; the sample of topotype *P. tonkinensis* from northern Vietnam was placed within the *P. formosensis* radiation with strong support (0.99/98; see Fig. 3).

 b) ***Pareas margaritophorus* species group** (subclade B2; 1.0/100) included four species from Indochina and Indo-Burma: *P. andersonii*, *P. modestus*, *P. macularius*, and *P. margaritophorus*. *Pareas andersonii* and *P. modestus* from Myanmar and Northeast India formed a well-supported clade (1.0/100), to which *P. macularius* (1.0/100) was recovered as a sister taxon. The latter species was represented in our analysis with two samples from Myanmar and Laos, which were assigned into two highly divergent lineages (see Fig. 3). Subclade B2 was suggested as a sister lineage with respect to subclade B1 though with significant nodal support (0.99/89; see Fig. 3).

 c) ***Pareas chinensis* species group** (subclade B3; 1.0/100) included *P. stanleyi*, *P. boulengeri*, and *P. chinensis* from mainland China; the latter two species formed a strongly supported monophyletic group (1.0/100). Subclade B3 was suggested as a

sister lineage with respect to the clade joining B1 + B2 with strong nodal support (1.0/92; see Fig. 3).

d) ***Pareas monticola* species group** (subclade B4; 1.0/100) included two species from East Himalaya and Indo-Burma: *P. monticola* and *P. victorianus*. Subclade B4 was suggested as a sister taxon with respect to other species groups B1–B3 with strong node support (1.0/100; see Fig. 3).

6. Within the remainder of the genus *Pareas* (clade A; Fig. 3), unexpectedly high numbers of divergent evolutionary lineages were detected. Present taxonomy recognizes three species within this group: *P. carinatus*, *P. menglaensis* and *P. nuchalis*. Altogether, nine divergent lineages were distinguished by robust BI PP and ML UB node support in analyses of the combined mtDNA + nuDNA dataset (Fig. 3). Of these lineages, those which are presently assigned to *P. carinatus*, were recovered as paraphyletic with respect to both *P. menglaensis* and *P. nuchalis*. The nine evolutionary lineages revealed within clade A were distributed across two major clades, which we name herein: the *P. carinatus* species group (A1, lineages 1–6), and the *P. nuchalis* species group (A2, lineages 7–9; see Fig. 3):

a) ***Pareas carinatus* species group** (subclade A1; 1.0/100) comprised six lineages formerly recognized under the sole combination *P. carinatus*, including the populations from Peninsular Malaysia southwards from the Isthmus of Kra, corresponding to *P. carinatus sensu stricto* (1.0/100; lineage 5, see Fig. 3). Two samples from Tenasserim Mountains in Peninsular Thailand and Myanmar northwards from the Isthmus of Kra formed a monophyletic group (1.0/100; lineage 6, see Fig. 3), which represented the sister clade to the Malayan *P. carinatus sensu stricto* (1.0/100). The populations of *P. carinatus* from the mainland Indochina formed a monophyletic group (1.0/100), including three well-supported subgroups: (1) populations from lowlands of southern Vietnam, corresponding to the subspecies *P. carinatus unicolor* (*Bourret, 1934*) (1.0/100; lineage 1, see Fig. 3); (2) populations from the northern portion of Annamite (Truong Son) Mountains in central Vietnam and Laos (1.0/100; lineage 2, see Fig. 3); (3) populations from montane areas of Western Indochina (1.0/100; lineage 3, see Fig. 3), including the recently described *P. menglaensis* from southern Yunnan (locality 22, samples 39–43), and the topotypic specimen of *P. berdmorei* Theobald, 1868 from Mon State, Myanmar (locality 17, sample 45). Finally, a single specimen initially identified as *P.* cf. *carinatus* from Phu Yen Province in southern part of Central Vietnam formed a divergent lineage with sister relationships with all other populations of *P. carinatus* species group members from the mainland Indochina (1.0/94; lineage 4, see Fig. 3).

b) ***Pareas nuchalis* species group** (subclade A2; 0.99/80) got moderate node support level in the ML analysis since *P. nuchalis* from Borneo was only represented in our work by the single partial sequence of ND4 mtDNA gene (lineage 9, see Fig. 3). The two reciprocally monophyletic lineages from montane areas of Vietnam initially

identified as *P.* cf. *carinatus* formed a well-supported clade (1.0/100) which is unexpectedly only distantly related to other mainland Southeast Asian members of *Pareas* and supposedly more closely related to *P. nuchalis*: the lineage from Kon Tum – Gia Lai Plateau in Central Annamites (1.0/100; lineage 7, see Fig. 3), and the lineage from Langbian Plateau in Southern Annamites, corresponding to the recently described *P. temporalis* (1.0/100; lineage 8, see Fig. 3).

Distribution of the phylogenetic lineages within the clade A is presented in Fig. 1. Most lineages that cluster together in each of our two major subclades A1 and A2 are allopatrically distributed within the clade (Fig. 1). Two lineages from different subclades are found sympatrically: *P. nuchalis* (lineage 9) occurs in sympatry with *P. carinatus sensu stricto* (lineage 5) in Borneo and Sumatra, while *P.* cf. *carinatus* (lineage 1) occurs syntopically with lineage 8 of *P. temporalis* in Langbian Plateau of southern Vietnam (locality 37, Fig. 1). The only case of distribution overlap of lineages belonging to the same species group includes the lineages 3 and 6 of *P. carinatus* which are occur sympatrically in Suanphueng area of Ratchaburi Province in western Thailand (locality 14, Fig. 1). However in Suanphueng the co-occurring lineages of *Pareas* have clearly different habitat preferences and are not syntopically distributed: vouchers of lineage 3 were recorded in lowland bamboo forest at 300 m asl., while the voucher of lineage 6 was collected in the montane forest at ca. 800–1,000 m asl.

## Sequence divergence

The interspecific uncorrected genetic *p*-distances in cyt *b* and *ND4* mtDNA genes within the genus *Pareas* are summarized in Tables S7 and S8, respectively. For cyt *b* gene genetic divergence varied from $p = 4.1\%$ (between *P. geminatus sensu stricto* and *P. xuelinensis*) to $p = 25.2\%$ (between *P. kaduri* and lineage 7 of *P.* cf. *carinatus* from Central Annamites) (Table S7). For *ND4* gene *p*-distances varied from $p = 5.2\%$ (between lineages 1 and 2 of the *P. carinatus* complex) to $p = 23.7\%$ (between *P. nuchalis* and *P. komaii*) (Table S8). In several cases the intraspecific distances within *Pareas* species were greater than the minimal interspecific divergence values, which is likely explained with the incompletely known taxonomy of the group: lineage 6 of the *P. carinatus* complex from Tenasserim (5.0/4.2, hereafter values correspond to intraspecific distances for cyt *b*/*ND4* genes), *P. geminatus sensu lato* (7.2/–), *P. macularius* (11.5/10.4), *P. margaritophorus* (5.2/4.9), and *P. monticola* (3.7/5.7).

## Divergence times estimation

The time-calibrated BEAST analysis recovered a phylogeny with well-supported nodes (BPP ≥ 90) throughout the tree, topologically identical to the BI tree (Fig. 2; Fig. S1). The phylogeny indicates that the most recent common ancestor (MRCA) of Pareinae originated in late Eocene (Fig. 2). Basal radiation of Pareinae likely happened during the late Eocene at approximately 39.3 mya, the group continued to radiate across Asia up until the Pleistocene (Fig. 2). Diversification of the genera *Asthenodipsas* and *Pareas* started during the early Oligocene (30.0 mya and 31.3 mya, respectively). The major lineages

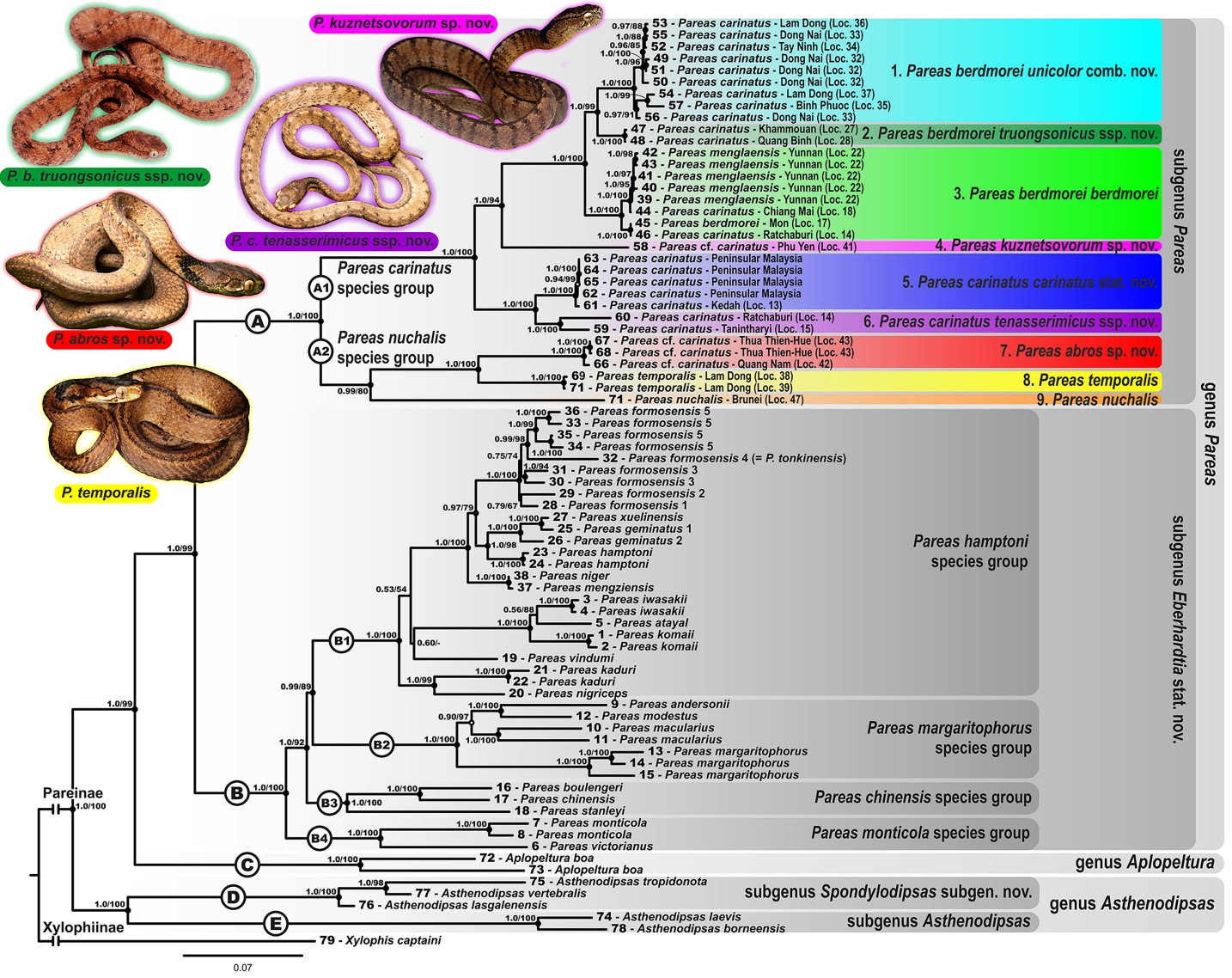

**Figure 3 Bayesian inference tree of the subfamily Pareinae derived from the analysis of 1,126 bp of cyt *b*, 681 bp of *ND4*, 737 bp of *cmos*, and 1,026 bp of *RAG1* gene fragments.** For voucher specimen information and GenBank accession numbers see Table S1. Colors denote the taxa of the subgenus *Pareas* and correspond to the color of icons in Figs. 1 and 4. Numbers at tree nodes correspond to PP/UFBS support values, respectively. Photos on thumbnails by N. A. Poyarkov (*Pareas abros* **sp. nov.**, *P. temporalis*, and *P. kuznetsovorum* **sp. nov.**), and P. Pawangkhanant (*P. berdmorei truongsonicus* **ssp. nov.** and *P. carinatus tenasserimicus* **ssp. nov.**).

(*i.e.* species groups) within the genus *Pareas* diversified between approximately 24.0–12.4 mya with species-level radiations evolving up until 5.0–2.0 mya (Fig. 2). Estimated node-ages and the 95% highest posterior density (95% HPD) for the main nodes are summarized in detail in Table S9.

## Biogeography

All the trees generated in RASP analyses generally recovered the same ancestral range for each node, thus converging on the same biogeographical scenario (Fig. 2). Model comparisons showed that the Langrange Dispersal-Extinction-Cladogenesis (DEC) model

is the best fit to the data and most likely to infer the correct ancestral range at each node being the it had the highest and lowest log likelihood and AIC scores, respectively. Our analyses unambiguously suggested that the MRCA of Pareinae (node 3; Fig. S1; Fig. 2) most likely inhabited Sundaland, which is also reconstructed as an ancestral range for the genera *Asthenodipsas* and *Aplopeltura* (nodes 4 and 9, respectively; Fig. S1; Fig. 2). The split between *Pareas* and *Aplopeltura* is likely explained by a vicariant event between Sundaland and West Indochina (Fig. 2). The divergence between the two major clades within the genus *Pareas* coincides with a vicariance between Indo-Burma and Eastern Himalaya (ancestral range for clade A) and West Indochina (ancestral range for clade B) (Fig. 2). Major ancestral nodes within the *Pareas* clade A remained within Indo-Burma and Eastern Himalaya, from where its members at least three times widely dispersed to the mainland East Asia and further southwards to Indochina and independently twice eastwards to Taiwan and the Ryukyus (Fig. 2). *Pareas* clade B expanded its range to East Indochina and at least twice dispersed to Sundaland (see Fig. 2). Overall, our analysis suggests an "upstream" colonization hypothesis for the Pareinae (from island to continent; see *Filardi & Moyle, 2005*; *Jønsson et al., 2011*), and, thus, the subfamily appears to have originated in Sundaland and then colonized the mainland Asia.

## Morphological differentiation

The PCA of the morphological dataset on *P. carinatus* – *P. nuchalis* complex revealed that the most distant morphospatial separation occurs in *P. nuchalis* (lineage 9), *P. temporalis* (lineage 8), *P.* cf. *carinatus* lineages from Kon Tum – Gia Lai Plateau (lineage 7), from Phu Yen Province (lineage 4), and from Tenasserim (lineage 6); followed with general separation of the *P.* cf. *carinatus* lineage from northern Annamites (lineage 2) and cluster consisting of the lineages of *P. carinatus sensu stricto* from Sundaland (lineage 5), and *P.* cf. *carinatus* from western Indochina and Yunnan (lineage 3) and southern Vietnam (lineage 1) (Fig. 4). PC1 accounted for 18.5% of the variation in the data set and loaded most heavily for relative tail length, number of subcaudal scales, number of ventral scales, and number of prefrontals bordering eye (TaL/TL, SC, VEN, and Prefr-eye; Table S10). PC2 accounted for 14.8% of the variation in the data set and loaded most heavily for number of keeled dorsal scale rows, total length, head length, and tail length (KMD, TL, HL, and TaL; Table S10). The univariate and multivariate morphological analyses further supported results of the molecular analyses by indicating that the lineages within the *P. carinatus* – *P. nuchalis* complex are well separated from each other in morphospace and bear a number of statistically significant mean differences in varying combinations of meristic and color pattern characters, thus providing reliable diagnostic character differences among the species (Table 2; Tables S11–13).

### Systematics
### Genus-level taxonomy of Pareinae

All recent phylogenetic studies on caenophidian snakes agree on the monophyly of Pareidae (*Pyron, Burbrink & Wiens, 2013*; *Figueroa et al., 2016*; *Zaher et al., 2019*) and of the subfamily Pareinae with respect to Xylophiinae (*Deepak, Ruane & Gower, 2019*).

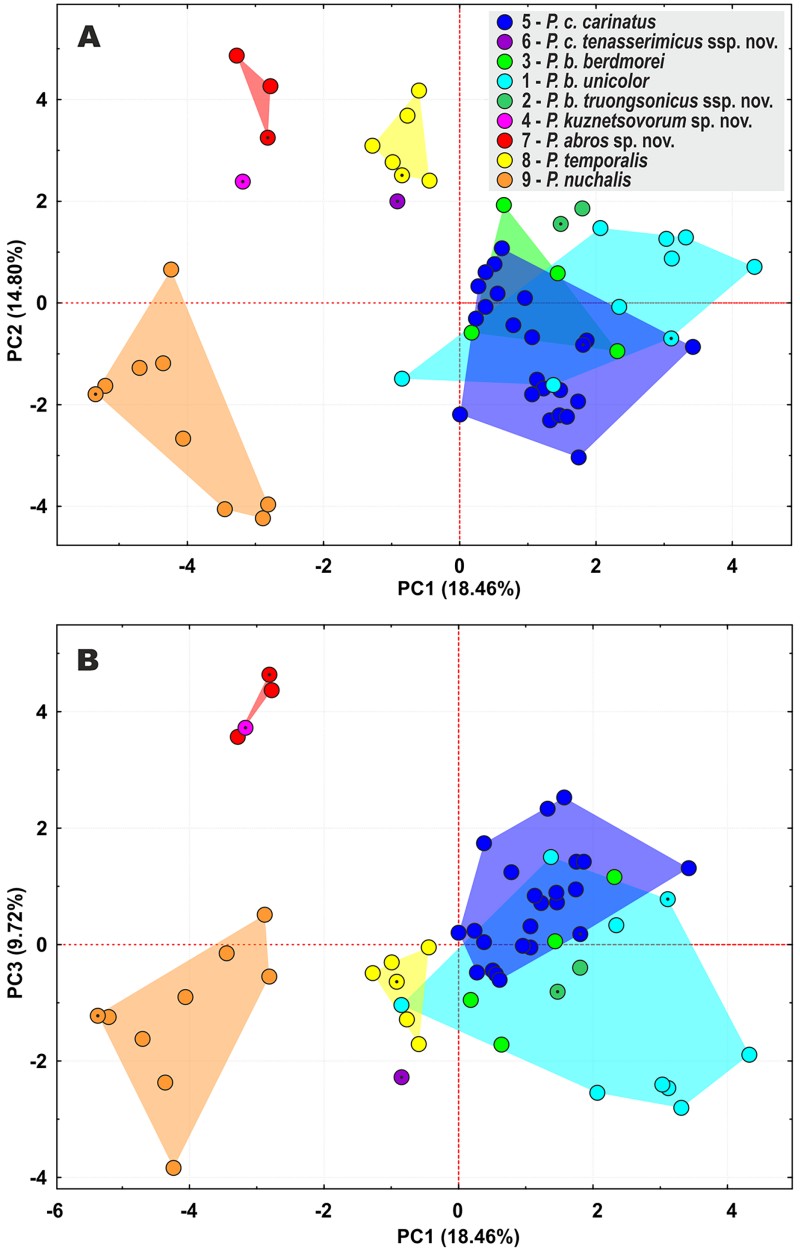

**Figure 4 Principal component analysis (PCA) of the species of the subgenus *Pareas* showing ordination along the first two (A) and the first and the third (B) principal components.** Colors denote the taxa of the subgenus *Pareas* and correspond to the color of icons in Figs. 1 and 3; dot in the center of an icon indicates the holotype or lectotype of a taxon.

Most works on phylogenetic relationships of this group agreed that the genus *Apolpeltura* is a sister taxon of *Pareas sensu lato*, however, the monophyly of the genera *Pareas* and *Asthenodipsas* has been questioned for a long time.

Several earlier studies demonstrated that the *P. carinatus* complex (including *P. nuchalis*) is phylogenetically distant from other members of the genus, which was recovered as paraphyletic (*e.g.*, *Guo et al., 2011*; *Pyron et al., 2011*). It was noted that these

**Table 2 Measurements and meristic characters of members of the subgenus *Pareas*: *Pareas abros* sp. nov., *P. kuznetsovorum* sp. nov., *P. carinatus*, *P. berdmorei*, *P. nuchalis* and *P. temporalis*.** Abbreviations are listed in the Materials and methods.

| Characters | | P. abros | P. kuznetsovorum | P. carinatus | P. berdmorei | P. nuchalis | P. temporalis |
|---|---|---|---|---|---|---|---|
| TL | Min–Max | 434–565 | 638.5 | 337–702 | 421–770 | 555–665 | 555–665 |
| | Mean ± SD | 506.7 ± 66.7 | | 494.3 ± 73.3 | 554.9 ± 73.3 | 577.1 ± 34.5 | 577.1 ± 34.5 |
| | *n* | 3 | 1 | 24 | 34 | 7 | 7 |
| TaL/TL | Min–Max | 0.26–0.29 | 0.25 | 0.18–0.25 | 0.17–0.27 | 0.20–0.26 | 0.20–0.26 |
| | Mean ± SD | 0.28 ± 0.01 | | 0.22 ± 0.02 | 0.21 ± 0.02 | 0.24 ± 0.02 | 0.24 ± 0.02 |
| | *n* | 3 | 1 | 24 | 32 | 7 | 7 |
| VEN | Min–Max | 180–184 | 167 | 158–194 | 162–187 | 185–198 | 185–198 |
| | Mean ± SD | 182.7 ± 2.3 | | 171.4 ± 9.3 | 176.9 ± 5.8 | 189.0 ± 4.4 | 189.0 ± 4.4 |
| | *n* | 3 | 1 | 26 | 38 | 7 | 7 |
| SC | Min–Max | 83–95 | 87 | 54–96 | 57–89 | 86–92 | 86–92 |
| | Mean ± SD | 90.0 ± 6.2 | | 69.3 ± 9.0 | 71.6 ± 7.3 | 88.7 ± 2.4 | 88.7 ± 2.4 |
| | *n* | 3 | 1 | 26 | 34 | 7 | 7 |
| KMD | Min–Max | 9–11 | 0 | 0–11 | 3–13 | 15 | 15 |
| | Mean ± SD | 10.3 ± 1.1 | | 6.5 ± 2.9 | 8.83 ± 2.76 | 15.0 ± 0.0 | 15.0 ± 0.0 |
| | *n* | 3 | 1 | 19 | 33 | 7 | 7 |
| VSC | Min–Max | 1 | 1 | 3 | 1–3 | 3 | 3 |
| | Mean ± SD | 1.0 ± 0.0 | | 3.0 ± 0.0 | 2.83 ± 0.56 | 3.0±0.0 | 3.0 ± 0.0 |
| | *n* | 3 | 1 | 26 | 38 | 7 | 7 |

genetic divergence are concordant with differences in a number of external morphology and scalation characters (*Guo et al., 2011*) and scale ultrastructure (*He, 2009*; *Guo, Wang & Rao, 2020*) (see Fig. 5). *Guo et al. (2011)* and *Guo, Wang & Rao (2020)* suggested that *P. carinatus* and *P. nuchalis* are different from other *Pareas* species in morphological, ultrastructural and molecular characteristics, and therefore, they "might be removed from the genus *Pareas*"; this idea was further supported by *Wang et al. (2020)*. However, due to incomplete sampling and insufficient morphological data, *Guo et al. (2011)*, *Guo, Wang & Rao (2020)* and *Wang et al. (2020)* refrained from making a formal taxonomic decision on the division of *Pareas* (Note - *Wang et al. (2020)* applied a new genus name 'Northpareas' to the clade including all *Pareas* species except *P. carinatus*, *P. nuchalis*, and *P. menglaensis*, however this name is only used in Appendix S3 of their paper and is not used in the text of their manuscript, and thus should be considered a nomen nudum). At the same time, both *Guo et al. (2011*; *Guo, Wang & Rao, 2020)* and *Wang et al. (2020)* have overlooked two taxonomic issues:

1. *Pareas carinatus Wagler, 1830* is the type species of the genus *Pareas Wagler, 1830*, and hence it cannot be placed to a different genus; in our analyses the name *Pareas Wagler, 1830* corresponds to the *Pareas* clade A (see Fig. 3).
2. *Eberhardtia Angel, 1920* is an available genus-level name, erected for *Eberhardtia tonkinensis Angel, 1920*, which was considered a junior synonym of *Pareas formosensis* (Van Denburgh, 1909) by *Ding et al. (2020)*; in our analyses it corresponds to the *Pareas* clade B (see Fig. 3).

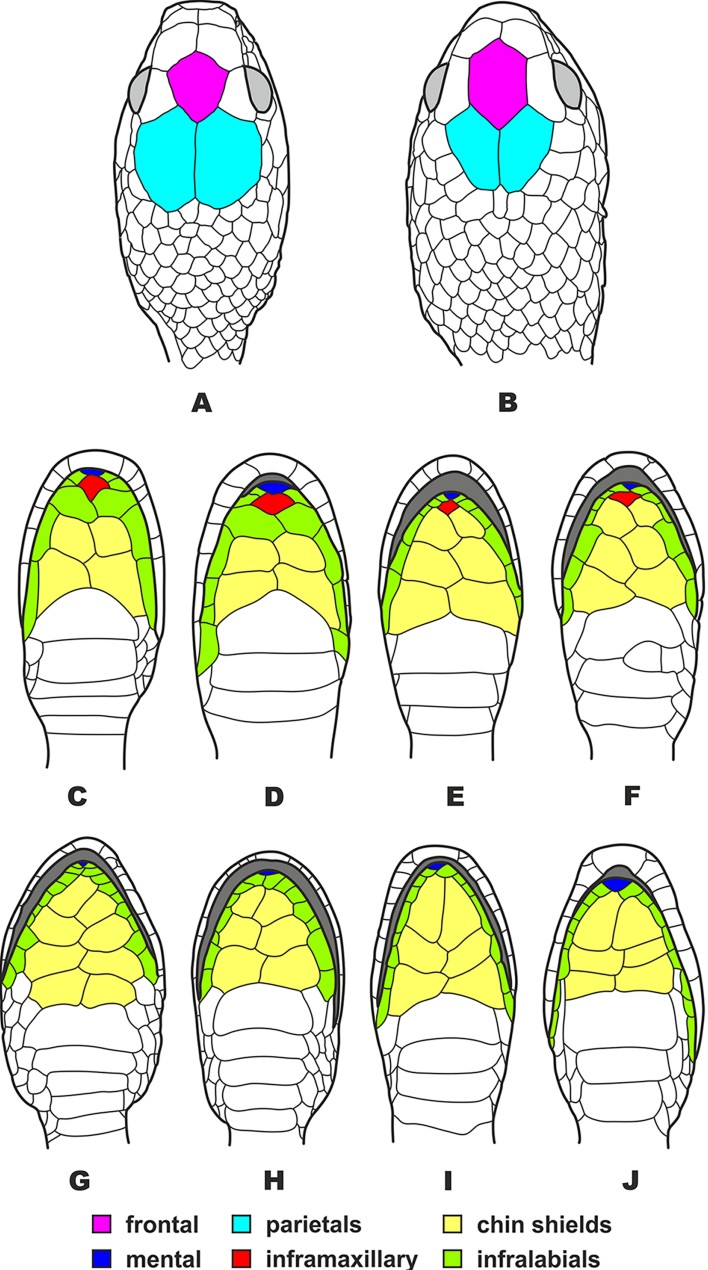

**Figure 5 Head scalation of the genera of the subfamily Pareinae.** Dorsal aspect: (A) *Pareas (Eberhardtia) formosensis* (FMNH 2555567); (B) *Pareas (Pareas) carinatus* (RMNH 954C, lectotype); Ventral aspect: (C) *Asthenodipsas (Asthenodipsas) malaccana* (SMF 32580); (D) *Asthenodipsas (Asthenodipsas) laevis* (SMF 81195); (E) *Asthenodipsas (Spondylodipsas* **subgen. nov.***) vertebralis* (ZMB 65285); (F) *Asthenodipsas (Spondylodipsas* **subgen. nov.***) tropidonota* (RMNH 4902B, lectotype); (G) *Aplopeltura boa* (ZMB 5397); (H) *Pareas (Pareas) carinatus* (ZMB 5397); (I) *Pareas (Eberhardtia) formosensis* (ZMB 30585); (J) *Pareas (Eberhardtia) margaritophorus* (ZMB 6339). Not to scale. Magenta, cyan, blue, red, green and yellow denote frontal, parietals, mental, inframaxillary, infralabials, and chin shields, respectively. Drawings by N. A. Poyarkov (A and B) and L. B. Salamakha (C–J).

Our phylogenetic analyses confirmed the existence of two highly divergent reciprocally monophyletic clades within *Pareas sensu lato*, while strongly supporting the monophyly of the genus.

The genus *Asthenodipsas Peters, 1864* (type species – *Asthenodipsas malaccana Peters, 1864*) was for the long time considered a junior synonym of *Pareas*. *Rao & Yang (1992)* examined morphological differences and placed the two species *laevis* Boie, 1827, and *malaccana Peters, 1864* (that time members of the genus *Pareas*) to a newly erected genus *Internatus Rao & Yang, 1992* (type species – *Amblycephalus laevis* Boie, 1827). Further studies have also added the taxa *tropidonota* van Lidth de Jeude, 1923 and *vertebralis Boulenger, 1900* to this genus (*David & Vogel, 1996*; *Grossmann & Tillack, 2003*). However, *Iskandar & Colijn (2001)* noted that *Rao & Yang (1992)* had clearly overlooked an available name for the taxa assigned to their new genus *Internatus*, *i.e.*, *Asthenodipsas* (*Grossmann & Tillack, 2003*). A number of recent works described five additional species within the genus *Asthenodipsas*, without addressing questions of genus-level taxonomy of the group (*Loredo et al., 2013*; *Quah et al., 2019, 2020*; *Quah, Lim & Grismer, 2021*). Several studies demonstrated that *Asthenodipsas* includes two major highly divergent clades, one including *A. laevis* and *A. borneensis* (belonging to the *A. malaccana* species complex) (clade E in our analyses, see Fig. 3), and another including *A. vertebralis*, *A. tropidonota*, and *A. lasgalenensis* (clade D in our analyses, see Fig. 3). Hence, both names *Asthenodipsas* and *Internatus* are referred to the members of clade E, while clade D has no available genus-level name. In previous phylogenetic studies, monophyly of the genus *Asthenodipsas sensu lato* was not supported (*Guo et al., 2011*) or got only moderate level of node support (*Wang et al., 2020*; *Ding et al., 2020*; *Vogel et al., 2021*). In our analyses, monophyly of *Asthenodipsas sensu lato* was strongly supported in the concatenated analysis of mtDNA + nuDNA data (Fig. 3), while the analysis of mtDNA genes alone suggested paraphyly of the genus with respect to *Pareas + Aplopeltura* (Fig. S2). This genetic divergence among two clades of *Asthenodipsas* is concordant with significant differences in taxonomically valuable scalation characters, such as the number of chin shields and the number of infralabials in contact (see Fig. 5).

In summary, in the molecular phylogenetic analysis, Pareinae is obviously divided into five major branches: (A) *Pareas carinatus + P. nuchalis* complex, (B) other species of *Pareas*, (C) *Aplopeltura*, (D) *Asthenodipsas vertebralis* group, and (E) other species of *Asthenodipsas* (Fig. 3). Monophyly of both *Pareas sensu lato* (clades A + B) and *Asthenodipsas sensu lato* (clades D + E) is strongly supported, while *Asthenodipsas* does not seem to be monophyletic according to mtDNA data alone. Should all five major lineages of Pareinae be recognized as distinct taxa?

As we argue below, we find there to be substantial evidence supporting the treatment of the major clades within *Pareas sensu lato* and *Asthenodipsas sensu lato* as separate subgenera. The taxonomic framework ideally should be optimized for utility, reflecting monophyly of taxa and their differences in sets of biologically significant characters, as well as stability, reducing the need for additional taxonomic changes in future (*Vences et al., 2013*; *Gorin et al., 2021*). Although, the present evidence indicates that we can be confident in the respective monophyly of *Pareas sensu lato* and of *Asthenodipsas sensu lato*, it
should be noted that the basal radiations within the both genera are very old: the two clades of *Pareas* diverged in early Oligocene (ca. 31.3 mya), while the basal radiation within *Asthenodipsas* happened soon afterwards (ca. 30.0 mya). These estimates are comparable with the split between *Pareas sensu lato* and *Aplopeltura* (ca. 33.6 mya), and are of equal or greater age than many other Caenophidian genera (see *Zaher et al., 2019*). While taxon age is usually not taken into account in higher taxonomy, it is however desirable for taxa of equal rank to be of generally comparable age (*Hennig, 1966*; *Vences et al., 2013*; *Gorin et al., 2021*). In addition to their substantial age, a number of important characters of external morphology, scalation, and scale ultrastructure distinguish the major clades within *Pareas* and *Asthenodipsas*, allowing their recognition both in collections and in the field (summarized below in taxonomic accounts). Furthermore, there are pronounced differences in the patterns of geographical distribution among the five clades of Pareinae: our hypothesis of the biogeographic history of this subfamily demonstrated that while the whole group evolved in Sundaland, *Pareas* clade A likely originated in Himalaya and Indo-Burma, and further dispersed to East Asia and Indochina, while *Pareas* clade B likely originated in Western Indochina, from where it colonized Sundaland and Eastern Indochina (Fig. 2). The cumulative evidence suggests that the lack of taxonomic recognition for the major clades within the genera *Pareas* and *Asthenodipsas* would conceal information on the ancient divergence between these lineages, as well as the significant differences between them in a set of biologically relevant traits (summarized in Table S14).

We propose to recognize the clades A and B of *Pareas* and clades D and E of *Asthenodipsas* as separate subgenera. This would enhance the diagnosability of the respective taxa and make them more comparable units to other genera of Pareinae, and as a consequence fully stabilize the taxonomy of the subfamily. This taxonomic action would therefore be in accordance with all three primary Taxon Naming Criteria (TNCs): Monophyly, Clade Stability, and Diagnosability, as well as the secondary TNCs: Time Banding and Biogeography (see *Vences et al., 2013*). The use of subgenera seems has been successfully applied in several recent revisions of taxonomically challenging groups of reptiles, *e.g.* cobras of the genus *Naja* (*Wallach, Wuester & Broadley, 2009*), *Trimeresurus*, the Asian pit-vipers (*David, Vogel & Dubois, 2011*), and *Gekko* (geckoes) (*Wood et al., 2020*). *Wood et al. (2020)* argued that the defining subgenera may aide taxonomists in species descriptions by allowing them to only diagnose putatively new species from the most relevant members of the same subgenus. By creating formally available supraspecific taxa, accompanied by character-based diagnoses and properly assigned type species, the practice of recognizing subgenera also has the potential to restrain taxonomic vandalism, a malpractice forming a long-standing problem in systematics (*Kaiser et al., 2013*; *Wood et al., 2020*; *Wüster et al., 2021*), and thus further enhance taxonomic stability.

### Species-level diversity in Pareinae

Based on our updated phylogeny of Pareinae, we report on previously unrecognized diversity within the subfamily, and also confirm several taxonomic conclusions made in earlier studies. We document the high degree of uncorrected pairwise sequence divergence

between the two samples of *Aplopeltura boa* from Peninsular Malaysia and Sabah in Malaysian Borneo: with *p*-distances of 13.0% in cyt *b* gene and 16.7% in *ND4* gene, the divergence between these populations is estimated as 12.3 mya (Fig. 2; Table S9). Further integrative taxonomic studies are needed to clarify the taxonomic status of Malayan and Bornean populations of *A. boa* that might lead to recognition of several species within the genus *Aplopeltura*.

In the present study we re-define species groups within the genus *Pareas* recognizing two species groups within the clade A (A1: *P. carinatus* group; and A2: *P. nuchalis* group), and four species groups within the clade B (B1: *P. hamptoni* group; B2: *P. chinensis* group; B3: *P. margaritophorus* group; and B4: *P. monticola* group) (Fig. 3). Within the clade B of *Pareas* our results are largely concordant with a number of earlier studies. In *P. montiocola* group our analysis further confirms the species status of the recently described *P. victorianus* (*Vogel et al., 2021*); the divergence between this species and its sister taxon *P. monticola* is estimated as 14.1 mya (Fig. 2). In *P. margaritophorus* group our results fully agree with the data of *Vogel et al. (2020)* in recognizing *P. andersonii*, *P. modestus*, and *P. macularius* as species distinct from *P. margaritophorus*. Moreover we report on a deep divergence between the two samples of *P. macularius* from Myanmar and Laos with *p*-distances of 11.5% in cyt *b* gene and 10.4% in *ND4* gene (Tables S7 and S8); the divergence between these populations is estimated as ca. 8.2 mya (Fig. 2), what might be an indicative of an incomplete taxonomy of the group. As the type locality of *P. macularius* is located in Mon State, Myanmar, this raises the question of the taxonomic status of the populations from Laos (this work is currently in progress and will be presented elsewhere). Within *P. hamptoni* species group we confirm the results of *Ding et al. (2020)* and *Yang et al. (2021)*, suggesting that *P. kaduri* and *P. nigriceps* are sister taxa, while the phylogenetic position of *P. vindumi* remains unresolved. Genetic divergence among the three members of the Taiwan – Ryukyus clade of this group (*P. atayal*, *P. komaii*, and *P. iwasakii*) is comparatively low (6.9% < *p* < 9.0% in cyt *b* gene; see Table S7), and the basal radiation of this clade is estimated to happen only ca. 5.7 mya (Fig. 2). However a number of recent integrative studies, combining molecular, morphological, behavioral, and ecological data provide strong evidence that these taxa represent distinct species (*You, Poyarkov & Lin, 2015*; *Chang et al., 2021*). Our data support the conclusions of *Liu & Rao (2021)* which state that *P. mengziensis*, recently described by *Wang et al. (2020)* is conspecific to *P. niger*, a taxon which has been for a long time placed into a synonymy of *P. hamptoni*. Genetic divergence between *P. niger* and *P. mengziensis* is minimal (0.3% in cyt *b* gene; see Table S7), and given the morphological data reported by *Liu & Rao (2021)* there is little doubt that the latter taxon represents a junior synonym of the former. We further confirm the earlier results of *Ding et al. (2020)* in assigning the majority of populations of *P. hamptoni* complex from Vietnam, including the specimen identified as "*P. tonkinensis*", to *P. formosensis* (Fig. 3). Our results also agree with that of *Ding et al. (2020)* in recognizing *P. geminatus* as a species distinct from *P. hamptoni*, but its relationships with the recently described *P. xuelinensis* (*Liu & Rao, 2021*; our data) are less clear. In our analysis, *P. geminatus* consists of two major lineages (*P. geminatus* 1 from northern Thailand and *P. geminatus* 2 from southern

Yunnan of China and northern Laos), and is paraphyletic with respect to *P. xuelinensis*, which is grouped with *P. geminatus* 1 with strong support (Fig. 3). Two taxonomic solutions are possible to keep the monophyly of the recognized taxa: (1) to split *P. geminatus sensu lato* and assign the Thai population to *P. xuelinensis*; or (2) to consider *P. xuelinensis* a junior synonym of *P. geminatus*. Genetic divergence among *P. geminatus* 1 + *P. xuelinensis* and *P. geminatus* 2 is low (4.1% in cyt *b* gene; see Table S7), the divergence between these clades is estimated as only ca. 2.0 mya (Fig. 2), while the morphological characters distinguishing *P. xuelinensis* from *P. geminatus* are vague (see Table S13). For the time being, we refrain from a taxonomic decision on *P. geminatus* – *P. xuelinensis* clade pending further integrative studies to address this problem, which should include additional materials from China and northern Indochina.

Our study reports on a previously unrecognized diversity within the clade A of the genus *Pareas*: altogether, we reveal nine well-supported and highly-divergent clades within this group, five of which were previously unknown. Phylogenetic relationships among these lineages are generally well resolved (Fig. 3) and genetic divergence between them varies from $p = 4.8\%$ to 22.1% in cyt *b* gene, and from $p = 5.2\%$ to 20.1% in *ND4* gene (Tables S7 and S8). Recently *Wang et al. (2020)* revised the *P. carinatus* complex and described a new species from southern Yunnan of China. In their analyses *Wang et al. (2020)* only included samples from Peninsular Malaysia (*P. carinatus sensu stricto*) and from Yunnan, and based on genetic divergence and concordant morphological differences between these two populations, concluded that the Yunnan population should be regarded as a new distinct species – *P. menglaensis*. However, this taxonomic decision had several flaws:

1. *Wang et al. (2020)* only included in their analyses two populations of *P. carinatus* complex (from Yunnan and Peninsular Malaysia), but omitted any samples or sequences of *P. carinatus* complex from the major part of its range in Indochina and Sundaland, including the sequences available in GenBank.
2. *Wang et al. (2020)* overlooked two available species-level names presently coined as junior synonyms of *P. carinatus*: *Pareas berdmorei* Theobald, 1868, and *Amblycephalus carinatus unicolor* Bourret, 1934 (see Table 1).
3. Finally, in their revision *Wang et al. (2020)* did not examine any type specimens of the *P. carinatus* species complex.

Our updated tree indicates that *P. carinatus sensu lato* is paraphyletic with respect to both *P. menglaensis* and *P. nuchalis* + *P. temporalis*, and that the taxonomy of the complex needs to be reconsidered. The preponderance of data suggests that the pronounced phylogeographic structure within *P. carinatus* – *P. nuchalis* groups that bear deep genetic divergences, generally wide morphospatial separation among the sampled populations, and statistically different character state means is indicative of a species complex and as such we consider each population to be recognized as a distinct taxon, which we formally describe below:

1. The lineage of *P*. cf. *carinatus* from mountains of central Vietnam (lineage 7) and *P. temporalis* from southern Vietnam (lineage 8) form a well-supported clade with sister

relationships to *P. nuchalis* (though with moderate node support) (Fig. 3). Genetic distance between these lineages is high (12.7% in cyt *b* gene, 9.6% in *ND4* gene; see Tables S7 and S8); the divergence between them is estimated as ca. 9.3 mya (Fig. 2); they differ in a number of taxonomically significant characters from each other and other congeners (see below; summarized in Table 2), and are widely separated in PCA analysis (Fig. 4). We recognize lineage 7 as a separate new species and together with *P. nuchalis* and *P. temporalis* assign them to the *P. nuchalis* species group (subclade A2, Fig. 3), while all other members of clade A we assign to the *P. carinatus* species group (subclade A1, Fig. 3).

2. The population from Peninsular Malaysia (lineage 5), which morphologically and biogeographically corresponds to *P. carinatus sensu stricto*, forms a clade with lineage 6 from Tenasserim Mountains in western Thailand and adjacent Myanmar. Lineages 5 and 6 are separated by the Isthmus of Kra and likely diverged ca. 5.0 mya (Fig. 2); they are characterized by a moderate level of divergence in mtDNA gene sequences ($p$ = 7.7% in cyt *b* gene, 5.7% in *ND4* gene; see Tables S7 and S8), well separated in PCA analysis (Fig. 4), and are diagnosed by stable differences in a number of morphological characters (see below). We propose to recognize the Tenasserim lineage 6 as a new subspecies within *P. carinatus*.

3. All samples of *P. carinatus* complex from the mainland Southeast Asia form a clade sister to the clade inhabiting Thai-Malay Peninsula and Sundaland. Within the mainland clade a single specimen from Phu Yen Province of Vietnam (lineage 4) is highly divergent ($p$ = 11.9–12.7% cyt *b* gene, 13.8–14.2% in *ND4* gene; see Tables S7 and S8), forming a sister lineage with respect to all remaining populations (Fig. 3). Though no geographic barrier is known to separate the Phu Yen population from other mainland lineages of *P. carinatus* complex, the divergence between them is estimated as ca. 11.3 mya (Fig. 2). Moreover, the Phu Yen specimen is different from all other congeners in a number of diagnostic morphological features (see below) and is widely separated from them in the PCA morphospace (Fig. 4). Below we describe the Phu Yen lineage 4 as a new species.

4. Finally, all mainland populations of *P. carinatus* complex except the Phu Yen lineage 4 form a clade with three well-supported subclades (see Fig. 3): (1) the basal subclade (lineage 3) encompasses populations from northern Tenasserim to Thailand and Yunnan, and includes the topotypic population of *P. berdmorei* from Mon State, Myanmar (Fig. 1, loc. 16) and *P. menglaensis* (Fig. 1, loc. 22); (2) populations from southern Vietnam (lineage 1), including the type of *Amblycephalus carinatus unicolor* in Kampong Speu, Cambodia (Fig. 1, loc. 29); and (3) populations from Northern Annamites in Vietnam and Laos (lineage 2). These three lineages are separated from each other by moderate genetic distances ($p$ = 4.8%–7.4% in cyt *b* gene, 5.2–6.8% in *ND4* gene; see Tables S7 and S8) with estimated divergence times of 5.9–4.0 mya. They are only partially separated in PCA analysis, with a wide overlap in the morphospace for lineages 1 and 3, and moderate separation of lineage 2 (Fig. 4), but are readily distinguished from each other in a number of chromatic and certain morphological

differences (see below). We thus suggest that *P. menglaensis Wang et al., 2020* represents a subjective junior synonym of *P. berdmorei* Theobald, 1868, and propose to recognize *P. berdmorei* as a full species with three subspecies: *berdmorei* (for lineage 3), *unicolor* (for lineage 1), and a new subspecies for lineage 2 described below.

In the updated taxonomy for *P. carinatus – P. nuchalis* complex we propose to recognize two new species and two new subspecies (see above). Though there has been a certain skepticism regarding the usage of subspecies in herpetological taxonomy in the past (*e.g.*, *Wilson & Brown, 1953*; *Frost & Hillis, 1990*; *Frost, Kluge & Hillis, 1992*), recently the category of subspecies is getting more popular in scope of wide application of phylogenomic data allowing to reveal new cases of mito-nuclear discordance due to ongoing or ancient hybridization (*e.g.*, *Kindler & Fritz, 2018*; *De Queiroz, 2020*; *Hillis, 2021*; *Marshall et al., 2021*). *Marshall et al. (2021)* define the subspecies as a geographically circumscribed lineage that may has been temporarily isolated in the past, but which has since merged over broad zones of intergradation that show no evidence of reproductive isolation. We tend to tentatively recognize the lineages 1–3 within *P. berdmorei*, and the lineages 5–6 within *P. carinatus* as subspecies but not as full species due to the following reasons: (1) genetic distances among the lineages 1–3 within *P. berdmorei*, and the lineages 5–6 within *P. carinatus* are notably lower than between the 'good' species within the *P. carinatus* group (see above); (2) though lineages 5–6 within *P. carinatus* are well separated in the PCA analysis, lineages 1–3 within *P. berdmorei* are poorly separated in the PCA analysis (see Fig. 4) and are differentiated from each other primarily by chromatic traits; (3) the estimated time of divergence between the lineages 1–3 (ca. 5.9–4.0 mya), and the lineages 5–6 (ca. 5.0 mya) within *P. carinatus* is notably younger than the age of divergence of 'good' species within the *P. carinatus* group (ca. 17.2–9.3 mya) and is comparable with the age of basal divergence of other wide-ranged species of *Pareas*, *e.g. P. margaritophorus* (ca. 5.2 mya), *P. formosensis* (ca. 3.6 mya), and *P. monticola* (ca. 3.4 mya) (see Fig. 2); (4) lineages 2 and 6 are represented in our analyses by a limited sampling of two specimens for each lineage; this material may be not sufficient to fully assess the variation of diagnostic morphological characters; (5) last but not least, *Amblycephalus carinatus unicolor* was traditionally recognized as a subspecies of *P. carinatus* (*Nguyen, Ho & Nguyen, 2009*), thus keeping this taxon along with its sister lineages 2 and 3 in the rank of subspecies would support the taxonomic stability. Further studies including examination of additional materials and localities are needed to test whether the lineages within *P. carinatus* and *P. berdmorei* have zones of intergradation or are reproductively isolated from each other.

**Taxonomic accounts**

**Family Pareidae Romer, 1956**
**Subfamily Pareinae Romer, 1956**

**Type genus:** *Pareas Wagler, 1830*, by original designation.

## Revised key to the genera and subgenera of the subfamily Pareinae

1a. Dorsal scales in 13 rows; subcaudals undivided.................... genus *Aplopeltura*

1b. Dorsal scales in 15 rows; all subcaudals divided .................................... 2

2a. Anterior single inframaxillary shield absent; vertebrals scales weakly or not enlarged; preocular and subocular scales present; supralabials usually not in contact with the eye ............................................................................. genus *Pareas*

  2aa. Frontal hexagonal with its lateral sides parallel to the body axis; anterior pair of chin shields generally broader than long or slightly longer; two or three distinct narrow suboculars........................................................... subgenus *Pareas*

  2ab. Frontal subhexagonal with the lateral sides converging posteriorly; anterior pair of chin shields much longer than broad; one thin elongated subocular ......... subgenus *Eberhardtia* **stat. nov.**

2b. Anterior single inframaxillary shield present; vertebral scales strongly enlarged; preocular and subocular scales absent; supralabials in contact with the eye........ genus *Asthenodipsas*

  2ba . Two pairs of chin shields; the third pair of infralabials in contact with each other ...................................................... subgenus *Asthenodipsas*

  2bb. Three pairs of chin shields; the first pair of infralabials in contact with each other .................... ..........................subgenus *Spondylodipsas* **subgen. nov.**

**Phylogenetic definition:** Pareinae is a maximum crown-clade name referring to the clade originating with the most recent common ancestor of *Asthenodipsas malaccana* and *Pareas carinatus*, and includes all extant species that share a more recent common ancestor with these taxa than with *Xylophis captaini*.

**Updated diagnosis:** Body strongly laterally compressed with long tail; a short skull, head strongly distinct from neck; large eyes with vertical pupil; vertebrals sharp, weakly enlarged or not enlarged; 13–15 rows dorsal scale rows (DSR) throughout the body; no mental groove; ventrals preceded by a strongly enlarged preventral larger than the first ventral scale.

**Natural history notes:** Pareinae are nocturnal, generally arboreal, oviparous snakes, mainly inhabiting of moist tropical and subtropical forests, all members are specialized feeders on snails and slugs.

**Distribution:** Widely distributed through the Oriental zoogeographic region from Eastern Himalaya and Northeastern India, central, southern and eastern China including the islands of Hainan and Taiwan, the Yaeyama group of the Ryukyus across the Indochina and the Thai-Malay Peninsula to the Greater Sunda Islands.

**Content:** includes all members of the three genera: *Pareas Wagler, 1830*, *Aplopeltura Duméril, 1853*, and *Asthenodipsas Peters, 1864* (see below).

**English name:** Slug-eating snakes or Snail-eating snakes.

**Remark:** Diagnostic morphological features for the genera and subgenera of the subfamily Pareinae recognized herein are summarized in Table S14.

**Genus *Asthenodipsas* Peters, 1864**

**Type species:** *Asthenodipsas malaccana Peters, 1864*: 273–274, pl., figs. 3, 3a–3d, by monotypy.

**Synonyms:** *Internatus* Yang & Rao in *Rao & Yang, 1992* (type species – *Amblycephalus laevis* Boie, 1827).

**Phylogenetic definition:** *Asthenodipsas sensu lato* is a maximum crown-clade name referring to the clade originating with the most recent common ancestor of *Asthenodipsas malaccana* and *A. vertebralis*, and includes all extant species that share a more recent common ancestor with these taxa than with any of the type species of other Pareinae genera recognized herein.

**Updated diagnosis:** Dorsal scales smooth, in 15 rows throughout the body; vertebrals enlarged, hexagonal; sharp vertebral keel developed; head distinct from neck, snout blunt; one or two loreals; preocular and subocular scales absent; supraoculars may be fused to the postoculars; nasal undivided; prefrontal, loreal and at least one supralabial in contact with the eye; supraoculars may be fused to the postocular; frontal subhexagonal with the lateral sides converging posteriorly; two anterior temporals; the anterior single inframaxillary shield present (Figs. 5C–5F); inframaxillaries wider than long in two or three pairs; the first or third pair of inframaxillaries in contact with each other (Figs. 5C–5F); cloacal plate entire; subcaudals divided (*Peters, 1864*; *Grossmann & Tillack, 2003*; *Quah et al., 2019*, *2020*; our data; see Table S14).

**Distribution:** Sundaic region, including the southern Peninsular Thailand, West Malaysia, and the Greater Sunda Islands (Sumatra, Java and Borneo).

**Content:** Nine species, including *A. borneensis* Quah, Grismer, Lim, Anuar & Chan; *A. ingeri* Quah, Lim & Grismer; *A. jamilinaisi* Quah, Grismer, Lim, Anuar & Imbun; *A. laevis* (Boie); *A. malaccana* Peters; *A. lasgalenensis* Loredo, Wood, Quah, Anuar, Greer, Ahmad & Grismer; *A. tropidonota* (Lidth de Jeude); *A. stuebingi* Quah, Grismer, Lim, Anuar & Imbun; and *A. vertebralis* (Boulenger).

**Etymology:** The genus name is derived from the Greek word "*asthenos*" (ασθενώς) for "weak", "lacking strength", and the generic name "*Dipsas*", which is believed to come from the name of a snake in Greek mythology "*Dipsas*" (Διψας), the bite of which was believed to cause intense thirst (or "*dipsa*" [διψά] in Ancient Greek meaning "thirst"). The word "*Dipsas*" is feminine in gender, therefore *Asthenodipsas* must be treated as feminine, and the names of the included species have to be adjusted to feminine gender (*e.g.*, *malaccana*). The species name "*tropidonotus*" (meaning "keel-backed" in Greek)

represents a latinized adjective and therefore its gender has to be adjusted to the feminine gender of *Asthenodipsas* as "*tropidonota*".

**Recommended English name:** Sundaic slug-eating snakes.

**Material examined ($n$ = 38):** For the detailed information (specimen IDs, locality, sex, and the main morphological characteristics) of *Asthenodipsas borneensis* ($n$ = 1), *A. laevis* ($n$ = 15), *A. lasgalensis* ($n$ = 5), *A. malaccana* ($n$ = 10), *A. stuebingi* ($n$ = 1), *A. tropidonota* ($n$ = 5), and *A. vertebralis* ($n$ = 1); see Table S15 and Appendix S2.

**Remark:** We recognize the following two subgenera within the genus *Asthenodipsas* for the *A. laevis* and *A. vertebralis* species groups based on stable morphological differences between their members concordant with the ancient phylogenetic divergence between these groups (see above). We propose the subgenus level for the taxa recognized below based on (1) strong support for the monophyly of the genus *Asthenodipsas sensu lato* (see above); (2) biogeographic similarity between these two groups, which are both distributed in the Sundaland region.

### Subgenus *Asthenodipsas Peters, 1864*

**Type species:** *Asthenodipsas malaccana Peters, 1864*: 273–274, pl., figs. 3, 3a–3d, by monotypy.

**Synonyms:** *Internatus* Yang & Rao in *Rao & Yang, 1992* (type species – *Amblycephalus laevis* Boie, 1827).

**Phylogenetic definition:** *Asthenodipsas sensu stricto* is a maximum crown-clade name referring to the clade originating with the most recent common ancestor of *Asthenodipsas malaccana* and *A. laevis*, and includes all extant species that share a more recent common ancestor with these taxa than with *A. vertebralis*.

**Diagnosis:** The subgenus *Asthenodipsas* differs from the subgenus *Spondylodipsas* **subgen. nov.** (described below) by the following morphological characters: two pairs of chin shields (*vs.* three pairs); 5–7 supralabials; 4–7 infralabials; the third pair of infralabials in contact with each other behind the single anterior inframaxillary shield (*vs.* the first pair) (Figs. 5C–5D; see Table S14 for details).

**Distribution:** Southern Peninsular Thailand, West Malaysia, Borneo, Sumatra, Java, and Bangka islands, Mentawai, and Natuna Archipelagos.

**Content:** Six species, including *A. borneensis* Quah, Grismer, Lim, Anuar & Chan; *A. ingeri* Quah, Lim & Grismer; *A. jamilinaisi* Quah, Grismer, Lim, Anuar & Imbun; *A. laevis* (Boie); *A. malaccana* Peters; and *A. stuebingi* Quah, Grismer, Lim, Anuar & Imbun.

**Recommended English name and Etymology:** as for the genus *Asthenodipsas*.

**Subgenus *Spondylodipsas* Poyarkov, Nguyen TV & Vogel subgen. nov.**
[urn:lsid:zoobank.org:act:3FE7563C-2BFE-4BA4-A084-1A66E3D9B706]

**Type species:** *Amblycephalus vertebralis Boulenger, 1900*: 307–308, by original designation.

**Phylogenetic definition:** *Spondylodipsas* **subgen. nov.** is a maximum crown-clade name referring to the clade originating with the most recent common ancestor of *Asthenodipsas vertebralis* and *A. lasgalenensis*, and includes all extant species that share a more recent common ancestor with these taxa than with *A. malaccana*.

**Diagnosis:** The subgenus *Spondylodipsas* **subgen. nov.** differs from *Asthenodipsas sensu stricto* by the following characteristics: three pairs of chin shields (*vs.* two pairs); 6–8 supralabials; 6–8 infralabials; the first pair of infralabials in contact with each other behind the mental (*vs.* the third pair) (Figs. 5E and 5F; see Table S14 for details).

**Distribution:** Mountain areas of Sumatra and Peninsular Malaysia, and Pulau Tioman.

**Content:** Three species, including *A. lasgalenensis* Loredo, Wood, Quah, Anuar, Greer, Ahmad & Grismer, *A. tropidonota* (Lidth de Jeude), and *A. vertebralis* (Boulenger).

**Etymology:** The genus name is a Latinized noun in feminine gender and is derived from the Greek word "*spondylon*" (σπονδύλων) for "vertebra", and the generic name "*Dipsas*" (for etymology of this name see above). The name is given in reference to the well-developed vertebral keel in the members of the subgenus.

**Recommended English name:** Ridged slug-eating snakes.

**Genus *Aplopeltura Duméril, 1853***

**Type species:** *Amblycephalus boa Boie, 1828*: 1035, by monotypy.

**Phylogenetic definition:** *Aplopeltura* is a maximum crown-clade name referring to *Aplopeltura boa* originating as the sister lineage to *Pareas sensu lato*.

**Updated diagnosis:** Dorsal scales smooth, in 13 rows throughout the body; vertebral keel weakly developed; two or three loreals; preocular and subocular scales present; supralabials not in contact with the eye; three anterior temporals; the anterior single inframaxillary shield absent (Fig. 5G); generally four (rarely three) pairs of chin shields, anterior pair of chin shields broader than long; at least the first and second pairs of chin shields in contact; subcaudals undivided (*Duméril, 1853*; *Taylor, 1965*; our data; see Table S14 for details).

**Distribution:** Sundaic region, including: southern Peninsular Thailand, Peninsular Malaysia, Borneo, Sumatra, Java, Nias, Bangka, and Natuna Islands, and the Philippines (reliably recorded from the Balabac, Basilan, Mindanao, Palawan, and Luzon islands). The published record from southern Peninsular Myanmar by *Dowling & Jenner (1988)* requires further verification.

**Content:** A monotypic genus including the single species, *A. boa* Boie.

**Etymology:** The genus name is likely derived from the Greek words "*aplos*" (απλώς) for "simple", and "*pelte*" (πέλτη), for "scale", originally a name of a type of a small shield used in Ancient Greece.

**Recommended English name:** Blunt-headed slug-eating snakes.

**Material examined (*n* = 2):** For detailed information (specimen IDs, locality, sex, and main morphological characteristics) of *Aplopeltura boa* (*n* = 2) see Table S15 and Appendix S2.

**Remark:** Our study reports on the significant genetic divergence between the samples of *A. boa* from Borneo (Sabah) and Peninsular Malaysia, corresponding to the species level in Pareinae (see above). Further integrative taxonomic studies are required to clarify the status of the lineages within this species.

### Genus *Pareas* Wagler, 1830

**Type species:** *Dipsas carinata* Wagler, 1830, by monotypy (see account for *Pareas carinatus* for details).

**Phylogenetic definition:** *Pareas sensu lato* is a maximum crown-clade name referring to the clade originating with the most recent common ancestor of *Pareas carinatus* and *P. formosensis*, and includes all extant species that share a more recent common ancestor with these taxa than with any of the type species of other Pareinae genera recognized herein.

**Updated diagnosis:** Dorsal scales smooth or keeled, in 15 rows throughout the body; vertebrals slightly larger than other dorsal scales or not enlarged; one (rarely two) loreals; preocular and subocular scales present; supralabials generally not contacting the eye (except for *P. monticola* and *P. stanleyi*); three anterior temporals; the anterior single inframaxillary shield absent (Figs. 5H–5G); three pairs of inframaxillaries, all in contact with each other; subcaudals divided (Wagler, 1830; Smith, 1943; Taylor, 1965; Vogel et al., 2020; our data; see Table S14 for details).

**Distribution:** Widely distributed throughout the Oriental zoogeographic region from Northeastern India and Himalaya to Eastern China and the Greater Sunda Islands.

**Content:** 26 species, including *P. andersonii* Boulenger; *P. atayal* You, Poyarkov & Lin; *P. berdmorei* Theobald; *P. boulengeri* (Angel); *P. carinatus* Wagler; *P. chinensis* (Barbour); *P. formosensis* (van Denburgh); *P. geminatus* Ding, Chen, Suwannapoom, Nguyen, Poyarkov & Vogel; *P. hamptoni* (Boulenger); *P. iwasakii* (Maki); *P. kaduri* Bhosale, Phansalkar, Sawant, Gowande, Patel & Mirza; *P. komaii* (Maki); *P. macularius* Theobald; *P. margaritophorus* (Jan); *P. modestus* Theobald; *P. monticola* (Cantor); *P. niger* (Pope); *P. nigriceps* Guo & Deng; *P. nuchalis* (Boulenger); *P. stanleyi* (Boulenger); *P. temporalis* Le, Tran, Hoang & Stuart; *P. victorianus* Vogel, Nguyen & Poyarkov; *P. vindumi* Vogel; *P. xuelinensis* Liu & Rao; and the two new species described herein below: *P. abros*

Poyarkov, Nguyen & Vogel **sp. nov.**; and *P. kuznetsovorum* Poyarkov, Yushchenko & Nguyen **sp. nov.**

**Etymology:** The genus name is a latinized noun in masculine gender derived from the Greek noun "*pareias*" (παρείας), a name of a mythological snake dedicated to Asclepius, and which was believed to be non-venomous and create a furrow anytime it moves.

**Recommended English name:** Oriental slug-eating snakes.

**Material examined (*n* = 265):** Detailed information (specimen IDs, locality, sex, and main morphological characteristics) for *P. abros* (*n* = 3), *P. berdmorei* (*n* = 21), *P. carinatus* (*n* = 26), *P. kuznetsovorum* (*n* = 1), *P. nuchalis* (*n* = 9), and *P. temporalis* (*n* = 6) is presented in Table 2 and Appendix S2; for *P. vindumi* (*n* = 1) see *Vogel (2015)*; for *P. andersonii* (*n* = 13), *P. macularius* (*n* = 15), *P. margaritophorus* (*n* = 51), and *P. modestus* (*n* = 8) see *Vogel et al. (2020)*; for *P. geminatus* (*n* = 9) and *P. hamptoni* (*n* = 5) see *Ding et al. (2020)*; for *P. formosensis* (*n* = 29), *P. kaduri* (*n* = 1), *P. monticola* (*n* = 24), and *P. victorianus* (*n* = 1) see *Vogel et al. (2021)*; for the abovementioned species and *P. atayal* (*n* = 6), *P. boulengeri* (*n* = 10), *P. chinensis* (*n* = 7), *P. komaii* (*n* = 9), *P. stanleyi* (*n* = 4), *P. xuelinensis* (*n* = 3) and *P.* cf. *yunnanensis* (*n* = 3) the information is summarized in Table S16 and Appendix S2.

**Remark:** The taxonomic status of *Amblycephalus yunnanensis* Vogt (1922), currently considered a junior synonym of *Pareas chinensis*, is unclear due to the high morphological similarity within the group and the geographic proximity of the type localities of two taxa (both described from Yunnan Province in China). *Ding et al. (2020)* discussed this issue and suggested that the integrative taxonomic analysis including detailed comparisons of the type specimens is required to clarify the relations of these taxa (see Table S13). Several recent studies on phylogenetic relationships within *Pareas* have revealed a deep divergence within the group, suggesting that its taxonomy still may be incomplete (*Guo et al., 2011*; *You, Poyarkov & Lin, 2015*; *Wang et al., 2020*; *Bhosale et al., 2020*; *Ding et al., 2020*; *Vogel et al., 2020*, *2021*).

### Subgenus *Pareas* *Wagler, 1830*

**Type species:** *Dipsas carinata* *Wagler, 1830*, by monotypy (see the account for *Pareas carinatus* for details).

**Phylogenetic definition:** *Pareas sensu stricto* is a maximum crown-clade name referring to the clade originating with the most recent common ancestor of *Pareas carinatus* and *Pareas nuchalis*, and includes all extant species that share a more recent common ancestor with these taxa than with *Pareas formosensis*.

**Diagnosis:** The members of the subgenus *Pareas* differ from the members of the subgenus *Eberhardtia* (designated below) by the following morphological characteristics: frontal hexagonal with the lateral sides parallel to each other (Fig. 5B); anterior pair of chin shields broader than long (Fig. 5H); two or three distinct narrow suboculars; and the ravine-like
ultrastructure of dorsal scales (*Wagler, 1830*; *Smith, 1943*; *Taylor, 1965*; *Vogel et al., 2020*; *He, 2009*; *Guo, Wang & Rao, 2020*; our data; see Table S14 for details).

**Distribution**: Distributed in the south-eastern part of the Oriental zoogeographic region from the southernmost China throughout the Indochina Peninsula to Peninsular Malaysia, Sumatra, Java, and Borneo (see Fig. 1).

**Content:** Six species, including *P. berdmorei* Theobald (with three subspecies: *P. b. berdmorei* **stat. nov.**, *P. b. unicolor* **comb. nov.**, and *P. b. truongsonicus* **ssp. nov.**); *P. carinatus* Wagler (with two subspecies: *P. c. carinatus*, and *P. c. tenasserimicus* **ssp. nov.**); *P. nuchalis* (Boulenger); *P. temporalis* Le, Tran, Hoang & Stuart; and the two new species described herein below: *P. abros* Poyarkov, Nguyen & Vogel **sp. nov.**; and *P. kuznetsovorum* Poyarkov, Yushchenko & Nguyen **sp. nov.**

**Recommended English name and Etymology:** as for the genus *Pareas*

## Key to the species of the subgenus *Pareas*

1a. Prefrontal contacting the eye...........................................*P. nuchalis*

1b. Prefrontal not contacting the eye ...................................................... 2

2a. Ratio TaL/TL≥0.25; large black blotch or a ring-shaped pattern on the nuchal area  3

2b. Ratio TaL/TL<0.25; large black blotch or a ring-shaped figure on the nuchal area absent................................................................................... 5

3a. All dorsal scales smooth, VEN<170; black blotch on the nuchal area not forming a ring-shaped pattern.........................................*P. kuznetsovorum* **sp. nov.**

3b. At least some dorsal scales strongly or slightly keeled, VEN>170; black blotch on the nuchal area forming a ring-shaped pattern........................................... 4

4a. All dorsal scale rows strongly keeled; VEN>185; no transverse dark bands on the body ...........................................................................*P. temporalis*

4b. 9–11 rows of dorsal scales keeled at midbody; VEN<185; faint transverse dark bands on the body .....................................................................*P. abros* **sp. nov.**

5a. Total length medium (up to 702 mm); dorsal scales generally keeled in 3–11 upper rows at midbody; upper postorbital stripes thick, contacting each other on the nape generally forming a X- or )(-shaped pattern; territories southwards from the Tenasserim Range in Thailand .................................................................*P. carinatus*

    5aa. VEN≤190; SC≤90; body with transverse dark bands; territories south of the Isthmus of Kra .................................................................*P. c. carinatus*

    5ab. VEN>190; SC>90; uniform light brown coloration of dorsum lacking transverse dark bands; Tenassenrim Range northwards from the Isthmus of Kra .................................................*P. c. tenasserimicus* **ssp. nov.**

5b. Total length large (up to 770 mm); dorsal scales keeled in 3–13 upper rows at midbody; upper postorbital stripes thin, generally forming a Y-shaped pattern on the nape or absent; mainland Indochina north from the Isthmus of Kra ......................*P. berdmorei*

*5ba.* VEN 166–186; SC 57–89; dorsal scales keeled in 5–13 rows at midbody; body with transverse dark bars; ventral surfaces immaculate; iris golden-bronze to orange; restricted to northern Vietnam, northern Laos, northern Thailand, eastern Myanmar (northern Tennasserin), and southern Yunnan ............. *P. b. berdmorei* **stat. nov.**

*5bb.* VEN 162–180; SC 57–75; dorsal scales keeled in 3–9 rows at midbody; body uniform orange to beige coloration lacking dark markings; ventral surfaces generally immaculate; iris bright orange-red; restricted to southern Vietnam and eastern Cambodia ............................................... *P. b. unicolor* **comb. nov.**

*5bc.* VEN 187; SC 66–73; dorsal scales keeled in 13 rows at midbody; dense brownish mottling and spots on dorsal, lateral, and ventral surfaces of the head and body; iris uniform off-white to golden; restricted to the northern Annanmites (Truong Son) Mountains in central Vietnam and Laos................ *P. b. truongsonicus* **ssp. nov.**

### *Pareas carinatus* species group

The monophyly of the *carinatus* group is well supported in both analyses (A1, Fig. 3); the members of this species group are widely distributed across the Indochina from southern Yunnan Province of China to the Thai-Malay Peninsula southwards to Sumatra, Java, Borneo and smaller adjacent islands (Fig. 1). Our phylogeny indicated that the group is composed of three species-level lineages, which were further supported by morphological analysis. The first lineage inhabits Sundaland and the Thai-Malay Peninsula and corresponds to *P. carinatus sensu stricto*. It includes two subgroups divided by the Isthmus of Kra, an important biogeographic boundary (*De Bruyn et al., 2005*). The populations from Sundaland and the Thai-Malay Peninsula south of Kra correspond to *P. c. carinatus* (lineage 5, Fig. 3), while the populations northwards of Kra inhabiting the southern part of Tenasserim Range in western Thailand and adjacent Myanmar, we assign to a new subspecies *P. c. tenasserimcus* **ssp. nov.** described below (lineage 6, Fig. 3). The second lineage which we identify as *P. berdmorei* inhabits the mainland Indochina and includes three subgroups which we treat as subspecies. The first subgroup (lineage 3, Fig. 3) is widely distributed from southern Myanmar, western and northern Thailand, to Yunnan Province, China, and corresponds to *P. b. berdmorei* **stat. nov.**). The second subgroup (lineage 1, Fig. 3) is restricted to southern Vietnam and southeastern Cambodia and represents the subspecies *P. b. unicolor* **comb. nov.** The third subgroup recorded from the northern part of the Annamite Range we assign to a new subspecies *P. b. truongsonicus* **ssp. nov.** described below (lineage 2, Fig. 3). Finally, the third species-level lineage of this group was recorded from the north-western foothills of the Langbian (Da Lat) Plateau in southern Vietnam (lineage 4, Fig. 3); we below describe it as a new species *P. kuznetsovorum* **sp. nov.** Morphological data on the *P. carinatus* group members is summarized in Table 2 and Tables S11–S13. All members of the *carinatus* group lack the characteristic large black ring-shaped blotch on the nape and lateral sides of the neck; in *P. kuznetsovorum* **sp. nov.** the black blotch on the nape is present, but it is not ring-shaped.

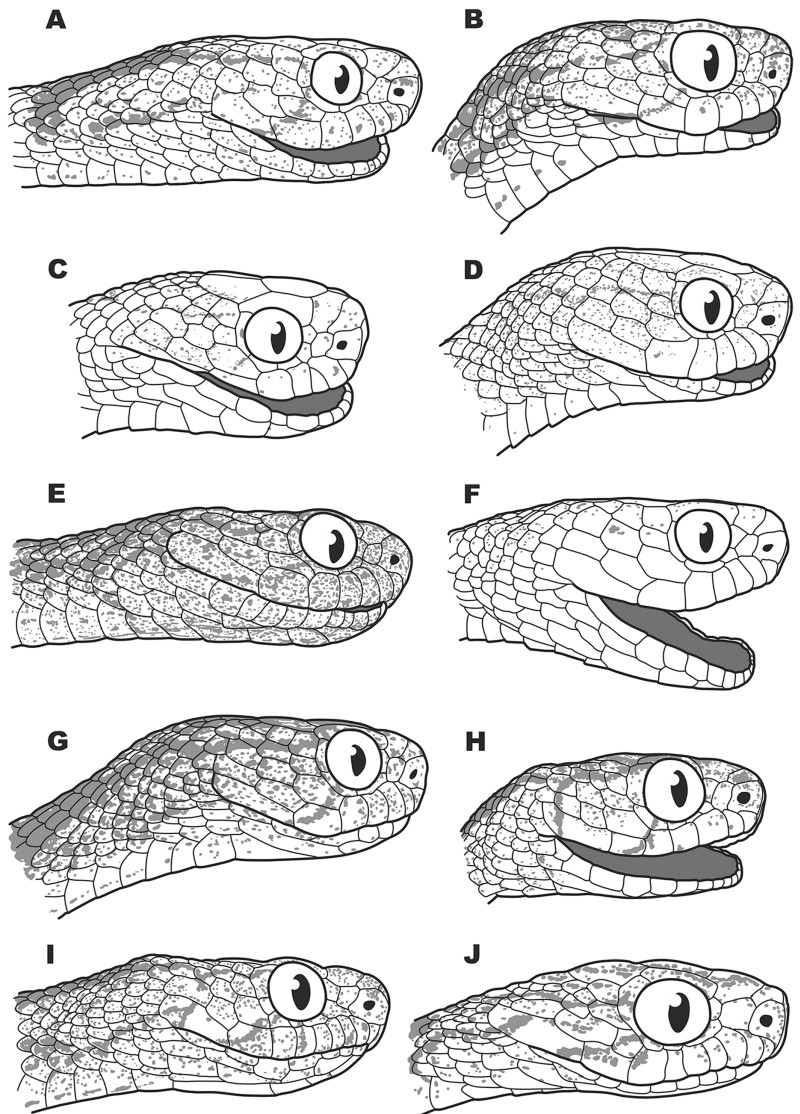

**Figure 6 Lateral views of head scalation of taxa of the subgenus *Pareas*.** (A) Lectotype of *Pareas carinatus* Wagler, 1830 (RMNH 954 C); (B) holotype of *P. carinatus tenasserimicus* **ssp. nov.** (ZMMU R-16800); (C) lectotype of *Pareas berdmorei* Theobald, 1868 (ZSI 8022); (D) holotype of *Pareas menglaensis* Wang, Che, Liu, Ki, Jin, Jiang, Shi & Guo, 2020 (YBU 14124); (E) holotype of *P. berdmorei truongsonicus* **ssp. nov.** (ZMMU R-16801); (F) holotype of *Amblycephalus carinatus unicolor* Bourret, 1934 (MNHN 1938.0149); (G) holotype of *Pareas kuznetsovorum* **sp. nov.** (ZMMU R-16802); (H) *Pareas nuchalis* (Boulenger, 1900) (FMNH 131635); (I) holotype of *Pareas abros* **sp. nov.** (ZMMU R-16393); (J) male of *Pareas temporalis* Le et al., 2021 (ZMMU R-13656). Not to scale. Drawings by L. B. Salamakha.

### *Pareas carinatus* Wagler, 1830
Figures 6A, 6B, 7–9; Table 2; Tables S11–S13.

*Dipsas carinata* Wagler, 1830: 181. – Type locality. Java (Indonesia), by inference and indication on the basis of the NHM and RMNH catalogues. None given in the original description. – Status. Species name, as published in the binomen *Dipsas carinata* Wagler,

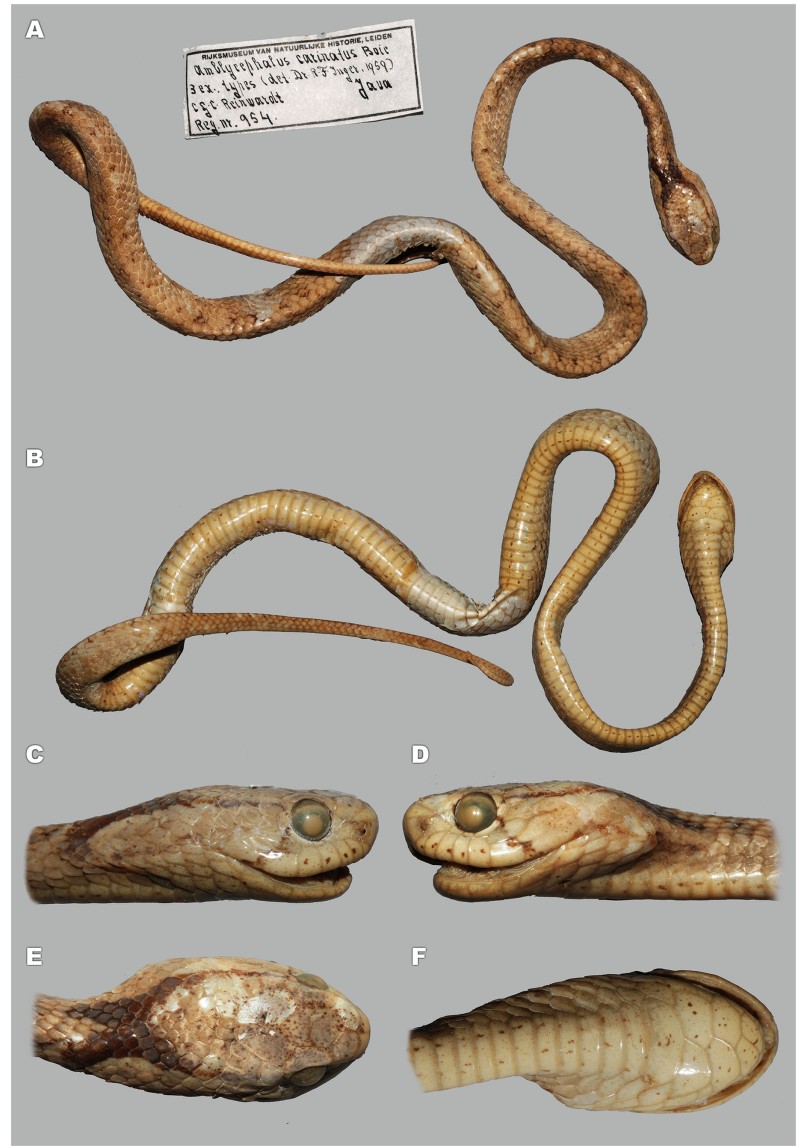

**Figure 7 Lectotype of *Pareas carinatus Wagler, 1830* in preservative (RMNH 954 C, adult male).**
(A) Dorsal view of body; (B) ventral view of body; (C–F) head in lateral right, lateral left, dorsal, and ventral aspects, respectively. Photos by G. Vogel.

*1830*, placed on the 'Official List of Specific Names in Zoology', as Name Nr 2452, after Opinion 963 of I.C.Z.N. (1971: 44).

**Chresonymy:**

*Amblycephalus carinatus* H. Boie in *Schlegel, 1826*: 1035 (*nomen nudum*).
*Amblycephalus carinatus Boie, 1828*: 251 (*nomen nudum*).
*Pareas carinata Wagler, 1830*: 181; *Duméril, Bibron & Duméril, 1854*: 439.
*Dipsas carinata* — *Schlegel, 1837*: 285; *Nguyen & Ho, 1996*: 61.
*Leptognathus carinatus* — *Jan, 1863*.

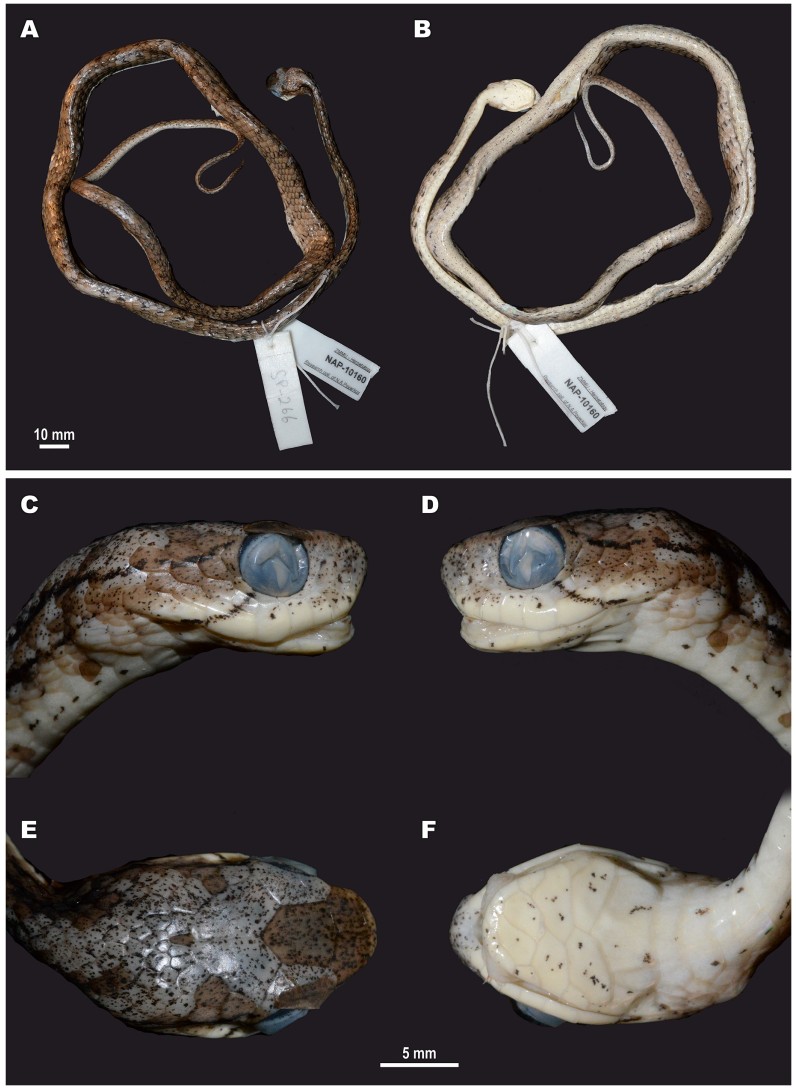

**Figure 8 Holotype of *P. carinatus tenasserimicus* ssp. nov. in preservative (ZMMU R-16800, adult male).** (A) Dorsal view of body; (B) ventral view of body; (C–F) head in lateral right, lateral left, dorsal, and ventral aspects, respectively. Photos by N. A. Poyarkov.

*Amblycephalus carinatus* — *De Rooij, 1917*: 277; *Smedley, 1931*: 53; *Kopstein, 1936*; *Deuve, 1961*: 30.

*Pareas carinatus* — *Cochran, 1930*; *Smith, 1943*: 121; *Manthey & Grossmann, 1997*: 376; *Cox et al., 1998*: 78; *Schmidt & Kunz, 2005*: 41; *Wallach, Williams & Boundy, 2014*: 535.

**Lectotype** (designated herein) (Fig. 6A; Fig. 7): RMNH 954C, adult male, collected by C. G. C. Reinwardt from Java Island, Indonesia. We designate the RMNH 954C as the lectotype, since it is the best preserved specimen of the type series fully agreeing with the original description, thus fulfilling the requirements of the Art. 74.7 of the Code (*ICZN, 1999*).

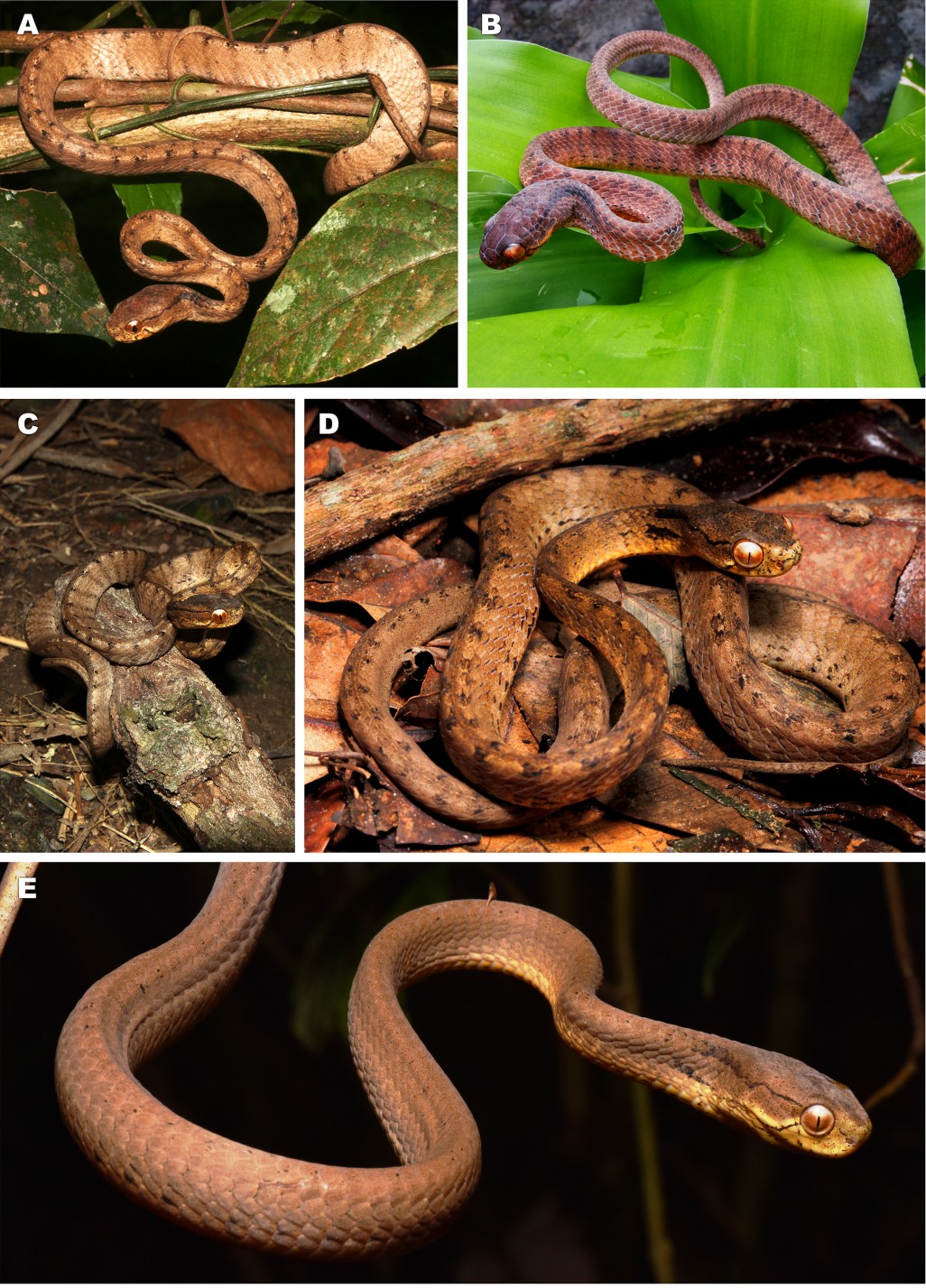

**Figure 9 Members of the *Pareas carinatus* complex in life.** (A) *P. c. carinatus* from Hala-Bala W.S., Narathiwat Province, Thailand; (B and C) *P. c. carinatus* from Gunung Leuser N.P., Bukit Lawang Province, Sumatra, Indonesia; (D) *P. c.* cf. *tenasserimicus* **ssp. nov.** from Kaeng Krachan N.P., Phetchaburi Province, Thailand; (E) *P. c. tenasserimicus* **ssp. nov.** from Suan Phueng, Ratchaburi Province, Thailand. Photos by L.A. Neimark (A), Guek Hock Ping aka Kurt Orion (B), H.X.N. Nguyen (C), P. Pawangkhanant (D), and M. Naiduangchan (E).

**Paralectotypes** (designated herein): Two specimens, RMNH 954A and RMNH 954B, both adult males, with the same collection data as the lectotype.

**Updated diagnosis:** *Pareas carinatus* differs from all other members of the genus *Pareas* by the following combination of morphological characters: body slender, maximal total length of 702 mm; frontal scale hexagonal with lateral sides parallel to the body axis; anterior pair of chin shields longer than broad; loreal and prefrontal not contacting the eye; 1–3 suboculars; usually one postocular; temporals 3 + 4 or 3 + 3; three median vertebral rows slightly enlarged; 7–9 infralabials; 15 dorsal scale rows, at midbody the five upper rows might be slightly keeled; 158–194 ventrals; 54–96 subcaudals, all divided; dorsum yellow-brown with dark vertebral blotches and dark mottling, transverse dark bands on the body present or absent; upper postorbital stripes continue to nape forming one or two longitudinal black spots; iris bronze laterally, beige dorsally (*Wagler, 1830*; our data).

**Material examined:** We directly examined 26 specimens of *P. carinatus sensu stricto* from Malaysia, Indonesia and Thailand (Table S11; Table 1).

**Description of the lectotype** (RMNH 954C) (Figs. 6A and 7): Adult male, body slender and notably flattened laterally; head comparatively large, narrowly elongated, clearly distinct from thin neck, snout blunt; eyes large, pupil vertical.

*Body size.* SVL 373 mm; TaL 101 mm; TL 474 mm; TaL/TL: 0.213

*Body scalation.* Dorsal scales in 15–15–15 rows, slightly keeled at midbody, and without apical pits; three vertebral scale rows slightly enlarged; outermost dorsal scale row not enlarged; ventrals 170 (+ 1 preventral), without lateral keels; subcaudals 67; cloacal plate single.

*Head scalation.* Rostral not visible from above; one nasal; two internasals, much wider than long, narrowing and slightly curving back laterally (in dorsal view), anteriorly in contact with rostral, laterally in contact with nasal and loreal, posteriorly in contact with prefrontal, not contacting preocular; two large irregular pentagonal prefrontals, much larger than internasals and with a slightly diagonal suture between them, not contacting the eye; frontal scale hexagonal with the lateral sides parallel to the body axis, longer than wide, smaller than parietals; one preocular; two suboculars; one postocular, not fused with subocular; one loreal in contact with prefrontal, not touching the eye; 7/7 supralabials, 3rd and 5th SL touching the subocular, none of them reaching the eye, 7th by far the largest, elongated; temporals 3 + 4; 9/9 infralabials, the anterior most in contact with the opposite along the midline, bordering mental, anterior 5 pairs of infralabials bordering anterior chin shields; 3 pairs of chin shields interlaced, no mental groove under chin and throat; anterior chin shields relatively large, generally wider than long, followed by two pairs of chin shields much wider than long.

*Coloration.* After over 200 years in preservative, the dorsal and ventral surfaces of the head, brownish with some dark-brown dusted spots (Fig. 7). Head with two lateral dark-brown postorbital stripes: the lower one is an interrupted dark line starting from the

posterior edge of the eye, going diagonally down onto the anterior part of the last supralabial; the upper postorbital stripe is a dark-brown line running from the postocular backwards to the dorsal scales on the neck, where it meets the similar line on the opposite side of the body forming a narrow X-shaped dark-brown marking on the nuchal area (Fig. 7A). Upper labials marked with some fine irregular brown speckling (Figs. 7C and 7D). Dorsal surface is nearly uniformly light brown with slightly visible dark cross bands; ventral surfaces yellowish with sparse brownish mottling forming the interrupted line along the midline, descending backwards. Coloration in life unknown.

**Comparisons:** *Pareas carinatus* differs from *P. berdmorei* (revalidated below) by the generally smaller body size (494.3 ± 73.3 mm *vs.* 554.9 ± 73.3 mm); by slightly lower number keeled dorsal scale rows (6.5 ± 2.9 *vs.* 8.9 ± 2.8); and by generally thicker upper postorbital stripe and more pronounced dark markings on the nape (*vs.* thinner postorbital stripe and less pronounced dark markings on the nape); *P. carinatus* differs from *P. nuchalis* by prefrontal not contacting the eye (*vs.* in contact); by the absence of the large ring-shaped black blotch on the nape (*vs.* present); by lower number of ventrals (171.35 ± 9.3 *vs.* 209.89 ± 5.3); lower number of subcaudals (73.3 ± 7.6 *vs.* 111.1 ± 6.1); and by having keeled dorsal scales (*vs.* dorsal scales totally smooth). Morphological comparisons between all species of the subgenus *Pareas* are detailed in Table 2. *Pareas carinatus* can be distinguished from other species of *Pareas* belonging to subgenus *Eberhardtia* **stat. nov.** by having two or three distinct narrow suboculars (*vs.* one thin and elongated) and by having a hexagonal frontal with its lateral sides parallel to the body axis (*vs.* subhexagonal) (see Tables S13 and S14).

**Distribution**: Based on molecular and morphological data, we suggest that this species is restricted to the Greater Sunda Islands (Java, Borneo and Sumatra), Peninsular Malaysia and Thailand, including the Tenasserim Mountains in western Thailand and south-eastern Myanmar, the northernmost known locality is Yaephyu, Tanintharyi Region, Myanmar (Fig. 1).

**Etymology:** The species name "*carinatus*" is a Latin adjective in nominative singular, masculine gender, derived from "*carina*" for a "keel of a ship", and is given in reference to the keeled dorsal scales in this species.

**Recommended English name:** Keeled slug-eating snake.

**Remark:** Based on the concordant results of morphological and molecular analyses, we recognize two subspecies within *P. carinatus*: the populations south of the Isthmus of Kra correspond to the nominative subspecies *P. c. carinatus*, while the populations from the Tenasserim Mountains northwards from Kra we describe below as *P. c. tenasserimcus* **ssp. nov.** Although the morphological variation among the sampled specimens of *P. c. carinatus* is significant (Fig. 4; Tables S11 and S12), in molecular analyses this subspecies is only represented by specimens from Peninsular Malaysia. Further phylogenetic analyses of *P. carinatus* populations from Java, Sumatra and Borneo are required and might reveal new presently unknown lineages within this species.

### *Pareas carinatus carinatus* *Wagler, 1830*

Figures 6A, 7, 9A–9C; Tables S11 and S12.

*Dipsas carinata* *Wagler, 1830*: 181.

**Chresonymy:**

*Amblycephalus carinatus carinatus* — (in part) *Mertens, 1930*.

*Pareas carinatus carinatus* — (in part) *Haas, 1950*; *Chan-ard et al., 1999*: 177; *Nguyen, Ho & Nguyen, 2009*: 374.

**Diagnosis:** *Pareas carinatus carinatus* differs from *Pareas carinatus tenasserimicus* **ssp. nov.** described below by the following combination of morphological characters: maximal total length of 608 mm; anterior pair of chin shields wider than long; one postocular; temporals generally 3 + 4 (rarely 3 + 3, 2 + 3); 15 dorsal scale rows slightly keeled in 3–11 scale rows at midbody; 158–190 ventrals; 54–84 subcaudals; dorsum light brown with distinct dark vertebral spots and generally 44–73 transverse dark brownish or blackish bands (Figs. 9A–9C); upper postorbital stripes generally contacting each other on the nuchal area forming a narrow X-shaped dark marking with curved branches; ventral scales yellowish with sparse brownish mottling forming the interrupted line along the midline.

**Variation:** Measurements and scalation features of the subspecies *P. c. carinatus* ($n = 25$) are presented in Table S11. There is a certain variation among the sexes observed in the body size and the number of ventral scales. Males are generally smaller (TL 337.0–571.0 mm, average 471.5 ± 59.5 mm, $n = 15$) than females (TL 446.0–608.0 mm, average 511.0 ± 56.8 mm, $n = 8$); males also have a generally lower number of ventrals than females (158–183, average 167.3 ± 6.2, $n = 15$ in males *vs.* 162–190 average 175.2 ± 8.8, $n = 10$ in females). In five specimens from from Java (ZMH R11546 and R11542), Sumatra (SMF 37825–37826) and Borneo (NMW 28131:1) keels on dorsal scales are hardly visible; it is unclear if this feature reflects the geographic variation, or it might arise from the poor preservation of the specimens. Other morphological features showed no significant variation among the specimens of the series examined. In our phylogenetic analysis, *P. c. carinatus* was represented only by specimens from Peninsular Malaysia; further integrative molecular and morphological studies are needed to assess the geographic variation among the populations of *P. c. carinatus* from Java, Sumatra, Borneo, and Peninsular Malaysia.

**Distribution:** Peninsular Thailand south of the Isthmus of Kra, Malaysia (Peninsular Malaysia, Sarawak and Sabah within Borneo), Brunei, and Indonesia (Kalimantan Province, Sumatra, Java, Lombok, and Bali Islands) (Fig. 1).

**Recommended English name and Etymology:** as for *Pareas carinatus*.

### *Pareas carinatus tenasserimicus* Poyarkov, Nguyen TV, Vogel, Pawangkhanant, Yushchenko & Suwannapoom ssp. nov.

[urn:lsid:zoobank.org:act:11F7F6BA-4733-41FB-8E2D-405DCA5743E5]

Figures 6B, 8, 9D and 9E; Tables S11 and S12.

**Chresonymy:**

*Pareas carinatus* — (in part) *Mulcahy et al., 2018*: 98.

**Holotype** (Figs. 6B and 8): ZMMU R-16800 (field number NAP-10160), adult male, collected by P. V. Yushchenko from mountain forest in Joot Chomwil area, Suan Phueng District, Ratchaburi Province, western Thailand (N 13.56346, E 99.19465, elevation 800 m asl.) on 16 July 2019.

**Diagnosis:** *Pareas carinatus tenasserimicus* **ssp. nov.** differs from the nominative subspecies by the following combination of morphological characteristics: total length 702 mm; anterior pair of chin shields as long as broad; two postoculars; temporals 3 + 3; 15 dorsal scale rows slightly keeled in 7 scale rows at midbody; 194 ventrals; 96 subcaudals; dorsum light brown to beige, 73 weak dark vertebral spots; transverse dark bands on the body absent (Figs. 9D and 9E); upper postorbital stripes not contacting each other on the nuchal area forming a weak X-shaped dark marking with curved branches; ventral scales beige with dense brownish mottling not forming the interrupted midventral line.

**Description of the holotype:** Adult male, specimen in a good state of preservation (Fig. 8), body slender and notably flattened laterally; head comparatively large, narrowly elongated, clearly distinct from the thin neck, snout blunt; eye rather large, pupil vertical and slightly elliptical.

***Body size.*** SVL 524 mm; TaL 178 mm; TL 702 mm; TaL/TL: 0.254.

***Body scalation.*** Dorsal scales in 15–15–15 rows, slightly keeled in 7 upper scale rows at midbody, without apical pits; vertebral scales (three median rows) slightly enlarged; outermost dorsal scale row not enlarged; ventrals 194 (+ 1 preventral), lacking lateral keels; subcaudals 96, paired; cloacal plate single.

***Head scalation.*** Rostral not visible from above; nasal single; two internasals, much wider than long, narrowing and slightly curving back laterally (in dorsal view), anteriorly in contact with rostral, laterally in contact with nasal and loreal, posteriorly in contact with prefrontal, not contacting preocular; two large irregular pentagonal prefrontals, much larger than internasals and with a slightly diagonal suture between them, not contacting the eye; frontal scale hexagonal with the lateral sides parallel to the body axis, longer than wide, of the same size as the parietals; two preoculars; two suboculars; two postoculars, not fused with suboculars; one loreal in contact with prefrontal, not touching the eye; 7/7 supralabials, 3rd and 5th SL touching suboculars, none of them reaching the eye, 7th subocular the largest, elongate; temporals 3 + 3; 9/9 infralabials, the anterior most in contact with the opposite along the midline, bordering mental, anterior five pairs of infralabials bordering anterior chin shields, 3rd pair of infralabials in contact with each other (Fig. 8F); unpaired inframaxillary shield absent; two pairs of chin interlaced shields contacting each other, no mental groove under chin and throat; anterior chin shields relatively large, as long as broad, the second pair of chin shields much broader than long.

*Coloration.* In life, the dorsal and ventral surfaces of the head are uniform light brown dorsally, yellowish-beige ventrally (Fig. 9E). Head with two lateral postorbital stripes: the lower one is a thin dark-brown line starting from the lower posterior edge of the eye onto the anterior part of the last supralabial; the upper one is a strong dark-brown line running from postocular backwards to the medial dorsal scale rows on the neck; upper postorbital stripes not contacting each other on the nuchal area forming a weak X-shaped dark marking with curved branches. Upper labials marked with numerous irregular dark-brown speckling continuing and getting denser on lateral and dorsal surfaces of the head; 5th supralabial with a larger black spot. Dorsal surfaces with ca. 73 faint dark blotches along the vertebral keel; transverse dark bands on the body absent; ventral surfaces of body and tail yellowish cream with very sparse small black spots concentrating laterally, dark spots and speckles getting denser on the posterior portion of the belly. Iris yellowish-orange laterally and ventrally, light beige dorsally; pupil black. **In preservative:** After 2 years of storage in ethanol (Fig. 8) the general coloration pattern has not changed; light brown of the coloration of dorsum, head and eye has faded becoming grayish-brown; other features of coloration remain unchanged.

**Variation:** A single male specimen observed in Kaeng Krachan N.P., Phetchaburi Province, Thailand (specimen not collected) is overall similar to the holotype of the new subspecies, but demonstrates certain differences in color pattern, including more pronounced dark vertebral spots and a series of dark spots along the lower raw of dorsal scales (Fig. 9D), while the holotype has a more uniform coloration lacking dark markings on dorsum and body sides (Fig. 9E). Given the geographic proximity of the Kaeng Krachan N.P. (Fig. 1A) to the type locality of the new subspecies, and morphological similarity, we tentatively identify the Kaeng Krachan population as *P. c.* cf. *tenasserimicus* **ssp. nov.**; its taxonomic status requires further verification using morphological examination and molecular data.

**Comparisons:** *Pareas carinatus tenasserimicus* **ssp. nov.** differs from *P. c. carinatus* by its generally larger size (702 mm *vs.* 337–608 [485.2 ± 59.7] mm); a slightly higher number of ventrals (194 *vs.* 158–190 [170.4 ± 8.2]); a higher number of subcaudals (96 *vs.* 54–84 [68.1 ± 7.2]); by two postoculars (*vs.* single postocular); by the 3rd pair of infralabials in contact with each other (*vs.* not in contact); by a uniform light brown coloration of dorsum lacking transverse dark bands (*vs.* transverse dark bands present); and by upper postorbital stripes not contacting each other on the nuchal area forming a weak dark marking with curved branches (*vs.* usually contacting each other forming a dark X-shaped marking). For the detailed comparison of the two subspecies of *P. carinatus* see Table S12.

**Distribution:** To date the new subspecies is reliably known from only three localities in the southern portion of Tenasserim Range: the type locality in Suan Phueng District, Ratchaburi Province and Kaeng Krachan N.P., Phetchaburi Province of Thailand (locality 14, Fig. 1), and from Yaephyu area in Tanintharyi Region of Myanmar (locality 15, Fig. 1).

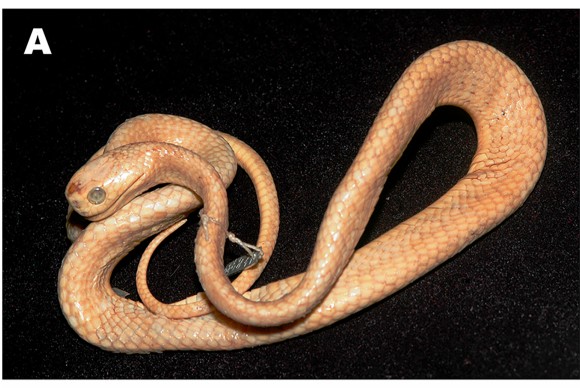

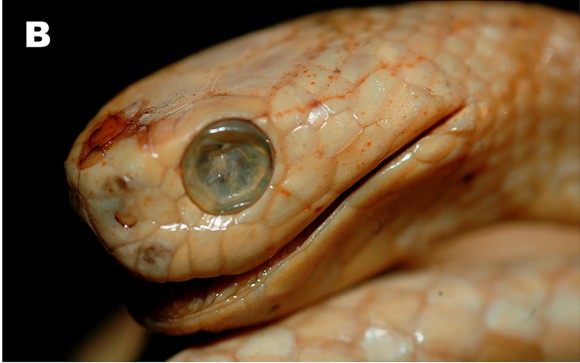

**Figure 10 Lectotype of *Pareas berdmorei* Theobald, 1868 in preservative (ZSI 8022, adult male).**
(A) General dorsolateral view of body; (B) lateral left aspect of head. Photos by I. Das.

**Etymology:** The new subspecies name "*tenasserimicus*" is a Latin toponymic adjective in nominative singular, adopting the masculine gender of the genus name *Pareas*, and is given in reference to the Tenasserim Mountain Range in western Thailand and southeastern Myanmar, where the new subspecies occurs.

**Recommended English name:** Tenasserim slug-eating snake.

**Ecology notes:** The new subspecies inhabits montane evergreen forests of the Tenasserim Range on elevations above 800 asl. This is a nocturnal snake, all three specimens were recorded at night while perching or crawling on the tree branches and bushes ca. 1–2 m above the ground. The diet of the new subspecies is not known in detail, though it likely consists of land snails or slugs. In Suan Phueng area (locality 14, Fig. 1), the new subspecies occurs in sympatry with *P. b. berdmorei*, but was not recorded in the same habitats: the new subspecies inhabits montane forests at ca. 800–1,000 m asl., while the specimens of *P. b. berdmorei* were recorded in lowland bamboo forest at 300 m asl. Other co-occurring species of *Pareas* include *P. margaritophorus*.

### *Pareas berdmorei* Theobald, 1868
Figures 6C–6F, 10–13; Tables S11–S13.

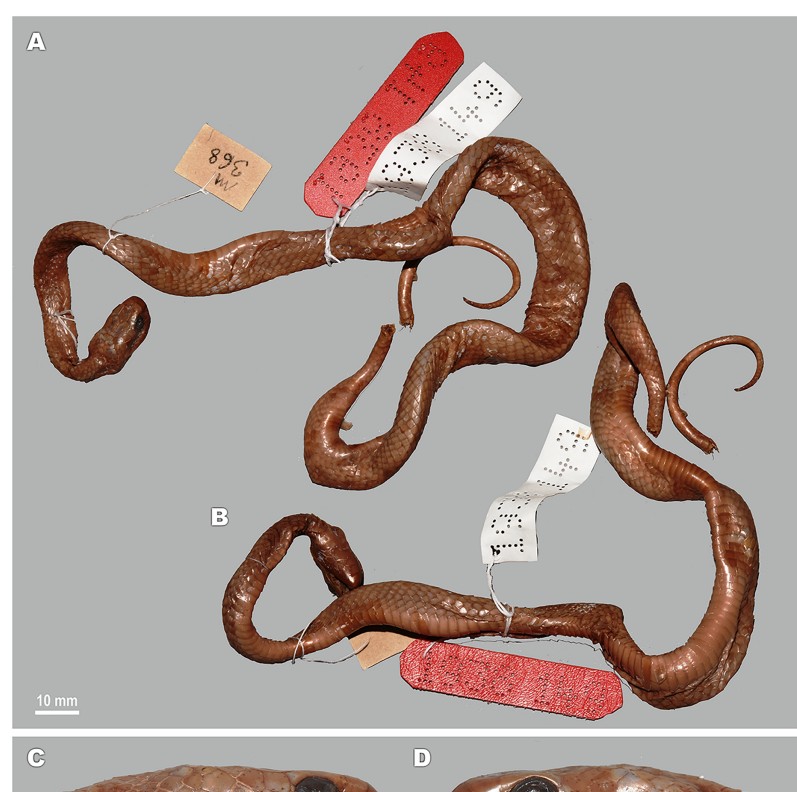

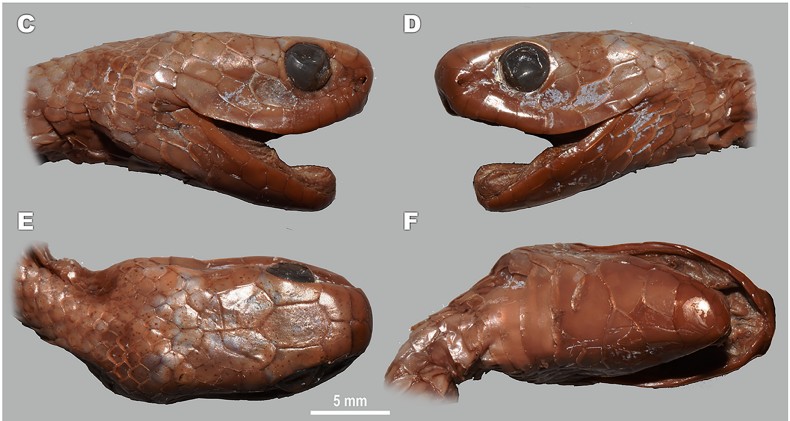

**Figure 11 Holotype of *Amblycephalus carinatus unicolor Bourret, 1934* in preservative (MNHN 1938.0149, adult female).** (A) Dorsal view of body; (B) ventral view of body; (C–F) head in lateral right, lateral left, dorsal, and ventral aspects, respectively. Photos by G. Vogel.

*Pareas berdmorei Theobald, 1868b*: 63.

**Chresonymy and synonymy:**

*Amblycephalus carinatus* unicolor *Bourret, 1934*.

*Pareas carinatus* — (in part) *Smith, 1943*; *Taylor, 1965*; *Yang & Rao, 2008*; *Nguyen, Ho & Nguyen, 2009*; *Teynié & David, 2010*; *Le et al., 2014*; *Wallach, Williams & Boundy, 2014*; *Chan-ard, Parr & Nabhitabhata, 2015*; *Pham & Nguyen, 2019*.

*Pareas berdmorei* — *Das, Dattagupta & Gayen, 1998*.

*Pareas carinatus unicolor* — (in part) *Nguyen, Ho & Nguyen, 2009*.

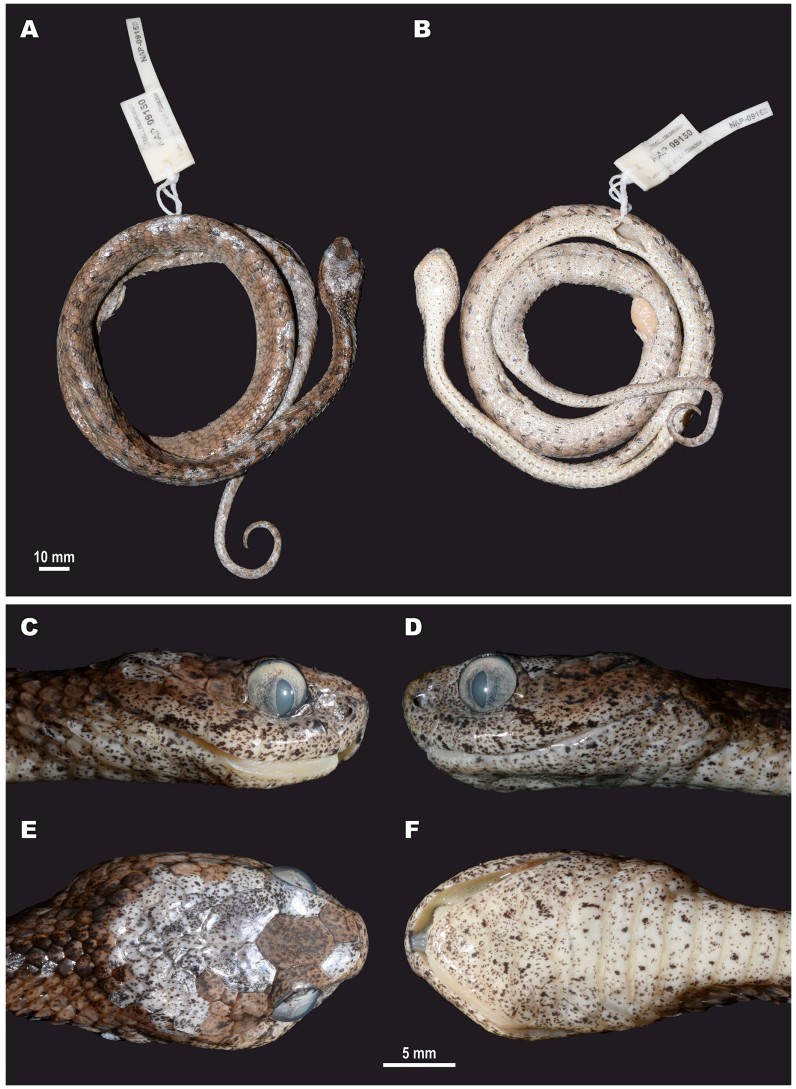

**Figure 12 Holotype of *P. berdmorei truongsonicus* ssp. nov. in preservative (ZMMU R-16801, adult female).** (A) Dorsal view of body; (B) ventral view of body; (C–F) head in lateral right, lateral left, dorsal, and ventral aspects, respectively. Photos by N. A. Poyarkov.

*Pareas menglaensis Wang et al., 2020.*

**Lectotype** (designated herein) (Figs. 6C and 10): ZSI 8022, adult male collected by T. M. Berdmore from "Tenasserim", corresponding to Mon Region in southeastern Myanmar according to *Das, Dattagupta & Gayen (1998)* (locality 16, Fig. 1). The specimen ZSI 8022 is designated herein as the lectotype as it is the best preserved specimen of the type series, and clearly corresponds to the morphological characteristics mentioned in the original description of *Pareas berdmorei Theobald, 1868b*, thus fulfilling the requirements of the Art. 74.7 of the Code (*ICZN, 1999*), modified by *International Commission on Zoological Nomenclature (2003*; Declaration 44).

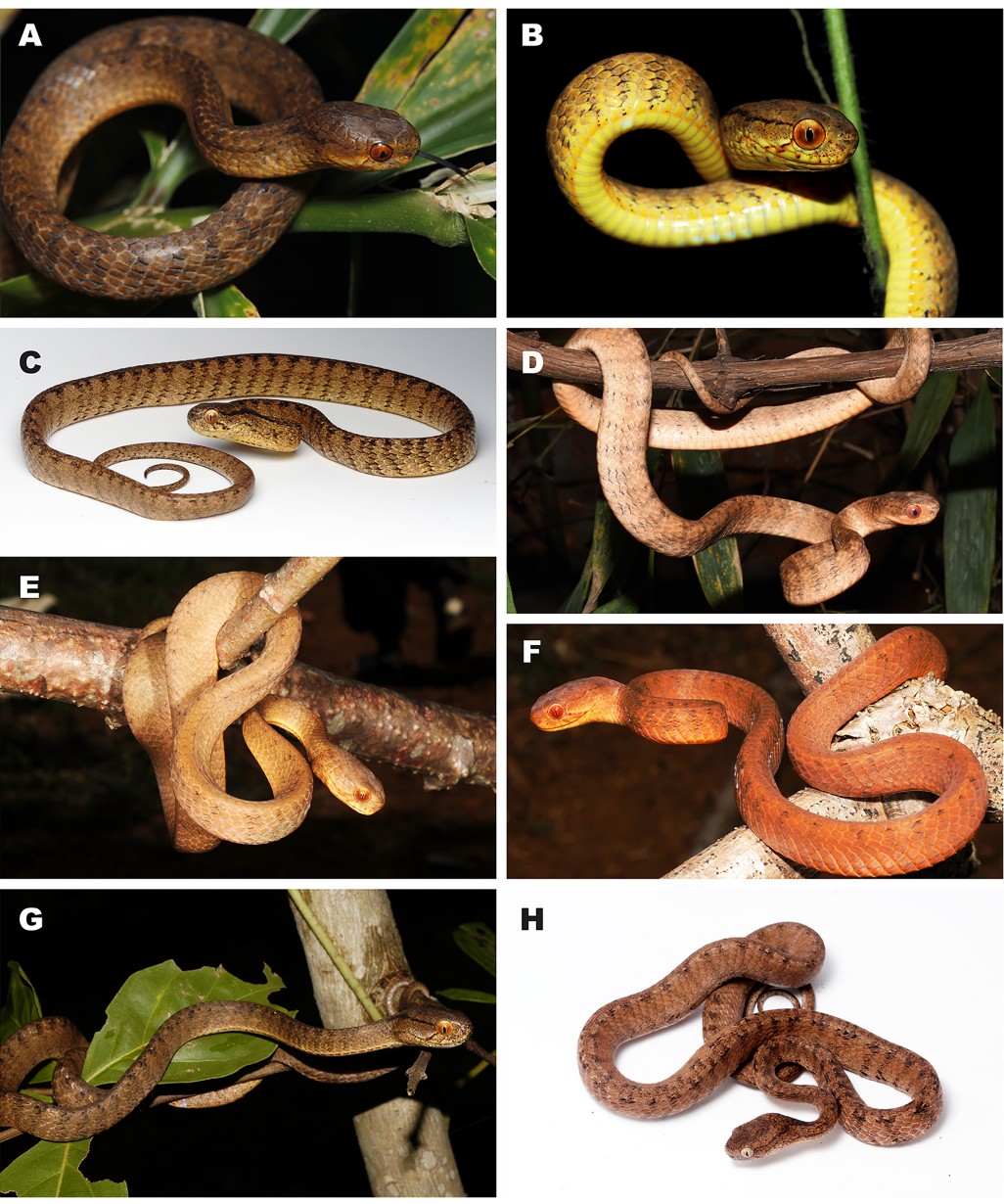

**Figure 13 Members of the *Pareas berdmorei* complex in life.** (A) *P. b. berdmorei* from Thung Yai Naresuan W.S., Kanchanaburi Province, Thailand; (B) *P. b. berdmorei* from Huai Kha Khaeng W.S., Uthai Thani Province, Thailand; (C) *P. b. berdmorei* from Phu Hin Rong Kla N.P., Phitsanulok Province, Thailand; (D) *P. b. berdmorei* from Jiangcheng, Pu'er City, Yunnan Province, China (close to the type locality of *P. menglaensis*); (E) *P. b. unicolor* from Cat Tien N.P., Dong Nai Province, Vietnam; (F) *P. b. unicolor* from Loc Bac Forest, Lam Dong Province, Vietnam; (G) *P. berdmorei* cf. *truongsonicus* **ssp. nov.** from Xe Pian N.P.A., Champasak Province, Laos; (H) *P. berdmorei truongsonicus* **ssp. nov.** from Nahin District, Khammouan Province, Laos (ZMMU R-16801, holotype in life). Photos by P. Pawangkhanant (A–B, H), N.A. Poyarkov (C, E–F), and G. Vogel (D, G).

**Paralectotype** (designated herein): one specimen, ZSI 8021, adult male, with the same collection information as the lectotype.

**Referred specimen:** ZSI 8023, juvenile, with the same collection information as the lectotype.
**Remark:** *Theobald (1868b)* described *Pareas berdmorei* based on a series of five specimens collected from "Tenasserim" by Major T. M. Berdmore. The description of *Pareas berdmorei* is based on a series of two adults; the three smaller specimens, which *Theobald (1868b)* were considered to be juveniles, were almost simultaneously described as *P. macularius* (*Theobald, 1868a*: 54). Therefore *Theobald (1868b)* proposed the new name for the two larger specimens "to prevent confusion of synonyms" (*Theobald, 1868b*: 63). According to *Das, Dattagupta & Gayen (1998)*, the type series of *P. macularius* includes three specimens ZSI 8024–26, while the specimens catalogued under the name *P. berdmorei* bear the museum numbers ZSI 8021–23. *Bhosale et al. (2020)* noted that only the specimens ZSI 8021 and ZSI 8022 are adults, therefore they correspond to the syntypes of *P. berdmorei* (Z. Mirza, 2021, personal communication).

We have obtained morphological data and a photograph of ZSI 8022, which we designate as the lectotype of *P. berdmorei*, as it is the best preserved specimen of the type series which fully agrees with the original description of *P. berdmorei* (*Theobald, 1868b*: 63). We designate ZSI 8021 as the paralectotype, while ZSI 8023, without nomenclatural status, is listed in the referred specimens. The designation of lectotype for *P. berdmorei* is necessary due to the historical confusion between the descriptions of *P. macularius* and *P. berdmorei*; thus fulfilling the requirements of the Art. 74.7 of the Code (*ICZN, 1999*).

**Updated diagnosis:** *Pareas berdmorei* differs from other members of the genus *Pareas* by the following combination of morphological characters: maximal total length of 770 mm; frontal scale hexagonal with its lateral sides parallel to the body axis; the anterior pair of chin shields broader than long; loreal and prefrontal not contacting the eye; generally 1 or 2 preoculars; regularly 2 (rarely 1 or 3) suboculars; generally single postocular (rarely 0 or 2); temporals 3 + 4 or 3 + 3; one to three median vertebral dorsal scale rows slightly enlarged; generally 8 (7–9) infralabials; 15 dorsal scale rows, of them 3–13 scale rows at midbody feebly keeled; 162–187 ventrals; 57–89 subcaudals, all divided; dorsum yellow-brown to orange, dark markings on dorsum variable; thin upper postorbital stripes continue to nape often forming elongated dark markings; iris uniform, color varies from beige to bright reddish-orange (*Theobald, 1868b*; *Bourret, 1934*; *Taylor, 1965*; *Ziegler et al., 2006*; *Yang & Rao, 2008*; *Le et al., 2014*; *Pham & Nguyen, 2019*; *Wang et al., 2020*; our data).

**Material examined:** In this study we used morphological data from 38 specimens of *P. berdmorei*, including the 21specimens examined directly (among them the lectotype of *Pareas berdmorei* Theobald and the holotype of *Amblycephalus carinatus unicolor* Bourret) and the published data for 17 specimens formerly listed as "*P. carinatus*" (*Taylor, 1965*; *Ziegler et al., 2006*; *Yang & Rao, 2008*; *Le et al., 2014*; and *Pham & Nguyen, 2019*), and "*P. menglaensis*" (*Wang et al., 2020*) (Table S11).

**Description of the lectotype** (ZSI 8022): Adult male, a well-preserved specimen, with coloration significantly faded due to the long time of preservation in ethanol (Fig. 10),

body slender and notably flattened laterally; head comparatively large, narrowly elongated, slightly distinct from neck, snout blunt; eyes large.

**Body size.** SVL 490 mm; TaL 120 mm; TL 610 mm; TaL/TL: 0.197.

**Body scalation.** Dorsal scales in 15–15–15 rows, slightly keeled in 9 scale rows at midbody, lacking apical pits; vertebral scales in three median rows slightly enlarged; the outermost dorsal scale row not enlarged; ventrals 174 (+ 1 preventral), lacking lateral keels; subcaudals 64; cloacal plate single.

**Head scalation.** Rostral not visible from above; single nasal; two internasals, much wider than long, narrowing and slightly curving back laterally (in dorsal view), anteriorly in contact with rostral, laterally in contact with nasal and loreal, posteriorly in contact with prefrontal, not contacting preocular; two large irregular pentagonal prefrontals, much larger than internasals and with a slightly diagonal suture between them, not in contact with eye; the frontal scale hexagonal with the lateral sides parallel to the body axis, longer than wide, smaller than parietals; two preoculars; one subocular; one postocular, not fused with subocular; one loreal in contact with prefrontal, not touching the eye; 7/7 supralabials, 3rd and 5th SL touching the subocular, none of them reaching the eye, 7th SL the largest, elongate; temporals 3 + 4; 8/8 infralabials, 3 pairs of chin shields interlaced, all notably broader than long, no mental groove under chin and throat; anterior chin shields relatively large.

**Coloration.** Due to preservation in ethanol for over 150 years, the colors of the holotype have significantly faded, the specimen is uniform light brownish-yellow (Fig. 10A); the present body pattern of the lectotype no longer retains the original characteristics, though the thin dark postorbital stripes are still discernable and faded to orange-brown (Fig. 10B). In the original description, *Theobald (1868b*: 63*)* gives the following information on the type coloration: "color is uniform ochraceous, with obsolete traces of vertical bands down the body; two dark lines converge on the nape; <…> belly white", indicating that the specimen has already significantly faded at the moment of the original description.

**Comparisons**: *Pareas berdmorei* is distinguishable from *P. carinatus* by the generally larger body size (554.2 ± 76.5 mm *vs.* 494.3 ± 73.3 mm); by slightly higher number of ventrals (177.3 ± 5.8 *vs.* 171.4 ± 9.3); slightly higher number of keeled dorsal rows (9.0 ± 2.8 *vs.* 6.1 ± 3.3); by generally less pronounced dark markings in the nuchal area, thinner postorbital stripes and a more uniform coloration of the iris; from *P. nuchalis* by prefrontal not contacting the eye (*vs.* in contact); by the absence of the ring-shaped black blotch on the nape (*vs.* present); by lower number of ventrals (177.3 ± 5.8 *vs.* 209.9 ± 5.3); lower number of subcaudals (71.2 ± 7.5 *vs.* 111.1 ± 6.1); and by the presence of keeled dorsal scale rows (*vs.* dorsals totally smooth). Morphological comparisons between all species of the subgenus *Pareas* are detailed in Table 2 and Table S13. *Pareas berdmorei* can be distinguished from other species of *Pareas* belonging to subgenus *Eberhardtia* **stat. nov.** by

having two or three distinct narrow suboculars (*vs.* one thin and elongated) and by having a hexagonal frontal with its lateral sides parallel to the body axis (*vs.* subhexagonal) (Tables S13 and S14).

**Distribution**: *Pareas berdmorei* is widely distributed across the mainland Indochina north from the Kra Isthmus (Fig. 1). Its range consists of the three major groups of populations (Fig. 1): (1) populations from southeastern Myanmar, western and northern Thailand, northern Laos and the southernmost parts of Yunnan Province of China; (2) populations from eastern Laos and central Vietnam located along the Truong Son Range; (3) populations from southern Vietnam and the adjacent parts of eastern Cambodia. As this species occurs in lowland to submontane tropical forests, it is seemingly absent from seasonally dry lowlands of central Indochina and the lower part of the Mekong Valley (Fig. 1). However, the occurence of *P. berdmorei* is expected in the Cardamom Mountains of Cambodia and the montane areas of eastern Thailand; therefore further field survey efforts are required to clarify the real extent of *P. berdmorei* distribution.

**Etymology:** Theobald (1868b) named his new species in honor of British naturalist Captain Thomas Matthew Berdmore (1811–1859), who was the collector of the type specimens.

**Recommended English name:** Berdmore's slug-eating snake.

**Remark:** The cumulative evidence from molecular and morphological data strongly suggests that the populations of "*P. carinatus*" from the mainland Indochina are divergent and morphologically different from *P. carinatus sensu stricto* from Malayan Peninsula and the Greater Sunda Islands. Our results thus agree with the data of Wang et al. (2020), who compared the samples of *P. carinatus* group from southern Yunnan Province of China and Peninsular Malaysia and based on the revealed differences described the Yunnan population as a new species *P. menglaensis*. Wang et al. (2020) postulated that *P. menglaensis* is endemic to China, but suggested that this species also may occur in the surrounding low mountainous areas of neighboring Laos and Myanmar. However, our analyses have demonstrated that the distribution of this lineage is much wider and covers the entire territory of the mainland Indochina, including the type localities of *Pareas berdmorei* Theobald, 1868 (Mon State, southern Myanmar), and of *Amblycephalus carinatus unicolor* Bourret, 1934 (Kampong Speu Province, eastern Cambodia), while the Yunnan population of "*P. menglaensis*" is deeply nested within this radiation (Fig. 3). We thus conclude that the name *Pareas berdmorei* Theobald, 1868, being the eldest available synonym, has to be applied to the mainland populations of *P. carinatus* species group, while *Amblycephalus carinatus unicolor* Bourret, 1934 and *Pareas menglaensis* Wang et al., 2020 represent subjective junior synonyms of this taxon.

Altogether, based on the concordant results of morphological and molecular analyses we report that three geographically restricted lineages exist within *P. berdmorei* (lineages 1–3, Fig. 3). Despite the significant morphological variation within *P. berdmorei*, these lineages can be readily distinguished from each other by a number of chromatic and

scalation characters. We propose to recognize three subspecies within *P. berdmorei*: *P. b. berdmorei* **stat. nov.** from Thailand, southern Myanmar and southern China (including *P. menglaensis* as a junior subjective synonym), *P. b. unicolor* **comb. nov.** from southern Vietnam and Cambodia, and a new subspecies *P. b. truongsonicus* **ssp. nov.** for populations from the Northern Annamite Mountains in Vietnam and Laos described below.

### *Pareas berdmorei berdmorei* Theobald 1868 stat. nov.
Figures 6C–6D, 10, 13A–13D; Tables S11 and S12.

*Pareas berdmorei* Theobald, 1868: 63.

**Chresonymy and synonymy:**
   *Pareas carinatus* — (in part) *Smith, 1943*; *Taylor, 1965*; *Yang & Rao, 2008*; *Wallach, Williams & Boundy, 2014*; *Chan-ard, Parr & Nabhitabhata, 2015*.
   *Pareas menglaensis Wang et al., 2020*.

**Updated diagnosis:** *Pareas berdmorei berdmorei* differs from other subspecies of *P. berdmorei* by the combination of the following morphological characters: body size large (TL 451–770 mm); anterior pair of chin shields wider than long; loreal and prefrontal not contacting the eye; two suboculars; one postocular; temporals generally 3 + 4 (rarely 3 + 3, or 2 + 3); three median vertebral scale rows slightly enlarged; 9 infralabial scales; 15 dorsal scale rows slightly keeled in 5–13 scale rows at midbody; 166–186 ventrals; 57–89 subcaudals, all divided; dorsum light brown to yellowish with distinct dark vertebral spots and 64–72 transverse dark brownish or blackish bands (Figs. 13A–13D); upper postorbital stripes weak generally not contacting each other on the nuchal area forming a narrow X-shaped dark marking with curved branches; ventral scales yellowish, generally immaculate, iris uniform from golden-bronze to orange (Figs. 13A–13D).

**Variation:** Measurements and scalation features of the subspecies *P. b. berdmorei* (*n* = 23) are summarized in Table S12. There is a certain variation among sexes observed in TaL/TL ratio and the number of subcadals scales: males have generally higher number of subcadals than females (73–89, average 78.33 ± 6.67, *n* = 6, in males *vs.* 57–82 average 70.67 ± 7.09, *n* = 12 in females). In coloration, the specimens of *P. b. berdmorei* showed variation in iris color: golden-bronze iris in specimens from eastern Thailand (Fig. 13C); to orange iris in specimens from southern Yunnan and northern Thailand (Figs. 13A, 13B and 13D). Specimens also varied in the arrangement of dark markings in the nuchal area: thicker dark-brown to black markings in specimens from Thailand (Figs. 13A–13C); less distinct dark markings in specimens from Laos and southern Yunnan (Fig. 13D). Other morphological features showed no significant variation among the specimens of the series examined.

**Distribution:** Southeastern Myanmar, Northern peninsular Thailand north of the Isthmus of Kra, western, northern and eastern mainland Thailand, northern Laos, northern Vietnam, southernmost Yunnan Province of China; the southernmost known locality is in Suan Phueng District, Ratchaburi Province, Thailand (Fig. 1).

**Recommended English name and Etymology:** as for *Pareas berdmorei*.

**Ecology notes:** In Suan Phueng area (locality 14, Fig. 1), *P. b. berdmorei* occurs in sympatry with *P. c. tenasserimicus* **ssp. nov.**, though the two taxa are restricted to different habitats (see the account for *P. c. tenasserimicus* **ssp. nov.** for details). Across its range, *P. b. berdmorei* occurs in sympatry with various congeners, including *P. margaritophorus*, *P. macularius*, *P. geminatus*, and *P. xuelinensis*.

### *Pareas berdmorei unicolor* (*Bourret, 1934*) comb. nov.
Figures 6F, 11, 13E and 13F; Tables S11 and S12.

*Amblycephalus carinatus unicolor* *Bourret, 1934*: 15.

**Chresonymy:**
   *Pareas carinatus unicolor* — *Nguyen, Ho & Nguyen, 2009*.

**Holotype**: MNHN 1938.0149, adult female collected by R. Bourret from "Kompong Speu" (indicated as "Kompong Pseu" on the original label, now Kampong Spoe), Kampong Spoe Prov., eastern Cambodia.

**Updated diagnosis:** *Pareas berdmorei unicolor* **comb. nov.** differs from other subspecies *P. berdmorei* by the following combination of morphological characteristics: body size medium to small (TL 459–576 mm); anterior pair of chin shields slightly longer than broad; loreal and prefrontal not contacting the eye; two (rarely one) suboculars; two (rarely one) postoculars; temporals generally 3 + 3 (rarely 3 + 4); three median vertebral scale rows slightly enlarged; 9 infralabial scales; 15 dorsal scale rows slightly keeled in 3–9 scale rows at midbody; 162–180 ventrals; 57–75 subcaudals, all divided; dorsum uniform yellow-ochre to bright orange lacking distinct dark vertebral spots and transverse dark bands (Figs. 13E and 13F); upper postorbital stripes generally absent or weakly discernable not contacting each other on the nuchal area; ventral scales yellowish to orange, generally immaculate, iris uniform bright orange-red (Figs. 13E and 13F).

**Description of the holotype** (MNHN 1938.0149): Adult female, specimen partially dehydrated due to preservation in ethanol for a long time (Fig. 11), body slender and notably flattened laterally; head comparatively large, narrowly elongated, clearly distinct from thin neck, snout blunt; eye rather large, pupil vertical and slightly elliptical.

*Body size.* SVL 390 mm; TaL 96 mm; TL 486 mm; TaL/TL: 0.198.

*Body scalation.* Dorsal scales in 15–15–15 rows, slightly keeled in seven scale rows at midbody, and without apical pits; three median vertebral scale rows slightly enlarged; the outermost dorsal scale row not enlarged; ventrals 164 (+ 2 preventrals), lacking lateral keels; subcaudals 64, all divided; cloacal plate single.

*Head scalation.* Rostral not visible from above; nasal entire; two internasals, much wider than long, narrowing and slightly curving back laterally (in dorsal view), anteriorly in contact with rostral, laterally in contact with nasal and loreal, posteriorly in contact with

prefrontal, not contacting preocular; two large pentagonal prefrontals, much larger than internasals and with a slightly diagonal suture between them, not contacting the eye; single hexagonal frontal scale with its lateral sides parallel to the body axis, frontal longer than wide, smaller than parietals; two preoculars; one subocular; one postocular, not fused with subocular; one loreal in contact with prefrontal, not touching the eye; 7/7 supralabials, 3rd to 5th SL touching the subocular, none of them reaching the eye, 7th by SL the largest, elongate; temporals 3 + 3; 8/8 infralabials, the anterior most in contact with the opposite along midline, bordering mental, anterior 5 pairs of infralabials bordering anterior chin shields; 3 pairs of chin shields interlaced, no mental groove under chin and throat; anterior chin shields relatively large, slightly longer than broad, followed by the two pairs of chin shields that are much broader than long.

*Coloration.* Due to preservation in ethanol for almost a century, the color pattern has significantly faded; as the consequence the specimen no longer retains the original coloration characteristics. Presently the specimen is uniform dark reddish-brown with no pattern discernable on the ground color (Fig. 11). The original description contains the following information on the type specimen coloration: "the color is light reddish brown, absolutely uniform, without any spots on the body or head, yellower and lighter below" (*Bourret, 1934*: 15).

**Variation:** Measurements and scalation features of the subspecies *Pareas berdmorei unicolor* **comb. nov.** ($n$ = 9) are presented in Table S11. There is a certain sexual dimorphism observed in body size and the number of subcadal scales: males (TL 505.0–576.0 mm, average 552.2 ± 40.9 mm, $n$ = 3) have slightly larger body size than females (TL 459.0–538.1 mm, average 498.4 ± 32.6 mm, $n$ = 6); males also have a generally higher number of subcadals than females (73–75, average 73.7 ± 1.2, $n$ = 3 in males *vs.* 58–75, average 66.3 ± 5, $n$ = 6 in females). Coloration of the examined specimens varied in the ground color (from yellow-ochre to bright orange), and in the dark markings on the head and nuchal area (from upper postorbital stripes absent to weakly discernable). Other morphological features showed no significant variation among the examined specimens.

**Comparisons:** *Pareas berdmorei unicolor* **comb. nov.** differs from *P. b. berdmorei* by slightly lower number of ventrals (162–180 [average 173.6 ± 5.1] *vs.* 166–186 [average 178.10 ± 5.19]), by generally lower number of keeled dorsal scale rows (3–9 [average 6.8 ± 1.9] *vs.* 5–13 [average 9.6 ± 2.2]), and by uniform orange to beige coloration lacking dark markings and transverse bands (*vs.* present) and brighter orange-red coloration of iris (*vs.* golden-bronze to orange). *Pareas berdmorei unicolor* **comb. nov.** differs from *P. b. truongsonicus* **ssp. nov.** described below by slightly smaller total length (459–576 mm [average 516.3 ± 42.5 mm] *vs.* 488–637 mm [average 587.0 ± 67.6]), by a lower number of ventrals (162–180 [average 173.6 ± 5.1] *vs.* 167–187 [average 179.5 ± 9.6]), by lower number of keeled dorsal scale rows (3–9 [average 6.8 ± 1.9] *vs.* 13), and by uniform orange to beige coloration lacking dark markings and transverse bands (*vs.* dark markings

present) and brighter orange-red coloration of iris (*vs*. off-white to golden). For the detailed comparisons of the three subspecies of *Pareas berdmorei* see Table S12.

**Distribution**: Based on our morphological and molecular data, *P. b. unicolor* **comb. nov.** inhabits the lowland and foothill tropical forests of southern Vietnam and eastern Cambodia (Fig. 1). The actual extend of the subspecies distribution in central Vietnam and central Cambodia is still unclear and requires further survey efforts.

**Etymology:** The subspecies name "*unicolor*" is a Latin adjective in nominative singular meaning "monochrome" and was given in reference to the uniform coloration of this snake.

**Recommended English name:** Cochinchinese slug-eating snake.

**Ecology notes:** In Di Linh District, Lam Dong Province of southern Vietnam (locality 37, Fig. 1), *P. b. unicolor* **comb. nov.** occurs in sympatry with *P. temporalis* described below; these two taxa were recorded in the same habitat within the mid-elevation evergreen tropical forests of the Langbian Plateau (see the account for *P. temporalis* for details). Across its range, *P. b. unicolor* occurs in sympatry with other congeners, including *P. margaritophorus*, *P. macularius*, and *P. formosensis*.

### *Pareas berdmorei truongsonicus* Poyarkov, Nguyen TV, Vogel, Brakels & Pawangkhanant ssp. nov.
[urn:lsid:zoobank.org:act:3E45EE5B-8DD5-4FB1-814A-76DC2B821E29]
Figures 6E, 12, 13G–13H; Tables S11 and S12.

**Chresonymy:**
   *Pareas carinatus* — (in part) *Ziegler et al., 2006*, 2016; *Nguyen, Ho & Nguyen, 2009*; *Teynié & David, 2010*; *Le et al., 2014*; *Pham & Nguyen, 2019*.

**Holotype**: ZMMU R-16801 (field number NAP-09150), adult female collected by N. A. Poyarkov, P. Brakels, P. Pawangkhanant and T. V. Nguyen from limestone forest near the Tham Mangkon Cave, in Ban Nahin-Nai District, Khammouan Province, central Laos (N 18.22111, E 104.81243; elevation 526 m asl.) on July 14, 2019.

**Paratype**: ZMMU R-14796, adult male collected by N. A. Poyarkov and N. L. Orlov from limestone forest in environs of Kim Lich Village, Kim Hoa Commune, Tuyen Hoa District, Quang Binh Province, central Vietnam (N 18.01206, E 105.92215; elevation 41 m asl.) on September 7, 2015.

**Referred specimens:** ZFMK 82890, adult male collected on July 5, 2014, and VNUH 15.6.'05-1 collected on June 15, 2005 from Phong Nha - Ke Bang National Park, Quang Binh Province, Vietnam (see *Ziegler et al., 2006*).

**Diagnosis:** *Pareas berdmorei truongsonicus* **ssp. nov.** differs from other subspecies of *P. berdmorei* by the combination of the following morphological characters: maximal total length of 637 mm; anterior pair of chin shields as long as broad; loreal and prefrontal not
contacting the eye; one subocular; one postocular; temporals 3 + 4; three median vertebral scale rows slightly enlarged; 9 infralabial scales; 15 dorsal scale rows keeled in 13 scale rows at midbody; 167–187 ventrals; 66–80 subcaudals, all divided; dorsum light brown with distinct dark-brown vertebral spots and 68–71 transverse dark bands, and with dense brownish-gray mottling covering dorsal, lateral and ventral surfaces of body and head (Figs. 13G–13H); upper postorbital stripes discernable, contacting each other on the nuchal area forming a clear Y-shaped pattern; ventral scales yellowish-white with dense brownish mottling, iris uniform off-white to golden (Figs. 13G–13H).

**Description of the holotype** (ZMMU R-16801): Adult female, specimen in a good state of preservation (Fig. 12), body slender and notably flattened laterally; head large, narrowly elongated and flattened, clearly distinct from thin neck, snout blunt; eye large, pupil vertical and slightly elliptical.

*Body size.* SVL 499 mm; TaL 123 mm; TL 622 mm; TaL/TL: 0.198.

*Body scalation.* Dorsal scales in 15–15–15 rows, the medial 13 scale rows slightly keeled at midbody, all dorsal scales lacking apical pits; three median vertebral scale rows enlarged; outermost dorsal scale row not enlarged; ventrals 187 (+ 1 preventral), all lacking lateral keels; subcaudals 66; cloacal plate single.

*Head scalation.* Rostral not visible from above; nasal single; internasals two, much wider than long, narrowing and slightly curving back laterally (in dorsal view), anteriorly in contact with rostral, laterally in contact with nasal and loreal, posteriorly in contact with prefrontal, not contacting preocular; two large irregular pentagonal prefrontals, much larger than internasals and with a slightly diagonal suture between them, not contacting the eye; the single frontal scale hexagonal with the lateral sides parallel to the body axis, longer than wide, smaller than parietals; one preocular; one subocular; one postocular, not fused with subocular; one loreal in contact with prefrontal, not touching the eye; 7/7 supralabials, 3rd to 5th SL touching the subocular, none of them reaching the eye, 7th SL the largest, elongate; temporals 3 + 4; 9/9 infralabials, the anterior most in contact with the opposite along the midline, bordering mental, anterior 5 pairs of infralabials bordering the anterior chin shields; 3 pairs of chin shields interlaced, no mental groove under chin and throat; anterior chin shields relatively large, as long as broad, followed by two pairs of chin shields that are much broader than long.

*Coloration.* In life, dorsal surfaces of the head brownish with numerous marbled markings and dense brownish mottling (Fig. 13H). Head with two lateral postorbital stripes: the lower one is thick dark-brown line continuing from the middle of the eye onto the anterior part of the last supralabial; the upper one is a slightly thinner dark line running from postocular backwards to the dorsal scales on the neck (Fig. 12D). The upper postorbital stripes from the both sides of the body meet each other in the nape area forming a dark Y-shaped pattern (Fig. 13H). Lateral and ventral surfaces of the head marked with a dense brown dusting and larger dark spots (Figs. 12D and 12E). Dorsal surfaces light-brown with ca. 68 faint vertical dark brown bands. Ventral surfaces of the head, body

and tail yellowish-cream with dense brown dusting. Iris uniform off-white, pupil black. **In preservative:** After two years of storage in ethanol the general coloration pattern has not changed (Fig. 12); yellowish tint in the coloration of dorsum, the head and eyes have faded becoming grayish-brown; other coloration features remain unchanged.

**Variation:** Measurements and scalation features of the subspecies Pareas *berdmorei truongsonicus* **ssp. nov.** (*n* = 4) are presented in Table S11. The two specimens examined in our study have a very similar coloration with the two specimens from Phong Nha-Ke Bang N. P. (ZFMK 82890 and VNUH 15.6.'05-1) reported by *Ziegler et al. (2006)*; this population is geographically close (ca. 50 km direct distance; see Fig. 1C) to the loaclity of the new subspecies in Kim Hoa Commune, Tuyen Hoa District, Quang Binh Province. However, the members of the type series of the new subspecies have slightly lower number of ventral and subcaudal scales as compared to the specimens reported in *Ziegler et al. (2006)*: VEN 187 *vs.* 167–177; SC 66–73 *vs.* 78–80. We also report on a population from Xe Pian N.P.A., Champasak Province, Laos (Fig. 1B) which agrees well with the specimens examined in our study in coloration (Fig. 13G); the morphological or genetic data on this population is lacking. Therefore, in this study we tentatively assign the specimens from Phong Nha-Ke Bang N.P. and from Xe Pian N.P.A. to *P.* cf. *b. truongsonicus* **ssp. nov.**; the taxonomic status of this population has to be clarified in the future.

**Comparisons:** In our sample of four specimens *Pareas berdmorei truongsonicus* **ssp. nov.** differs from *P. b. berdmorei* by slightly higher of number of ventrals (187 *vs.* 166–186 [178.10 ± 5.19]), by slightly higher number of keeled dorsal scale rows (13 *vs.* 5–13 [9.60 ± 2.20]); by the dense brownish mottling and bigger brown spots on dorsal, lateral, and ventral surfaces of the head and body (*vs.* ventral surfaces immaculate, lateral and dorsal surfaces with sparse dusting); and by uniform off-white to golden color of iris (*vs.* golden-bronze to orange). The new subspecies differs from *P. b. unicolor* **comb. nov.** by slightly larger maximal total length (622–637 mm *vs.* 459–576 mm [516.3 ± 42.5 mm]), by a generally higher number of ventrals (187 *vs.* 162–180 [173.6 ± 5.1]), by a higher of number keeled dorsal scale rows (13 *vs.* 3–9 [6. 8 ± 1.9]); by the presence of dark markings on dorsum and ventral surfaces, including the dark transverse bands and brownish mottling (*vs.* uniform orange to beige coloration lacking dark markings), and by off-white to golden coloration of iris (*vs.* bright orange-red color of iris). Detailed comparisons of the three subspecies of *Pareas berdmorei* are presented in Table S12.

**Distribution**: To date the new subspecies is known only from the northern portion of the Annamite (Truong Son) Mountain Range in central Vietnam and eastern Laos (localities 27–28, Fig. 1).

**Etymology:** The new subspecies name "*truongsonicus*" is a Latin toponymic adjective in nominative singular, adopting the masculine gender of the genus name *Pareas*, and is given in reference to the Truong Son (Annamite) Mountain Range in Vietnam and Laos, where the new subspecies occurs.

**Recommended English name:** Annanmite slug-eating snake.

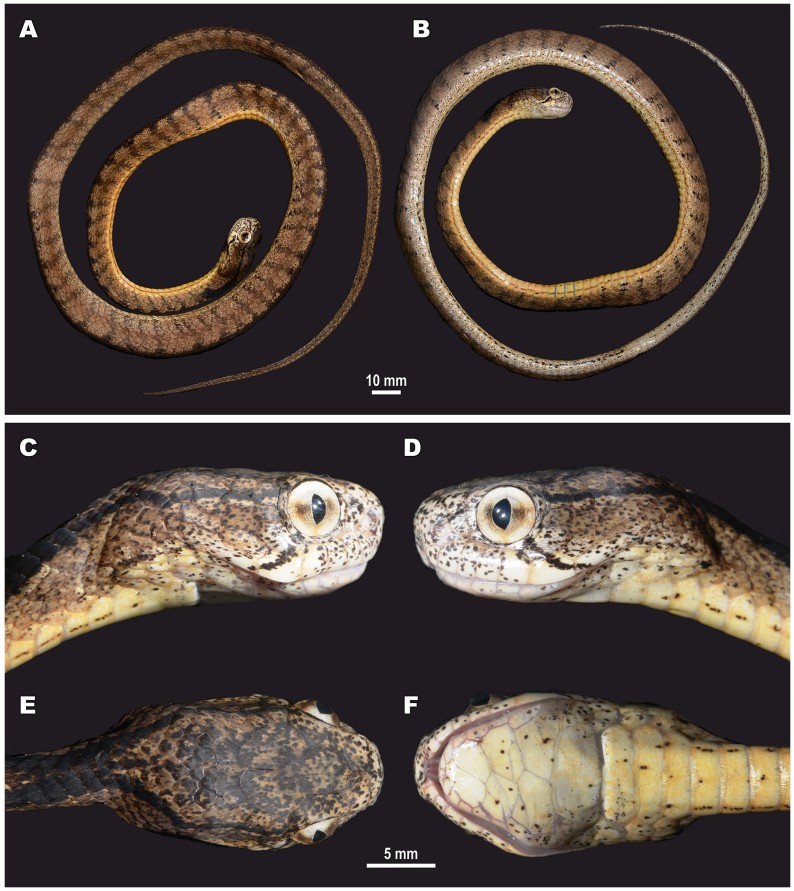

**Figure 14 Holotype of *Pareas kuznetsovorum* sp. nov. in life (ZMMU R-16802, adult male).** (A) Dorsal view of body; (B) ventral view of body; (C–F) head in lateral right, lateral left, dorsal, and ventral aspects, respectively. Photos by N. A. Poyarkov.

**Ecology notes:** In Tuyen Hoa District, Quang Binh Province of Vietnam, where the paratype of the new subspecies was collected, it was recorded in sympatry with *P. margaritophorus* and *P. formosensis*. This taxon seems to be associated with karst evergreen forests (*Ziegler et al., 2006*; this study). As other members of the genus *Pareas*, *Pareas berdmorei truongsonicus* **ssp. nov.** is a nocturnal semi-arboreal snake, all specimens were recorded while crawling on branches of bushes ca. 1 m above the ground or on limestone rocks; the diet is unknown but it likely includes terrestrial mollusks.

### *Pareas kuznetsovorum* Poyarkov, Yushchenko & Nguyen TV sp. nov.
[urn:lsid:zoobank.org:act:1CD26CB3-F3E9-4370-B501-6F678851C9FB]
Figures 6G, 14–15; Table 2; Tables S11 and S13.

**Holotype:** ZMMU R-16802 (field number NAP-10333), adult male collected by N. A. Poyarkov from the lowland semideciduous monsoon forest in Song Hinh Protected Forest, Song Hinh District, Phu Yen Province, southern Vietnam (N 12.77522, E 109.04606; elevation 583 m asl.) on January 16, 2021.

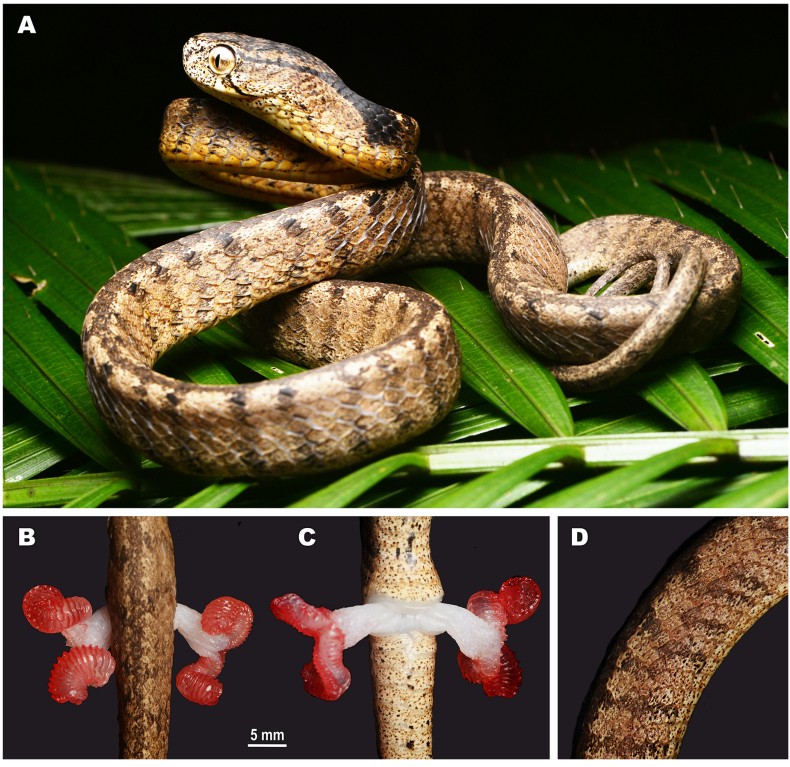

**Figure 15 Holotype of *Pareas kuznetsovorum* sp. nov. in life (ZMMU R-16802, adult male) from Song Hinh, Phu Yen Province, Vietnam.** (A) general view; (B and C) sulcal and asulcal aspects of fully everted hemipenis; (D) close-up of midbody showing smooth dorsal scales. Photos by N. A. Poyarkov.               

**Diagnosis:** *Pareas kuznetsovorum* **sp. nov.** differs from other members of the genus *Pareas* by the combination of the following morphological characteristics: total length up to 639 mm; anterior pair of chin shields longer than broad; loreal and prefrontal not contacting the eye; two suboculars; one postocular; temporals 3 + 4; the single median vertebral scale row slightly enlarged; 7 supralabial scales; 7–8 infralabial scales; 15 dorsal scale rows, all smooth; 167 ventrals; 87 subcaudals, all divided; dorsum tan to light brown with distinct dark-brown vertebral line, blackish vertebral spots and 70 transverse dark-brown bands; upper postorbital stripes thick, black, contacting each other on the nuchal area forming a dark black Ψ-shaped chevron; lower postorbital stripes thin, black, reaching the anterior part of SL7, not continuing to the lower jaw and chin; belly yellow with sparce dark-gray dusting and brown elongated spots forming three longitudinal lines on ventrals; iris uniform off-white with beige lateral parts.

**Description of the holotype** (ZMMU R-16802): Adult male, specimen in a good state of preservation (Fig. 14); body slender and notably flattened laterally; head large, narrowly elongated, clearly distinct from thin neck (head more than two times wider than neck width near the head basis); snout blunt; eye rather large, pupil vertical and elliptical.

***Body size.*** SVL 478 mm; TaL 161 mm; TL 639 mm; TaL/TL: 0.252.

*Body scalation.* Dorsal scales in 15–15–15 rows, all scales smooth and lacking apical pits; vertebral scales slightly enlarged; the outermost dorsal scale row not enlarged; ventrals 167 (+ 1 preventral), lacking lateral keels; subcaudals 87, all divided; cloacal plate single.

*Head scalation.* Rostral not visible from above; nasal single; two internasals, much wider than long, narrowing and slightly curving back laterally (in dorsal view), anteriorly in contact with rostral, laterally in contact with nasal and loreal, posteriorly in contact with prefrontal, not contacting preocular; two large pentagonal prefrontals, much larger than internasals and with a straight suture between them, not in contact with eye; one hexagonal frontal, longer than wide, with the lateral sides parallel to the body axis, roughly the same size as the parietals; single preocular; single postocular, semicrescentic in shape, not fused with subocular; two suboculars; single loreal, in contact with preocular, prefrontal, internasal and nasal, not touching the eye; 7/7 supralabials, 3rd to 5th SL touching the subocular, none of them reaching the eye, 7th by far the largest, elongated; 1/1 supraoculars; 3/3 anterior temporals and 3/4 posterior temporals; 8/7 infralabials, the anteriormost in contact with opposite along midline forming a straight suture, bordering mental, the anterior 5 pairs of infralabials bordering the anterior chin shields; 3 pairs of chin shields interlaced, no mental groove under chin and throat; anterior chin shields relatively large, notably longer than broad, followed by the two pairs of chin shields that are much broader than long.

*Hemipenial morphology.* Fully everted hemipenis symmetrical, bilobed, forked (Figs. 15B and 15C); the surface from base to crotch smooth, with several (5–6) weakly discernable dermal ridges on the asulcal surface (Fig. 15C) and few (3–4) shallow folds on the sulcal surface (Fig. 15B). Sulcus spermaticus deep, with fleshy swollen edges, bifurcate into two separate canals towards or on the apical lobes. Apical lobes curved with well-developed ornamentation, covered with large fleshy transverse occasionally interwining folds, separated with deep slits and forming a complex pattern resembling the bellows of an accordion (Figs. 15B and 15C).

*Coloration.* In life, the dorsal surfaces of the head brownish with dense darker marbling (Fig. 15A). Head laterally off-white with dark-brown spots and blotches, ventrally yellowish-white with few small black spots. Head with two lateral postorbital stripes: the lower one is a thin black line starting from the posterior portion of subocular and running ventrally and posteriorly towards lower temporals to the posterior part of the 6th supralabial and further to the anterior part of 7th supralabial; the upper one is a well-defined thick black line starting from the upper part of postocular backwards to the dorsal scales of neck (Figs. 14C and 14D), where it joins a large rectangular black spot covering the nape, overall forming a dark black Ψ-shaped chevron pattern (Fig. 14E). Upper labials marked with a dense brown dusting. Dorsal surfaces of the body tan to light brown with a distinct dark-brown line running along the vertebral scale row, and with about 70 black vertebral spots and transverse dark-brown bands (Figs. 14A and 14B); ventral surfaces of the head, body and tail yellowish with sparce dark-gray dusting and brown elongated spots forming three longitudinal lines on ventrals (Figs. 14A and 14B).

Iris uniform off-white with beige lateral parts; pupil black (Figs. 14C and 14D). **In preservative**: After 6 months of storage in ethanol the general coloration pattern has not changed; the tan coloration of dorsum slightly faded becoming light grayish-brown, light coloration on head and iris faded becoming brownish; other features of coloration remain unchanged.

**Comparisons:** *Pareas kuznetsovorum* **sp. nov.** differs from *P. berdmorei* by all dorsal scales smooth (*vs.* 3–13 dorsal scale rows keeled), higher number of subcaudals (87 *vs.* 63–78 [average 71.13 ± 7.23]), by the presence of black chevron on the nuchal area (*vs.* absent); the new species further differs from *P. carinatus* by the presence of two postoculars (*vs.* single or absent); by a generally higher number of subcaudals (87 *vs.* 54–96 [average 69.24 ± 8.98]), by all dorsal scale rows smooth (*vs.* 3–11 dorsal scale rows keeled [average 6.52 ± 2.94]), by the presence of black nuchal chevron (*vs.* absent); and by a lower number of enlarged vertebral scale rows (1 *vs.* 3 [average 2.83 ± 0.56]); it further differs from *P. nuchalis* by prefrontal not contacting the eye (*vs.* in contact); by lower number of ventrals (167 *vs.* 201–220 [average 209.89 ± 5.25]); and by a lower number of subcaudals (87 *vs.* 102–120 [average 111.11 ± 6.05]). Morphological comparisons between all species of the subgenus *Pareas* are detailed in Table S13.

**Distribution:** To date *Pareas kuznetsovorum* **sp. nov.** is known only from the type locality in the north-eastern foothills of the Langbian Plateau in Phu Yen Province of Vietnam (locality 41, Fig. 1). Though only single specimen of the new species is known up to date, its occurrence is expected in the remaining fragments of lowland to mid-elevation evergereen forests of the north-western slopes of the plateau, particularly in the adjacent parts of Dak Lak and Khanh Hoa provinces of southern Vietnam.

**Etymology:** The new species name "*kuznetsovorum*" is the plural possessive form of the family name Kuznetsov. This species is named in honor of two biologists, Andrei N. Kuznetsov and Svetlana P. Kuznetsova. They have greatly contributed to organization of biological expedtions of the Joint Russian-Vietnamese Tropical Center in various parts of Vietnam from 1996 to 2021; without their enthusiasm and support our fieldwork in Vietnam, including the expedition during which the holotype of the new species was collected, would have not been possible.

**Recommended English name:** Kuznetsovs' slug-eating snake.

**Ecology notes:** The single specimen of the new species was collected in middle January during the period where most of reptile species were not active; the specimen was recorded at 00:00 h while perching on a *Calamus* sp. palm leaf near a forest trail ca. 1.5 m above the ground, when the air temperature was around 12 °C under a drizzling rain. The specimen was not moving. No other members of Pareidae were recorded in the area of survey. The diet of the new species is unknown but, as in other congeners, it presumably consists of terrestrial mollusks.

**Pareas nuchalis species group**

The monophyly of the *nuchalis* group is strongly supported in BI-analysis, and got only poor support in ML-analysis (0.99/80); this group includes *P. nuchalis* from Borneo and Sumatra (lineage 9, Fig. 3) and two lineages from the montane areas in central and southern Vietnam, one of which we describe below as a new species, forming a strongly-supported group (1.0/100) (lineages 7–8, Fig. 3). The lineage inhabiting the Kon Tum – Gia Lai Plateau in central Vietnam represents *Pareas abros* **sp. nov.** (lineage 7, Fig. 3), and the second lineage from Langbiang Plateau in southern Vietnam corresponds to the recently described *P. temporalis* (lineage 8, Fig. 3). All members of the *nuchalis* group are characterized by the presence of a large black ring-shaped blotch on the nape, connected to the upper and lower postorbital stripes anteriorly.

**Pareas nuchalis (*Boulenger, 1900*)**
Figures 6H, 16–17; Table 2; Tables S11 and S13.

*Amblycephalus nuchalis Boulenger, 1900*: 185.

**Chresonymy:**

*Amblycephalus nuchalis* — *De Rooij, 1917*: 277.
*Pareas nuchalis* — *Malkmus & Sauer, 1996*; *Malkmus et al., 2002*; *Wallach, Williams & Boundy, 2014*: 537.
*Pareas carinatus* — (in part) *David & Vogel, 1996*.

**Holotype:** NHMUK 1912247 (formely BMNH 1901.5.14.2), adult male from Saribas, Betong Division, State of Sarawak, Borneo, Malaysia (approximately N 1.410, E 111.527; elevation 15 m asl.), collected by A. H. Everett.

**Updated diagnosis:** *Pareas nuchalis* differs from other members of the genus *Pareas* by the following combination of morphological characters: maximal total length up to 678 mm; anterior pair of chin shields longer than broad; loreal not contacting the eye; prefrontal in contact with the eye; 1–3 suboculars; 1–2 postoculars; temporals generally 3 + 4 or 3 + 3; one to three median vertebral scale rows slightly enlarged; 7–8 supralabial scales; generally 7 (rarely 6 or 8) infralabials; 15 dorsal scale rows at midbody, all totally smooth; 201–220 ventrals; 102–120 subcaudals, all divided; dorsum tan to light brown with weak dark-brown vertebral spots and 61–78 distinct transverse dark-brown bands (Figs. 16 and 17); upper postorbital stripes thick, black, bifurcating posterior to the secondary temporals, forming a vertical black bar to the mouth angle; upper postorbital stripes contacting each other on the nuchal area forming a large black ring-shaped blotch (Fig. 17); lower postorbital stripes thick, black, reaching the anterior part of SL6, often continuing to the lower jaw and chin; belly yellowish immaculate or with sparce brown dusting (Figs. 16 and 17); iris in life whitish with brownish speckles and veins getting denser around the pupil (Fig. 17) (*Boulenger, 1900*; *Stuebing, Inger & Lardner, 2014*; our data).

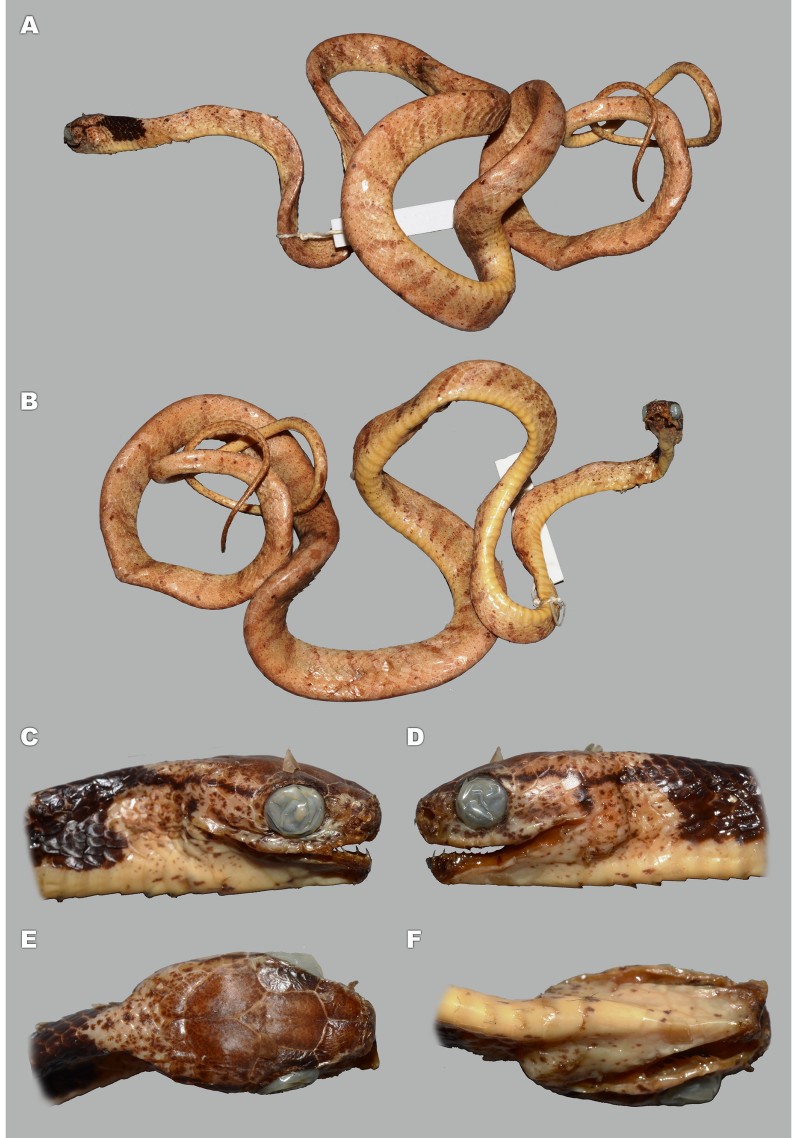

**Figure 16 Holotype of *Amblycephalus nuchalis* Boulenger, 1900 in preservative (NHMUK 1901.5.14.2, adult male).** (A) Dorsolateral view of body; (B) ventrolateral view of body; (C–F) head in lateral right, lateral left, dorsal, and ventral aspects, respectively. Photos by G. Vogel.

**Material examined:** In this study we directly examined nine specimens of *Pareas nuchalis* from Borneo (Malaysia, Indonesia) and Sumatra, including the holotype of *Amblycephalus nuchalis* (see Table S11, Appendix S2).

*Coloration.* Due to preservation in ethanol for more than a century, the coloration and the pattern of the holotype has been changed, as the consequences the type specimen no longer retains the original coloration characteristics (Fig. 16). Dorsal surface of the head uniform dark brown, head with two postorbital stripes, the upper running laterally backwards to the dorsal scales on the neck, bifurcating posterior to the secondary temporal

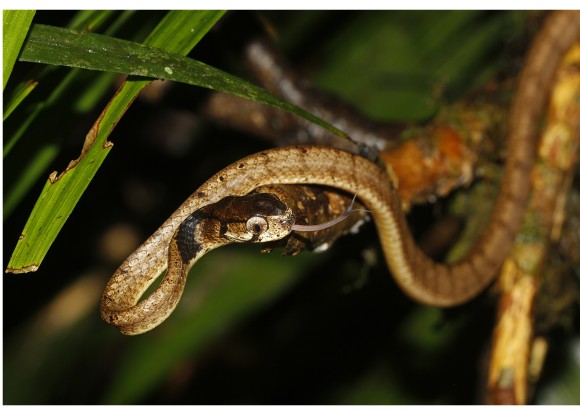

**Figure 17 *Pareas nuchalis* in life, adult male from Kota Kinabalu N.P., Kundasang, Sabah, Borneo, Malaysia.** Photo by L.A. Neimark.               

scales, forming a dark vertical bar reaching the mouth angle, upper postorbital stripes contacting each other on the nuchal area forming a large black ring-shaped blotch (Fig. 16); lower postorbital stripes partially faded, reaching the anterior part of SL6. Dorsal surface of the body light brown with 78 vertical faint dark-brown bands; ventral surface of the body and tail yellowish-cream with sparse brown dusting.

**Variation:** Measurements and scalation features of *Pareas nuchalis* specimens examined ($n = 9$) are presented in Table S11. There is a certain sexual variation observed in the body size, numbers of ventral and subcadal scales: males have slightly lager total length (TL 345.0–678.0 mm, average 529.0 ± 110.2 mm, $n = 6$) than females (TL 352.0–503.0 mm, average 422.3 ± 76.0 mm, $n = 3$); males also have generally slightly higher number of ventral and subcaudal scales than females (VEN 207–220, average 212.2 ± 4.5, $n = 6$; SC 108–120, average 114.2 ± 4.7, $n = 6$ in males *vs.* VEN 201–208, average 205.3 ± 3.8, $n = 3$; SC 102–107, average 105.0 ± 2.7, $n = 3$ in females). Other morphological and coloration features showed no significant variation among the specimens of the examined series.

**Distribution:** Until recently, this species was considered to be endemic to the island of Borneo (*Stuebing, Inger & Lardner, 2014*), and was recorded both from the State of Sarawak and Sabah in Malaysia, Brunei and from the Indonesian part of the island (Kalimantan). In this study we for the first time recorded *P. nuchalis* from central Sumatra, where it was previously confused with *P. carinatus* (see *David & Vogel, 1996*).

**Etymology:** The species name "*nuchalis*" is a Latin adjective in nominative singular meaning "nuchal" and was given in reference to the characteristic black ring-shaped spot in the nuchal area in this species.

**Recommended English name:** Barred slug-eating snake.

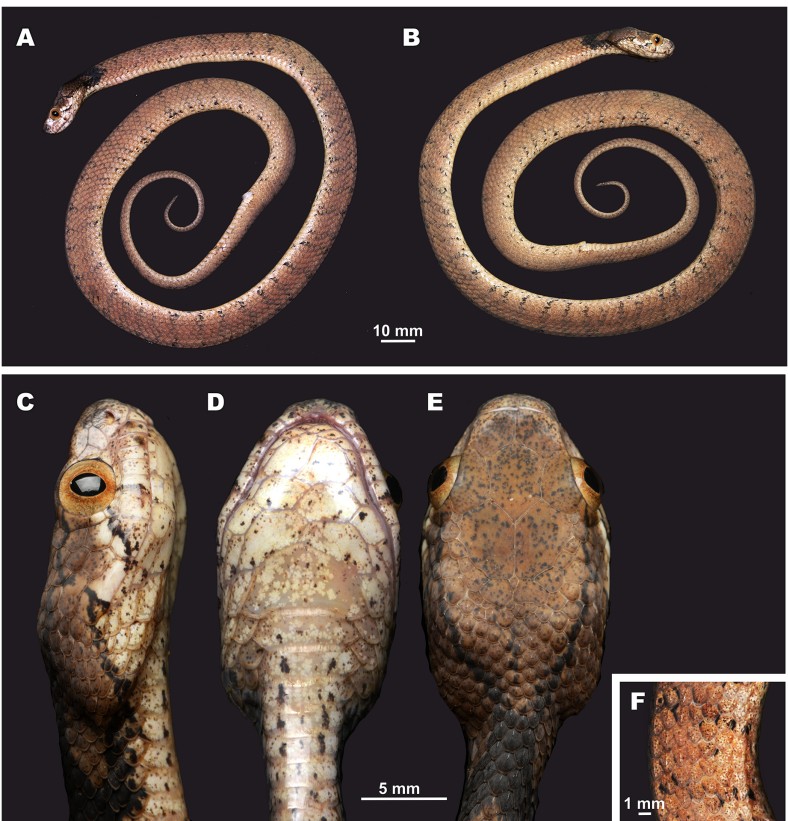

**Figure 18** **Holotype of *Pareas abros* sp. nov. in life (ZMMU R-16393, adult male).** (A) Lateral left view of body; (B) later right view of body; (C–E) head in lateral right, ventral, and dorsal aspects, respectively; (F) close-up of midbody showing keeled dorsal scales (in eleven rows, five to six of them seen on one side of the body). Photos by N. A. Poyarkov.             

### Pareas abros Poyarkov, Nguyen TV, Vogel & Orlov sp. nov.

[urn:lsid:zoobank.org:act:85CA3212-E8D4-48D1-8ED2-DC8CB183E7E9]

Figures 6I, 18–19; Table 2; Tables S11 and S13.

**Holotype:** ZMMU R-16393 (field number NAP-08867), adult male collected by N. A. Poyarkov from the montane evergreen tropical forest near the offsprings of the Paete River, within the Song Thanh N.P., La Dee Commune, Nam Giang District, Quang Nam Province, central Vietnam (N 15.53353, E 107.38434; elevation 1083 m asl.) on May 05, 2019.

**Paratypes:** ZMMU R-16392 (field number NAP-06251), adult male, and ZMMU R-14788 (field number NAP-06252), adult female, both collected by N. A. Poyarkov and N. L. Orlov from the montane evergreen tropical forest within the Sao La Nature Reserve, A Roang area, A Luoi Distict, Thua Thien – Hue Province, central Vietnam (N 16.10334, E 107.444453; elevation 796 m asl.) on September 11–17, 2015.

**Diagnosis:** *Pareas abros* **sp. nov.** differs from all other members of the genus *Pareas* by the combination of the following morphological characters: body size medium (TL 434–565

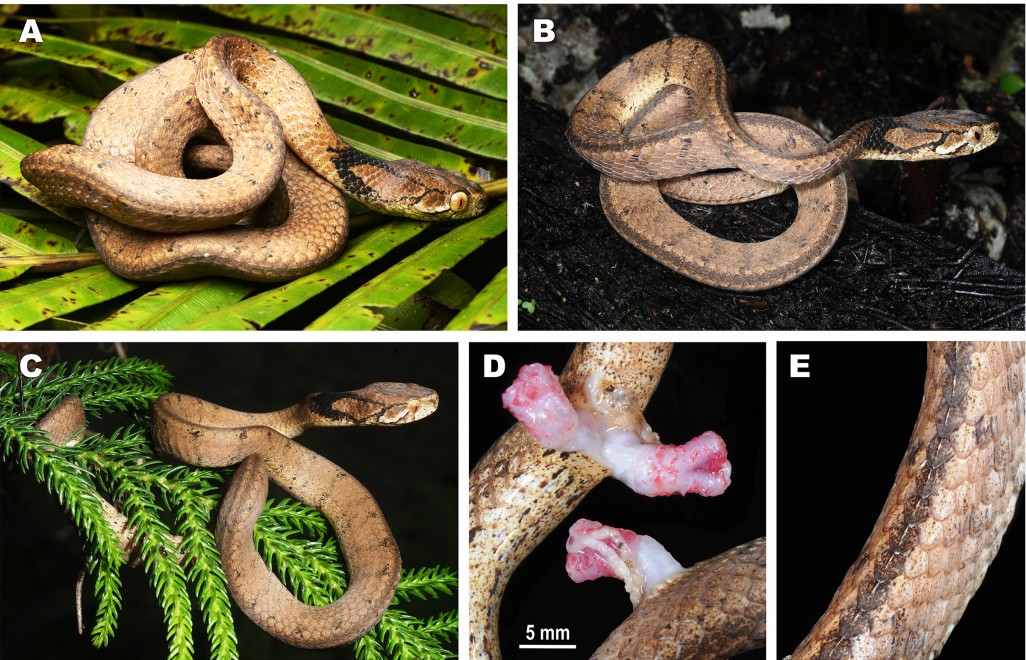

**Figure 19** *Pareas abros* **sp. nov. in life.** (A) Male holotype from Song Thanh N.P., Quang Nam Province, Vietnam (ZMMU R-16393); (B) male paratype from A Roang, Sao La N.R., Thua Thien – Hue Province, Vietnam (ZMMU R-16392); (C) female paratype from A Roang, Sao La N.R., Thua Thien – Hue Province, Vietnam (ZMMU R-14788); (D) partially everted hemipenis of ZMMU R-16392 from asulcal (above) and sulcal (below) aspects; (E) close-up of midbody of ZMMU R-16392 showing enlarged vertebrals and keeled dorsal scales (in eleven rows). Photos by N. A. Poyarkov.

mm); head notably flattened dorsoventrally; anterior pair of chin shields longer than broad; loreal and prefrontal not contacting the eye; three suboculars; two postoculars; temporals 3 + 3; the single median vertebral scale row slightly enlarged; 9 supralabial scales; generally 8 (rarely 9) infralabials scales; 15 dorsal scale rows at midbody, of them 9–11 median scale rows slightly keeled; 180–184 ventrals; 83–95 subcaudals, all divided; dorsum yellowish-brown with distinct dark-brown vertebral line, barely distinct blackish vertebral spots and 44–56 faint interrupted transverse dark-brown bands (Figs. 19A–19C); upper postorbital stripes thick, slate-black, bifurcating posterior to the secondary temporals, forming a thick black line, continuing to the 7th SL and further on the neck; upper postorbital stripes contacting each other on the nuchal area forming a large ring-shaped blotch (Figs. 19A–19C); two thick, black lower postorbital stripes reaching the 6th and 8th SL, and continuing to the lower jaw; belly beige with dense brownish-gray dusting and dark brown elongated spots forming two longitudinal lines on the lateral sides of ventrals (Fig. 18D); iris in life beige with ochraceous to orange speckles and veins getting denser around the pupil (Fig. 19).

**Description of the holotype** (ZMMU R-16393): Adult male, specimen in a good state of preservation (Fig. 18); body slender and notably flattened laterally; head very large, notably flattened dorsoventrally, clearly distinct from thin neck (head more than two times

wider than neck near the head basis), snout obtusely rounded in profile and in dorsal view; eye very large, pupil vertical and elliptical. Hemipenis not everted.

*Body size.* SVL 314 mm; TaL 120 mm; TL 434 mm; TaL/TL: 0.276.

*Body scalation.* Dorsal scales in 15–15–15 rows, slightly keeled on 11 upper scale rows at midbody (Fig. 18F), all lacking apical pits; the single median vertebral scale row slightly enlarged; the outermost dorsal scale row not enlarged; ventrals 184 (+ 1 preventral), lacking lateral keels; subcaudals 92; cloacal plate single.

*Head scalation.* Rostral not visible from above; single nasal; two internasals, much wider than long, narrowing and slightly curving back laterally (in dorsal view), anteriorly in contact with rostral, laterally in contact with nasal and loreal, posteriorly in contact with prefrontal, not contacting preocular; two large irregularly pentagonal prefrontals, much larger than internasals and with a straight suture between them, not in contact with eye; the single frontal scale hexagonal with the lateral sides slightly concave, parallel to the body axis, longer than wide, smaller than parietals; single preocular; two postoculars, semicrescentic in shape, not fused with subocular; single presubocular; three suboculars; two loreals, upper larger then lower, irregularly pentagonal, in contact with presubocular, prefrontal, internasal and nasal, not touching the eye; 9/9 supralabials, 4th to 6th SL touching the subocular, none of them reaching the eye, 9th by far the largest, elongated; 1/1 supraoculars; 3/3 anterior temporals and 3/3 posterior temporals; 8/8 infralabials, the anteriormost in contact with the opposite along midline forming a diagonal suture between them, bordering mental, the anterior five pairs of infralabials bordering the anterior chin shields; 3 pairs of chin shields interlaced, no mental groove under chin and throat; the anterior chin shields relatively large, much longer than broad, followed by two pairs of chin shields that are much broader than long.

*Coloration.* In life, dorsal and ventral surfaces of the head brownish with dense dark-brown mottling. Head with three lateral postorbital stripes: the upper postorbital stripes thick, slate-black lines running from postocular backwards towards the dorsal scales of neck, bifurcating posterior to the secondary temporals, forming a ventral branch – a thick black line, continuing to the 9th SL and further to the posterior corner of the jaw and on the neck; the dorsal branch extending to the top of the nape contacting each other on the nuchal area; both of the branches of the upper postorbital stripe join at the front of the neck to form a large black ring-shaped blotch that covers the entire nape area (Fig. 18C). Two lower postorbital stripes: the posterior one is a thick, black line starting from the lower portion of postorbital, running ventrally and posteriorly towards lower temporals to 8th and 9th supralabials, and further and continuing to the lower jaw; the anterior lower postorbital stripe is a short thick black vertical stripe starting from the middle subocular, reaching the 6th SL and continuing further to the 4th IL as a line of black spots (Fig. 18C). Upper labials beige with dense brown dusting. Dorsal surfaces of body yellowish-brown with distinct dark-brown line running along the vertebral dorsal scale row, barely distinct blackish vertebral spots and ca. 56 faint
interrupted transverse dark-brown bands (Figs. 18A and 18B). Ventral surfaces of the head, body and tail cream-beige with dense brownish-gray dusting and dark brown elongated spots forming two longitudinal lines on the lateral sides of ventrals (Fig. 18D). Iris in life beige with orange speckles and veins getting denser around the pupil; pupil black (Fig. 18C). **In preservative**: After 2 years of storage in ethanol the general coloration pattern did not change; the tan tint of the dorsal coloration, and orange tints on the head and eye have faded becoming grayish-brown; other features of coloration remain unchanged.

**Variation:** Measurements and scalation features of the type series ($n = 3$) is presented in Table S11. The holotype has two loreals while there is only one in the two paratypes. There is a certain variation observed in the number of ventral and dorsal scales: males have higher number of subcaudals (92–95, $n = 2$) than the single female (83, $n = 1$); dorsal scales are keeled in 11 scale rows at midbody in males *vs.* in 9 scale rows are keeled in the single female. Coloration features among the members of the type series were very similar.

*Hemipenial morphology.* The hemipenis is partially everted in the adult male paratype ZMMU R-16392 (Fig. 19D). The partially everted organ symmetrical, bilobed, forked, the surface from base to crotch smooth, with numerous (7–11) shallow folds and on the sulcal surface and fewer 3–6 larger dermal ridges on the asulcal surface. Sulcus spermaticus deep, with fleshy swollen edges, bifurcating into two separate canals towards the apical lobes. Apical lobes with well-developed ornamentation, covered with large fleshy irregularly curved folds in 4–5 rows and fleshy protuberances, separated with deep slits, forming a complex pattern resembling brain cortex.

**Comparisons:** *Pareas abros* **sp. nov.** differs from *P. berdmorei* by the anterior pair of chin shields longer than broad (*vs.* broader than long), by slightly longer tail (TaL/TL 0.26–0.29 [average 0.28 ± 0.01] *vs.* 0.17–0.27 [average 0.21 ± 0.02]), by slight higher number of ventrals (83–95 [average 90.00 ± 6.24] *vs.* 57–89 [average 71.13 ± 7.25]), by the presence of a large ring-shaped black blotch in the nuchal area (*vs.* absent); and by the presence of the dark vertebral line (*vs.* absent). The new species differs from *P. carinatus* by longer tail (TaL/TL 0.26–0.29 [average 0.28 ± 0.01] *vs.* 0.18–0.25 [average 0.22 ± 0.02]), by a slightly higher number of subcaudals (83–95 [average 90.0 ± 6.2] *vs.* 54–96 [average 69.2 ± 9.0]), by the presence of a large ring-shaped black blotch in the nuchal area (*vs.* absent); by the presence of the dark vertebral line (*vs.* absent); and by weakly-discernable faint transverse dark bands on body (*vs.* well-discernable dark bands). *Pareas abros* **sp. nov.** differs from *P. nuchalis* by prefrontal not in contact with the eye (*vs.* in contact); by 9–11 dorsal scale rows keeled (*vs.* all dorsal scales totally smooth); and by the black nuchal blotch forming a complete ring (*vs.* incomplete ring-shaped blotch). The new species differs from *P. kuznetsovorum* **sp. nov.** (described above) by a higher number of ventrals (180–184 [average 182.7 ± 2.3] *vs.* 167); by keeled 9–11 dorsal scale rows (*vs.* all dorsal scales totally smooth); by smaller body size (TL 434–565 mm [average 506.7 ± 66.7 mm] *vs.* 639 mm). *Pareas abros* **sp. nov.** differs from Pareas temporalis by a slightly lower number of ventrals (180–184 *vs.* 185–198), by weak keels present only on 9–11 dorsal scale rows

(*vs.* all dorsal scale rows strongly keeled), by a lower number of enlarged vertebral scales rows (1 *vs.* 3), and by having 44–56 faint dark transverse bands (*vs.* absence of dark cross-bands on the body). Morphological comparisons between all species of the subgenus *Pareas* are detailed in Table 2 and Table S13.

**Distribution:** The new species is to date known only from two localities in Quang Nam (locality 42, Fig. 1) and Thua Thien – Hue (locality 43, Fig. 1) provinces of central Vietnam, both of them are located within the Kon Tum – Gia Lai Plateau, the northern portion of the Central Highlands (Tay Nguyen) Region of Vietnam. The Kon Tum – Gia Lai Plateau is isolated from the adjacent mountain massifs by lowland areas and is characterized by a high level of herpetofaunal endemism (*Bain & Hurley, 2011*; *Poyarkov et al., 2021*); the new species is also likely an endemic of this mountain region.
The holotype of the new species was collected in just 2 km from the national border of Vietnam and Lao PDR (locality 42, Fig. 1), hence the occurrence of *Pareas abros* **sp. nov.** in Laos is highly anticipated.

**Etymology:** The new species name "*abros*" is a Latinized adjective in nominative singular derived from the classical Greek word "*abros*" (αβρός), meaning "cute", "handsome", and "delicate". The name is given in reference to the appealing and cute appearance of the new species, as well as other members of the genus *Pareas*.

**Recommended English name:** Cute slug-eating snake.

**Ecology notes:** Pareas abros **sp. nov.** inhabits montane evergreen tropical forests of Kon Tum – Gia Lai Plateau and was recorded on the elevations from 796 to 1,083 m asl.
In both localities, the new species was recorded in fragments of primary polydominant forest along the banks of montane streams. The new species was active at 21:00–00:00 h, the specimens were usually located while crawling on the branches of bushes and trees ca. 1–1.5 m above the ground; the holotype was spotted while crossing a small forest trail. Other sympatric members of Pareidae in both localities include *Pareas formosensis*.
The diet of *Pareas abros* **sp. nov.** is unknown; it likely consists of terrestrial mollusks as in other congeners.

### *Pareas temporalis* Le, Tran, Hoang & Stuart, 2021
Figures 6J, 20–21; Table 2; Tables S11 and S13.

**Holotype:** UNS 09992 (field number LD25711), adult female, Doan Ket Commune, Da Huoai District, Lam Dong Province, Vietnam (11.340370°N, 107.620561°E, elevation of 496 m a.s.l.), coll. 25 July 2020 by Duong T.T. Le and Thinh G. Tran.

**Reffered specimens** (*n* = 6): ZMMU R-13656 (field number NAP-01610), adult male collected by N. A. Poyarkov from the low-elevation disturbed bamboo forest within the valley of the Sui Lan River in the environs of Ben Cau and Phuok Son ranger stations, Cat Loc sector of the Cat Tien National Park, Lam Dong Province, southern Vietnam (N 11.69444, E 107.30639; elevation 135 m asl.) on June 20, 2011; DTU 471, adult female, collected by L. H. Nguyen and H. M. Pham from the valley of Suoi Lanh Stream, Rung Ge

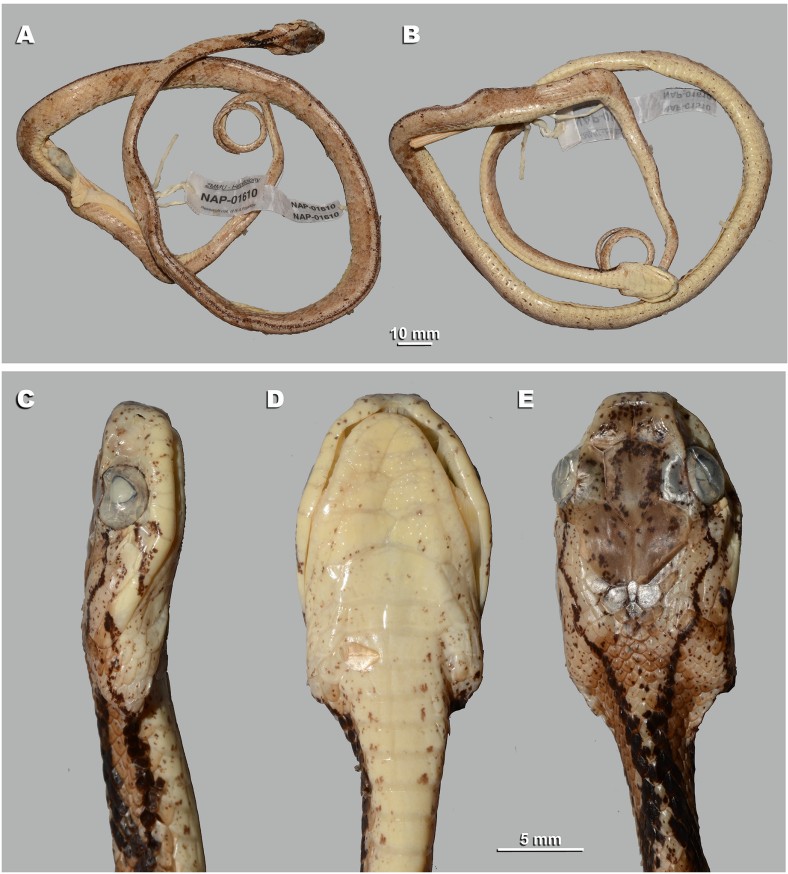

**Figure 20 Specimen of *Pareas temporalis* *Le et al., 2021* in preservative (ZMMU R-13656, adult male).** (A) Dorsal view of body; (B) ventral view of body; (C–E) head in lateral right, ventral, and dorsal aspects, respectively. Photos by G. Vogel.

Commune, Di Linh District, Lam Dong Province, southern Vietnam (N 11.46725, E 108.06915; elevation of ca. 1,320 m asl.) on March 1, 2019; SIEZC 20214, adult female, collected by L. H. Nguyen in Gia Bac District, Lam Dong Province, southern Vietnam (N 14.220392, E 108.317133; elevation of ca. 1,050 m asl.) on August 10, 2018; and SIEZC 20215, adult female, collected by V. B. Tran from Biduop – Nui Ba N.P., Lam Dong Province, southern Vietnam (N 12.23383, E 108.44866; elevation of ca. 790 m asl.) on May 30, 2017; DTU 486–487 (two adult females) collected by T. A. Pham and T. V. Nguyen in Rung Ge Commune, Di Linh District, Lam Dong Province, southern Vietnam (N 11.46725, E 108.06915; elevation of ca. 1,320 m asl.), on May 10, 2020.

**Remark:** Until our study this recently discovered species was known only from a single female specimen (*Le et al., 2021*). In the present work we provide morphological and genetic data for six additional specimens of this species of the both sexes from three previously unkown localities from southern Vietnam (see Fig. 1).

**Updated diagnosis:** *Pareas temporalis* differs from other members of the genus *Pareas* by the following combination of morphological characters: body size large (TL 555–665 mm);

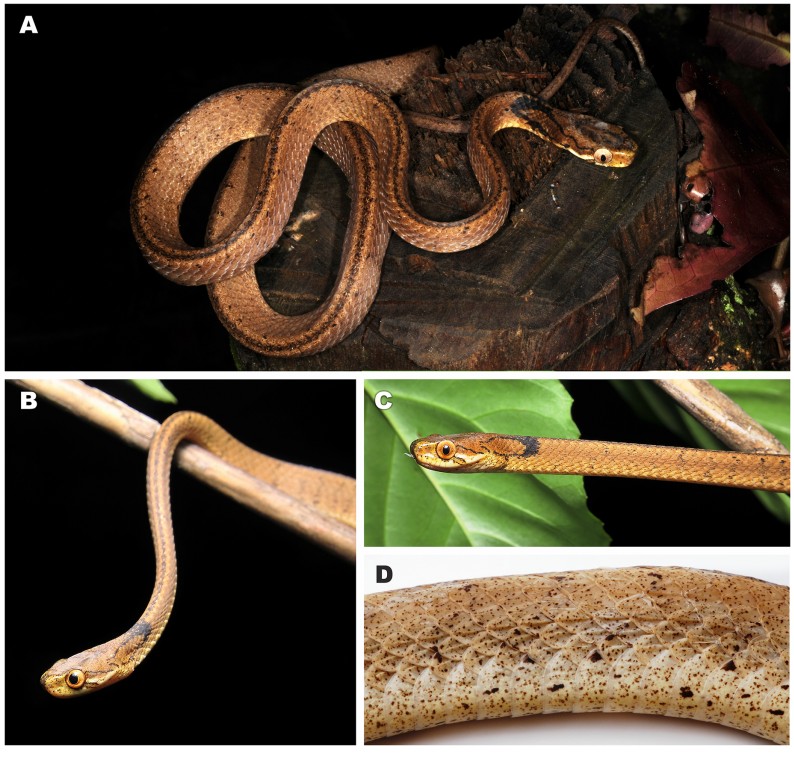

**Figure 21** *Pareas temporalis* **in life.** (A) Male specimen from Cat Loc, Lam Dong Province, Vietnam (ZMMU R-13656); (B and C) female specimen from Di Linh, Lam Dong Province, Vietnam (SIEZC 20214); (D) close-up of midbody of SIEZC 20214 showing all dorsal scales keeled (in 15 rows). Photos by N. A. Poyarkov (A), and L.H. Nguyen (B–D).  

head distinctly flattened dorsoventrally; the anterior pair of chin shields notably longer than broad; loreal and prefrontal not contacting the eye; two suboculars; generally two (rarely one or absent) postoculars; temporals generally 3 + 3 (rarely 3 + 4); three median vertebral scale rows slightly enlarged; generally 8 (rarely 7) supralabial scales; generally 8 (rarely 7) infralabials scales; 15 dorsal scale rows, all of them notably keeled; 187–198 ventrals; 86–92 subcaudals, all divided; dorsum bright yellowish-brown to light-orange with distinct blackish vertebral line edged with two light yellowish paravertebral lines; vertebral spots and transverse dark bands absent (Fig. 21); dorsal scales with few scattered small black spots; two very clear thin black postorbital stripes beginning from the lower and upper edges of each postorbital scale; the lower postorbital stripe as two thin parallel black lines reaching the anterior part of 8th SL, not continuing to the lower jaw and chin; the left and right upper postorbital stripes contacting each other at the nuchal area forming a black ring-shaped blotch (Figs. 21A–21C); belly yellowish-cream with sparse brownish dusting and irregular small spots; iris in life amber-colored to bright-orange (Figs. 21A–21C).

**Description of a male specimen** (ZMMU R-13656): Adult male, specimen in a good state of preservation, body dissected longitudinally along the ventral scales (Fig. 20); body slender and notably flattened laterally; head very large, distinctly flattened dorsoventrally,

clearly distinct from the thin neck (head more than three times wider than neck width near the head basis), snout blunt in dorsal and lateral views; eye very large, pupil vertical and elliptical. Hemipenis not everted.

*Body size.* SVL 413 mm; TaL 142 mm; TL 555 mm; TaL/TL: 0.256.

*Body scalation.* Dorsal scales in 15–15–15 rows, all scales notably keeled and lack apical pits; vertebral scale rows and the two adjacent rows of scales (3 medial dorsal scale rows) slightly enlarged; the outermost dorsal scale row not enlarged; ventrals 198 (+ 1 preventral), lacking lateral keels; subcaudals 98, all paired; cloacal plate single.

*Head scalation.* Rostral not visible from above; single nasal; two internasals, much wider than long, narrowing and slightly curving back laterally (in dorsal view), anteriorly in contact with rostral, laterally in contact with nasal and loreal, posteriorly in contact with prefrontal, not contacting preocular; two large irregular pentagonal prefrontals, much larger than internasals and with a slightly diagonal suture between them, not in contact with the eye; one subhexagonal frontal, longer than wide, smaller than parietals, with the lateral sides almost parallel to each other, slightly converging posteriorly; single preocular; single postocular present on the left side, absent on the right side of the head, semicrescent in shape, not fused with subocular; presubocular absent; two suboculars; single loreal, in contact with presubocular, prefrontal, internasal and nasal, not touching the eye; 9/9 supralabials, 3rd to 5th or 3rd to 7th SL touching the subocular, none of them reaching the eye, 9th by far the largest, elongated; 1/1 supraocular; 3/3 anterior temporals and 3/4 posterior temporals; 7/8 infralabials, the anterior most in contact with the opposite along the midline forming a diagonal suture between them, bordering mental, the anterior five pairs of infralabials bordering the anterior chin shields; three pairs of chin shields interlaced, no mental groove under chin and throat; the anterior chin shields relatively large, much longer than broad, followed by two pairs of chin shields that are much broader than long.

*Coloration.* In life, dorsal surface of the head brownish with some blackish mottling and larger spots, getting denser on frontal and prefrontals. Head with two clear thin black lateral postorbital stripes: the upper one is a well-developed slate-black line starting from postocular and running backwards to dorsal scales of the neck, the left and right upper postorbital stripes contacting each other at the nuchal area forming a black X-shaped pattern in dorsal view (Fig. 20E), and a large ring-shaped blotch in lateral view (Fig. 20C); the lower postorbital stripe is an interrupted black line starting from the lower portion of postorbital, and running ventrally and posteriorly towards lower temporals and to 8th and 7th supralabials and not continuing to the lower jaw and chin. Other markings on the lateral surfaces of the head include a large dark, elongated spot on the 6th supralabial, and a thick slate-black line running from the posterior edge of the 9th supralabial backwards to the and further on the lateral surfaces of the neck, where it joins the ring-shaped nuchal blotch ventrally (Fig. 20C; Fig. 21A). Supralabials yellowish-white with rare tiny brown spots. Dorsal surfaces of the body bright yellowish-brown to light-orange

with distinct blackish vertebral line edged with two light yellowish paravertebral lines; vertebral spots and transverse dark bands absent (Fig. 21); dorsal scales on the sides of the body with few irregularly scattered small dark spots; the ventral surfaces of the head, body and tail is yellowish-cream with sparse brownish dusting and irregular small spots; iris in life amber-colored to bright-orange; pupil black (Figs. 21A–21C). **In preservative**: After ten years of storage in ethanol, the general coloration pattern has not changed; light brownish and yellowish tints in the coloration of dorsum, head and eyes faded becoming grayish-brown; other features of the coloration remain unchanged (Fig. 20).

**Variation:** Measurements and scalation features of additional specimens of *P. temporalis* (*n* = 7) is presented in Table S11. All coloration features of the additional specimens are very similar to those described for the holotype (*Le et al., 2021*). The holotype UNS 09992 generally agrees with the series of specimens examined by us, it has slightly higher number of supralabials (9/8 *vs.* 7–8) and infralabials (8/9 *vs.* 7–8), higher number of anterior temporals (4/5 *vs.* 3/3), and generally slightly lower number of posterior temporals (3/3 *vs.* 3–4), it also has 2/3 postoculars, while in the specimens we examined generally had 2/2 postoculars and 1/0 postoculars in the single male specimen ZMMU R-13656. We observed a certain sexual dimorphism in *P. temporalis*: male ZMMU R-13656 has a higher number of ventrals (198 in a single male *vs.* 185–188 in five females), and subcaudals (92 in male *vs.* 86–89 in five females). Other morphological and chromatic features showed no significant variation among the examined specimens.

**Updated comparisons:** *Pareas temporalis* differs from its sister species *Pareas abros* **sp. nov.** by a higher number of ventrals (185–198 *vs.* 180–184), by all dorsal scale rows strongly keeled (*vs.* weak keels present only on 9–11 dorsal scale rows), by a higher number of enlarged vertebral scales rows (3 *vs.* 1), and by the absence of dark cross-bands on the body (*vs.* 44–56 faint dark transverse bands present). *Pareas temporalis* differs from *P. berdmorei* by a higher number of ventrals (185–198 *vs.* 164–186), by a higher number of subcaudals (86–92 *vs.* 63–78), by the presence of a black ring-shaped blotch on the collar (*vs.* absent), and by having all dorsal scale rows strongly keeled (*vs.* weak keels present only on 3–13 dorsal scale rows). *Pareas temporalis* differs from *P. carinatus* by having two postoculars (*vs.* single or absent); by generally slightly higher number of subcaudals (86–92 *vs.* 54–96), by having all dorsal scale rows strongly keeled (*vs.* weak keels present on 3–11 dorsal scale rows), and by the presence of a black ring-shaped blotch on the collar (*vs.* absent). *Pareas temporalis* differs from *P. nuchalis* by prefrontal not contacting the eye (*vs.* in contact), by a slightly lower number of ventrals (187–198 *vs.* 201–220), by a lower number of subcaudals (86–92 *vs.* 102–120), and by having all dorsal scale rows strongly keeled (*vs.* all dorsal scales smooth). Finally, *P. temporalis* differs from *Pareas kuznetsovorum* **sp. nov.** in having all dorsal scale rows strongly keeled (*vs.* all dorsal scales smooth), by a higher number of enlarged vertebral scales rows (3 *vs.* 1), and by a higher number of vetrals (185–198 *vs.* 167). Morphological comparisons between all species of the subgenus *Pareas* are detailed in Table S13.

**Updated distribution:** In addition to the type locality of this species, *P. temporalis* is also known from four localities, all in the Lam Dong Province of southern Vietnam (localities 37–40, Fig. 1). All these localities belong to the Langbian (Da Lat) Plateau – the southernmost part of the Annamite Range, well-known by its high level of endemism in herpetofauna (*Bain & Hurley, 2011*; *Poyarkov et al., 2021*). We assume that *Pareas temporalis* is endemic to the Langbian Plateau; it is expected to occur on middle elevations in the adjacent provinces of southern Vietnam: Binh Phuoc, Dak Nong, Dak Lak, Ninh Thuan and Binh Thuan, and also likely might inhabit the southeasternmost part of the Mondulkiri Province of Cambodia.

**Etymology:** The species name "*temporalis*" is a Latin adjective in nominative singular, meaning "temporal" and was given in reference to the high number of temporal scales in this species (*Le et al., 2021*).

**Recommended English name:** Di Linh slug-eating snake.

**Ecology notes:** Pareas temporalis is a nocturnal, elusive forest-dwelling snake inhabiting mid-elevation montane evergreen tropical forests of the Langbian Plateau and its foothills; it was recorded from elevations from 135 to 1,320 m asl. All specimens were spotted after rain at night between 21:00 and 01:00 h while crawling or perching on branches of bushes, bamboo and Calamus sp. palm leafs. The holotype was found at 21:00 h on a tree branch 1.5 m above the ground in disturbed mixed broadleaf and bamboo forest, where it occurred in sympatry with *P. margaritophorus* (*Le et al., 2021*). Diet of *P. temporalis* is unknown, but as in other congeners, it likely consists of terrestrial mollusks. It is sympatric with a number of other *Pareas* species across its range, and is commonly recorded in the same habitats with *P. b. unicolor* **comb. nov.** In Di Linh District of Lam Dong Province, *P. temporalis* was recorded in sympatry with four species of the genus *Pareas*, including *P. b. unicolor* **comb. nov.**, *P. margaritophorus*, *P. macularius* and *P. formosensis*. With five species of *Pareas* co-occurring in the same habitat, the area of Di Linh represents the center of the genus diversity in Vietnam.

### Subgenus *Eberhardtia Angel, 1920* stat. nov.

*Eberhardtia Angel, 1920*: 291, by monotypy.

**Synonymy:**

*Northpareas Wang et al., 2020*: Appendix S3 (*nomen nudum*).

**Type species:** *Eberhardtia tonkinensis Angel, 1920*; this taxon is currently considered a junior synonym of *Pareas formosensis* (Van Denburgh, 1909); see *Ding et al. (2020)* for discussion.

**Phylogenetic definition:** *Eberhardtia* is a maximum crown-clade name referring to the clade originating with the most recent common ancestor of *Pareas formosensis* and *Pareas monticola*, and includes all extant species that share a more recent common ancestor with these taxa than with *Pareas carinatus*.

**Diagnosis:** The members of the subgenus *Eberhardtia* differ from the members of the subgenus *Pareas* by the following combination of morphological characters: frontal subhexagonal to diamond-shaped with its lateral sides converging posteriorly (Fig. 5A); anterior pair of chin shields longer than broad (Figs. 5I and 5J); a single thin elongated subocular; and the ultrastructure of dorsal scales not ravine-like, having pore and arc structures, with arcs connecting to each other forming characteristic lines (*Wagler, 1830*; *Smith, 1943*; *Taylor, 1965*; *Vogel et al., 2020*; *He, 2009*; *Guo, Wang & Rao, 2020*; our data; see Table S14 for details).

**Etymology:** *Angel (1920)* dedicated his new genus to the collector of the single specimen of its type species, the French botanist Philippe Albert Eberhardt (1874–1942).

**Distribution**: Distributed in the north-eastern part of the Oriental zoogeographic region from the Eastern Himalaya to central and eastern China, islands of Hainan, Taiwan and the southern Ryukyus, southwards throughout the Indochina to the Peninsular Malaysia and Sumatra.

**Content:** 20 species, including *P. andersonii* Boulenger; *P. atayal* You, Poyarkov & Lin; *P. boulengeri* (Angel); *P. chinensis* (Barbour); *P. formosensis* (van Denburgh); *P. geminatus* Ding, Chen, Suwannapoom, Nguyen, Poyarkov & Vogel; *P. hamptoni* (Boulenger); *P. iwasakii* (Maki); *P. kaduri* Bhosale, Phansalkar, Sawant, Gowande, Patel & Mirza; *P. komaii* (Maki); *P. macularius* Theobald; *P. margaritophorus* (Jan); *P. modestus* Theobald; *P. monticola* (Cantor); *P. niger* (Pope); *P. nigriceps* Guo & Deng; *P. stanleyi* (Boulenger); *P. victorianus* Vogel, Nguyen & Poyarkov; *P. vindumi* Vogel; and *P. xuelinensis* Liu & Rao.

**Recommended English name:** Northern slug-eating snakes.

## DISCUSSION

### Phylogeny and classification of Pareinae

In this study, we present an updated multilocus phylogeny for the ancient Asian subfamily of slug-eating snakes, the Pareinae. We estimate the basal divergence within the Pareinae as the late Eocene (ca. 39.3 mya) making this group one of the oldest radiations of Colubroidea snakes (*Zaher et al., 2019*; *Li et al., 2020*). Our study includes representatives of all currently recognized taxa within of the subfamily and is, to the best of our knowledge, the most comprehensive among the published works both in terms of taxon and gene sampling. Our integrative analysis of the molecular and morphological data resolves several long-standing systematic controversies regarding the subfamily Pareinae.
In particular, we confidently resolve the phylogenetic relationships among the three genera of the Pareinae: *Aplopeltura* is strongly suggested as the sister genus of *Pareas sensu lato*, while the genus *Asthenodipsas* is reconstructed as the most basal taxon within the subfamily with sister relationships to the clade *Aplopeltura* + *Pareas*. This topology contradicts several earlier studies on the taxonomy of the group (*e.g.*, *Guo et al., 2011*; *Wang et al., 2020*), but generally agrees with the recent multilocus phylogenetic study by

*Deepak, Ruane & Gower (2019)*, though in our phylogeny we got higher values of node support.

We also provide strong evidence for the monophyly of all Pareinae genera. While monophyly of the presently monotypic genus *Aplopeltura* was never questioned, a number of studies suggested that the genera *Pareas* and *Asthenodipsas* might represent paraphyletic taxa (*Guo et al., 2011*; *Pyron et al., 2011*; *Wang et al., 2020*), or were recovered as monophyletic groups but without a significant support (*e.g.*, *Deepak, Ruane & Gower, 2019*). At the same time, we demonstrate the deep differentiation within both *Pareas* and *Asthenodipsas*, each of these genera comprises two reciprocally monophyletic groups, which diverged almost simultaneously during the early to middle Oligocene (ca. 30–31.3 mya). We also show that, though the monophyly of *Pareas* and *Asthenodipsas* is not questioned anymore, the major clades within these genera demonstrate significant differences among each other in external morphology, scale microornamentation, and biogeographic affinities; similar results were also obtained by a number of earlier studies (*Grossmann & Tillack, 2003*; *He, 2009*; *Guo et al., 2011*; *Guo, Wang & Rao, 2020*; *Wang et al., 2020*). We argued that the groups within *Pareas* and *Asthenodipsas* should be taxonomically recognized, what would enhance the diagnosability and clade stability of these taxa, make them more comparable units to other snake genera, restrain taxonomic vandalism, and eventually fully stabilize the taxonomy of Pareinae. Therefore, we recognize two subgenera within the genus *Pareas sensu lato* (*Pareas sensu stricto* and *Eberhardtia* **stat. nov.**), and two subgenera within the genus *Asthenodipsas sensu lato* (*Asthenodipsas sensu stricto* and *Spondylodipsas* **subgen. nov.**). Earlier studies which addressed the genus-level taxonomy of *Pareas* have either wrongly identified the type species of the genus (*Guo et al., 2011*), or have overlooked the existence of an available genus-level name for one of the clades (*Eberhardtia* *Angel, 1920*), what resulted in the creation of a nomen nudum ('*Northpareas*', see *Wang et al., 2020*). We would like to emphasize the importance of a thorough analysis of the available literature and possible synonyms prior to making a taxonomic decision in order to prevent publication of unavailable names or junior synonyms. Within the genus *Pareas*, our phylogenetic results support the recognition of two species groups within the subgenus *Pareas sensu stricto* (*P. carinatus* and *P. nuchalis* groups), and four species groups within the subgenus *Eberhardtia* **stat. nov.** (*P. hamptoni*, *P. chinensis*, *P. margaritophorus*, and *P. monticola* groups) (see Fig. 3); this taxonomy is largely concordant with the results of the previous studies (*Guo et al., 2011*; *You, Poyarkov & Lin, 2015*; *Bhosale et al., 2020*; *Vogel et al., 2020*, *2021*; *Wang et al., 2020*; *Ding et al., 2020*).

Although our understanding of the phylogenetic relationships within the Pareinae is now improved, it is still far from complete. For example, our phylogeny included only five of nine currently recognized species of *Asthenodipsas*; five species of the genus were described within the last decade, of which four were described based solely on morphological evidence (*Quah et al., 2019*, *2020*; *Quah, Lim & Grismer, 2021*). We would like to further stress herein that in this age of molecular genetics and biodiversity crises, the application of molecular methods became crucial for taxonomic practice in studies of herpetofaunal diversity in Southeast Asia (*Smith, Poyarkov & Hebert, 2008*; *Murphy et al.,*

*2013*). Not only the phylogenetic hypothesis is crucial for any comparative or biogeographic analyses, it now also became a keystone of biodiversity conservation (*Shaffer et al., 2015*; *Chomdej et al., 2020*). Phylogenetic studies on the remaining species of *Asthenodipsas* are required to fully resolve the taxonomy of the genus; furthermore, additional taxon and gene sampling will likely enhance the phylogenetic resolution on the level of the subfamily Pareinae and might lead to discovery of additional new lineages and species.

## Underestimated species diversity of *Pareas* in Indochina

Though not being the most species-rich group of Asian snakes, the slug-eating snakes are widely distributed across the Southeast Asia and have a number of specialized morphological and ecological characteristics that are hypothesized to facilitate speciation. Being dietary specialists on terrestrial slugs and snails, the Pareinae occupy ecological niches inaccessible to other groups of Asian snakes, at the same time several species of the slug-eating snakes can successfully coexist with their congeners, likely due to niche partitioning and further specialization in preferred prey, dentition asymmetry, and feeding behavior (*Chang et al., 2021*). For example, up to three closely-related species of *Pareas* share same habitats in the areas of sympatry in Taiwan (*You, Poyarkov & Lin, 2015*; *Chang et al., 2021*); we report for the first time the sympatric co-occurrence of six *Pareas* species in the montane forests of Lam Dong Province of southern Vietnam (*P. temporalis*, *P. kuznetsovorum* **sp. nov.**, *P. b. unicolor*, *P. macularius*, *P. margaritophorus*, and *P. formosensis*), of which five species were recorded sharing the same habitat in Di Linh District. On the other hand, the sympatric co-occurrence of several often closely related species of *Pareas*, often makes correct species identification difficult, and may also lead to some cryptic species being overlooked.

Several recent taxonomic studies on *Pareas* have demonstrated that this genus has a high level of hidden and yet undescribed diversity (*e.g. You, Poyarkov & Lin, 2015*; *Bhosale et al., 2020*; *Vogel et al., 2020*, *2021*; *Ding et al., 2020*; *Liu & Rao, 2021*; *Yang et al., 2021*). Out of 26 currently recognized species of *Pareas* (including those described in this work), twelve species were discovered within the last twelve years (*Uetz, Freed & Hošek, 2021*), of which eleven species were described based on an integrative evidence from morphological and molecular data. At the same time, hasty taxonomic revisions may often lead to creation of unnecessary synonyms and taxonomic inflation (*Isaac, Mallet & Mace, 2004*). For example, recently *Wang et al. (2020)* revised the taxonomy of the genus *Pareas* and described two new species from China: *P. mengziensis* (member of the *P. hamptoni* group) and *P. menglaensis* (member of the *P. carinatus* group). Subsequent work by *Liu & Rao (2021)* noted that *Wang et al. (2020)* described their species without first resolving the historical taxonomic confusions of *P. yunnanensis* (Vogt) and *P. niger* (Pope), at that time considered junior synonyms of *P. chinensis* (*Wallach, Williams & Boundy, 2014*). *Liu & Rao (2021)* further showed that *P. mengziensis* represents a subjective junior synonym of *P. niger*, and clarified distribution and the phylogenetic placement of this species, which was also confirmed by our analyses. Furthermore, as demonstrated in our study, in their description of *P. menglaensis*, *Wang et al. (2020)* did

not provide any comments on the distribution and existing junior synonyms of *P. carinatus*, including *P. berdmorei*, originally described from Myanmar. Herein we also analyze the distribution of phylogenetic relationships within the *P. carinatus* species group and further demonstrate that *P. menglaensis* actually represents a subjective junior synonym of *P. berdmorei* (see Results). Therefore we would like to further emphasize herein the importance of careful treatment of the available synonyms and especially of the examination of the respective type specimens in taxonomic practice. It is thus recommended that scientists, before describing a new taxon would thoroughly evaluate the available old names, the existing type specimens and/or new materials from the respective type localities. This would prevent the taxonomy from becoming confusing and the available taxa from being overlooked. We would also like to further stress herein the importance of international collaboration in resolving taxonomically confusing species complexes distributed across the international borders.

The combination of molecular and morphological data allowed this study to assess the diversity, clarify the actual geographical distribution as well as to evaluate the validity of the taxa included in the *P. carinatus-nuchalis* complex. As a result, our study revealed an unprecedented diversity within this complex, with six major lineages representing distinct species, each with significant genetic and morphological differences from the others (see Results). In our study, we consider *P. carinatus sensu stricto* distributed from the Tenasserim Range in the Peninsular Thailand and Myanmar southwards to Malayan Peninsula, Sumatra, Java, and Borneo Islands. We also revise the populations from the mainland Indochina and southern China previously referred to as *P. carinatus* or *P. menglanensis* (*Wang et al., 2020*), and revalidate *P. berdmorei* as a distinct species; this taxon is widely distributed across Indochina and the adjacent parts of Yunnan and eastern Myanmar, while *P. menglaensis* is considered a subjective junior synonyms of this species. We also describe two new previously completely unknown species of *Pareas* from Vietnam, namely: *Pareas kuznetsovorum* **sp. nov.** (it belongs to *P. carinatus* species group and represents a sister species of *P. berdmorei*) and *Pareas abros* **sp. nov.**, respectively. We provide additional information on morphological variation and distribution of the recently described *P. temporalis*. The recent discovery of the latter two species is quite unexpected since they are morphologically profoundly different from all other mainland members of the subgenus *Pareas* and according to our phylogeny and morphological similarities belong to *P. nuchalis* species group, what is also indicated by an earlier study of *Le et al. (2021)*. It also should be noted, that our study represents the first record of *P. nuchalis* on Sumatra Island; this species has been previously considered to be restricted to Borneo. Overall, the revalidation of *P. berdmorei* along with description of *Pareas abros* **sp. nov. and** *Pareas kuznetsovorum* **sp. nov.** brings the total number of species in *Pareas* to 26 and the number of Pareinae species to 36.

In our study we also analyze geographic variation of morphological, chromatic and molecular characters within the wide-ranged species of the subgenus *Pareas*, namely *P. carinatus* and *P. berdmorei*. We report on a significant diversity within these species with two divergent, allopatric (to the best of our knowledge), and morphologically diagnosable groups revealed within *P. carinatus*, and three such groups within

*P. berdmorei*. Should they be taxonomically recognized? The phylogenetic species concept (PSC, see *Cracraft, 1983*; reviewed by *De Queiroz, 2007*) suggests that the minimal monophyletic group on a tree should be considered a species. However, the recent progress in evolutionary phylogenomics allows revealing population genetic structure and estimate the geneflow among populations and even species in unparalleled detail (*e.g.*, *Benestan et al., 2015*). This, however, often makes an accurate characterization of species boundaries within an evolutionary framework quite challenging: distinguishing between population-level genetic structure and species divergence is often problematic (*Chan et al., 2020*). A number of recent phylogenomic studies have demonstrated that a number of what was considered complexes of cryptic species actually represent highly admixed and structured metapopulation lineages, rather than true cryptic species (*e.g.*, *Chan et al., 2020*, *2021*). One of the adverse consequences of ignoring gene flow in species delimitation is the overestimation of species numbers by interpreting population structure as species divergence, thus enhancing taxonomic inflation (*Chan et al., 2020*). Therefore, in the present paper, in order to assess the revealed diversity within *P. carinatus* and *P. berdmorei* we apply the subspecies concept *sensu Hillis (2020)* and *Marshall et al. (2021)*, where subspecies are defined as geographically circumscribed lineages that may have been temporarily isolated in the past, but which may have since merged over broad zones of intergradation that not necessarily show the evidence of reproductive isolation between them. We recognize two subspecies within *P. carinatus*: *P. c. carinatus* (Sundaland and Malayan Peninsula south of Kra) and *P. c. tenasserimicus* **ssp. nov.** (Tenasserim Range north of Kra), and three subspecies within *P. berdmorei*: *P. b. berdmorei* (from eastern Mayanmar across Thailand to northern Laos, northern Vietnam and southern China), *P. b. unicolor* (southern Vietnam and Cambodia), and *P. b. truongsonicus* **ssp. nov.** (Northern Annamites in central Vietnam and Laos). Though with the data in hand we do not have any evidence of genetic admixture between these groups, it cannot be excluded that the future studies with a finer sampling might reveal a certain degree of geneflow among them. We herein prefer recognizing them as subspecies due to the overall morphological similarity of these lineages, which are mostly distinguished by coloration rather than scalation features, their presumably allopatric distribution pattern, their comparatively young evolutionary age (5.9–4.0 mya), and the historical precedent of use of the subspecies for describing diversity within these snakes (*P. b. unicolor* was originally described as a subspecies of *P. carinatus*) (see Results).

Despite the recent significant progress (*Ding et al., 2020*; *Vogel et al., 2020*, *2021*; *Le et al., 2021*), our understanding of *Pareas* diversity is still incomplete. Our study revealed a high morphological variation among the examined specimens of *P. berdmorei* and *P. carinatus*; however our genetic sampling is not fully comparable to the morphological sampling. For example, our phylogenetic analysis lacked specimens of *P. carinatus* from the Greater Sunda Islands and the adjacent smaller offshore islands; many areas in the central Indochina also remained unassessed. Further field survey and taxonomic efforts both in Indochina and Sundaland will likely reveal additional lineages within the widely-distributed and insufficiently sampled species of *Pareas*.

## Historical biogeography of Pareinae

The results of our biogeographic reconstruction suggest that the common ancestor of the Pareinae likely inhabited Sundaland (Fig. 2), while its sister group the Xylophiinae is restricted to peninsular India (*Deepak, Ruane & Gower, 2019*). The split between Pareinae and Xylophiinae is dated to happen during the middle Eocene (ca. 42.2 mya in our analysis, estimated as ca. 44.9 mya in *Deepak, Ruane & Gower, 2019*), and likely reflects the ancient faunal exchange between the Indian Subcontinent and the Sundaland *via* a land bridge which existed during the early and middle Eocene (*Ali & Aitchison, 2008*; *Morley, 2018*). Similar patterns were reported, for example, in Draconinae agamid lizards (*Grismer et al., 2016a*), and Microhylinae narrow-mouth frogs (*Garg & Biju, 2019*; *Gorin et al., 2020*). In particular, the assumptive vicariance between Pareinae and Xylophiinae and the distribution patterns of the two subfamilies remarkably resembles the divergence pattern between microhylid genera *Micryletta* (widely distributed across the Southeast Asia) and *Mysticellus* (restricted to southern peninsular India), which was dated as 39.7–40.6 mya (*Garg & Biju, 2019*). Our study thus provides further evidence for faunal exchange between the Indian Subcontinent and Sundaland during the middle Eocene.

Our results accord with the "upstream" colonization hypothesis in Pareinae. The general pattern of colonization in Pareinae is from Sundaland to the mainland Asia, this dispersal and subsequent vicariance likely happened during the early Oligocene (ca. 33.6 mya; Fig. 2). It should be noted, that though now the Sundaland is mostly represented by a number of archipelagos, in Oligocene it was connected to the mainland Southeast Asia *via* the Sunda shelf, which remained subaerial during the most part of Cenozoic (*Cao et al., 2017*; *Morley, 2018*). Therefore, the general direction of diversification in Pareinae was likely from the tropical continental margins of Sundaland to a nontropical Asian landmass. Starting with at least middle Eocene, Sundaland was covered with perhumid rainforests and became a major source of mainland Asian lineages for a vast number of taxa of plants and animals (see *De Bruyn et al., 2014*; *Grismer et al., 2016a*; *Morley, 2018*, and references therein). Examples include the stream toad genus *Ansonia* (*Grismer et al., 2016b*), the litter toads *Leptobrachella* (*Chen et al., 2018*), and the breadfruit genus *Artocarpus* (*Williams et al., 2017*). Our study is probably the first example of "upstream" colonization in Asian snakes. Further studies on the diversification patterns of other endemic Asian genera on a broad geographic scale might yield key insights into the drivers of speciation in Asia and result in a comprehensive picture of the regional source-sink dynamics between islands and continents.

Our analyses suggest that the common ancestor of the genus *Pareas* likely inhabited parts of the Indochinese Peninsula, Indo-Burma, and the areas which now became the Himalayas (Fig. 2). The basal split within *Pareas* is dated as early Oligocene (ca. 31.3 mya; Fig. 2), and temporally coincides with climatic shifts during this time. The period of late Eocene-early Oligocene transition was characterized by dramatically cool and dry climate in Southeast Asia (*Zachos et al., 2001*); during this time the perhumid forests contracted and fragmented (*Milne & Abbott, 2002*; *Bain & Hurley, 2011*; *Buerki et al., 2013*; *Morley, 2018*). These processes could potentially drive the initial diversification of

*Pareas* through vicariance (Fig. 2). Moreover, the two major clades of *Pareas* (Fig. 2) are estimated to have begun diversification during the Miocene along with the significant growth of average temperature and humidity (ca. 23–12 mya; see *Bain & Hurley, 2011*). These climatic changes lead to expansion of perhumid tropical forests and likely promoted the expansion and further diversification of *Pareas*. A similar pattern was recently reported by *Chen et al. (2018)* for *Leptobrachella* toads.

The large region of the Himalayas and Indo-Burma is suggested as the possible ancestral area for the subgenus *Eberhardtia* **stat. nov.**, the most species-rich group of *Pareas*. The further diversification of this clade into species groups took place during the early to middle Miocene and likely took place in Himalaya and the adjacent parts of western and southern China, with subsequent dispersals to Indochina and the East Asian islands (Fig. 2). The Himalaya are now recognized as an area of exceptional diversity and endemism largely due to the uplift-driven speciation, suggesting that orogeny created conditions favoring rapid *in situ* diversification of resident lineages, which accelerated during the Miocene (reviewed in *Xu et al., 2020*). Our data suggest that geomorphological factors are also likely responsible for shaping the diversification within the subgenus *Eberhardtia* **stat. nov.** The origin and diversification of the four species groups within *Eberhardtia* **stat. nov.** temporally coincide with the rapid increase of uplifting of the Qinghai–Tibet Plateau during the Miocene (15–7 mya, *An et al., 2001*; *Che et al., 2010*), which finally gave rise to the intensification of the modern South Asian monsoon climate. This process is concidered to have accelerated the diversification of numerous Asian animal groups that share similar distributions with *Eberhardtia* **stat. nov.** (*e.g.*, *Che et al., 2010*; *Blair et al., 2013*; *Gao et al., 2013*; *Chen et al., 2017, 2018*; *Xu et al., 2020*). It is noteworthy that following the dispersal of *Eberhardtia* **stat. nov.** across the mainland East Asia, its members at least twice have independently colonized the East Asian islands of Taiwan and the southern Ryukyus during the late Miocene to early Pliocene, giving rise to an *in situ* diversification of a number of endemic island species (*You, Poyarkov & Lin, 2015*; *Chang et al., 2021*). Similar patterns were reported in other groups of Asian herpetofauna (*e.g.*, *Yuan et al., 2016*; *Nguyen et al., 2020a*; *Yang & Poyarkov, 2021*; *Gorin et al., 2020*).

The origin of the subgenus *Pareas* likely took place in what is now Western Indochina, from where, during the early to middle Miocene, it colonized Eastern Indochina leading to formation of several endemic lineages and species (Fig. 2). The Annamite or Truong Son Range, including the mountains areas of the Kon Tum–Gia Lai and Langbian plateaus, are reknown as the center of floral and faunal endemism (*e.g.* *Averyanov et al., 2003*; *Bain & Hurley, 2011*; *Monastyrskii & Holloway, 2013*; *Poyarkov et al., 2014, 2021*). According to our analyses, Eastern Indochina appears as an evolutionary hotspot for *Pareas*, having the high species diversity and degree of endemism, with up to six species of the genus sympatrically distributed in the montane forests of the Langbian Plateau. Among the surprising results of our study is the independent "downstream" colonization of Sundaland from Indochina during the late Miocene, which happened twice by the members of the *P. carinatus* and *P. nuchalis* species groups (Fig. 2). The recent discovery of two new species of the *P. nuchalis* group in mountain areas of Eastern Indochina (*Pareas*

*abros* **sp. nov.**, and *P. temporalis* by *Le et al., 2021*) is quite unexpected, and provides further evidence for faunal interchange between Eastern Indochina and Borneo (*e.g.*, see *Teynié et al., 2004*). Throughout the late Cenozoic these territories were directly connected by a landbridge along the eastern edge of Sunda Shelf formed by the ancient delta joining the modern river systems of Mekong and Chao Phraya, and covered by lowland evergreen rain-forests (*De Bruyn et al., 2014*). However, similar biogeographic patterns are rarely reported for the herpetofauna (but see *Wood et al., 2012*; *Geissler et al., 2015*; *Chen et al., 2017*; *Suwannapoom et al., 2018*; *Poyarkov et al., 2018a*; *Gorin et al., 2020*; Grismer et al., in press), therefore additional sampling from other regions, including the different parts of the Sundaland, is needed to test this hypothesis. Overall, our study reinforces the idea that Indochina represents an indispensable hotspot for the evolution and maintenance of Southeast Asian biodiversity (*De Bruyn et al., 2014*).

## CONCLUSIONS

In this work, we provide an updated phylogenetic hypothesis for the slug-eating snakes of the subfamily Pareinae based on mtDNA and nuDNA markers for 29 of 33 currently recognized Pareinae species (88%). We also included data for six lineages that have not been examined phylogenetically before our work, including the previously unknown two new species and two new subspecies of *Pareas*. Therefore, our study provides the most comprehensive taxon sampling for Pareinae published to date. This, along with morphological examination of 269 preserved specimens of Pareinae, including the available type specimens for the genus *Pareas*, allowed us to revise the phylogenetic relationships and taxonomy of the subfamily. Our work further highlights the importance of broad phylogenetic sampling, ground-level field surveys, and careful examination of type materials to achieve an accurate picture of phylogenetic relationships, global biodiversity, and evolutionary patterns in cryptic groups such as the Pareinae slug-eating snakes.

The present work clearly indicates a vast underestimation of diversity in the subgenus *Pareas*, and that the present taxonomy of the group is incomplete. Further integrative studies combining morphological and genetic analyses are essential for a better understanding of evolutionary relationships within this cryptic and taxonomically challenging radiation of Asian snakes. Overall, our study further highlights the importance of comprehensive and accurate taxonomic revisions not only for the better understanding of biodiversity and its evolution, but also for the elaboration of adequate conservation actions.

While *P. c. carinatus* and *P. b. berdmorei* are quite widely distributed taxa and their conservation status is of the least concern, the distribution of *P. c. tenasserimicus* **ssp. nov.**, *P. b. unicolor*, and *P. b. truongsonicus* **ssp. nov.** is most likely restricted to comparatively narrow areas within the Indochina. At the same time, among the two newly described species *Pareas kuznetsovorum* **sp. nov.** is to date known only from a single specimen, while the ranges of *Pareas abros* **sp. nov.** and *P. temporalis* are restricted to isolated montane areas of Kon Tum – Gia Lai and Lam Dong plateaus, respectively. The estimated ranges of the two new *Pareas* species are likely relatively small, however the actual extent of

their distribution and population trends remain unknown; urgent actions are needed for careful assessment of their conservation status. We herein tentatively suggest that at present *Pareas kuznetsovorum* **sp. nov.**, *Pareas abros* **sp. nov.**, and *P. temporalis* should be categorized as Data Deficient (DD) according to the *IUCN Standards and Petitions Committee (2019)*. Further research is required to clarify the extent of their distribution population trends, and natural history, thereby facilitating elaboration of adequate conservation actions.

Our work further highlights the importance of the Indochinese region, including the territories of Vietnam, Laos, Cambodia, and Thailand, as one of the key biodiversity hotspots with high levels of herpetofaunal diversity and endemism (*Bain & Hurley, 2011*; *Geissler et al., 2015*; *Duong et al., 2018*; *Nguyen et al., 2018*, *2019*, *2020b*, *2020c*; *Poyarkov et al., 2018b*, *2019*, *2021*; *Grismer et al., 2019*, *2021a*, *2021b*; *Chomdej et al., 2021*; *Uetz, Freed & Hošek, 2021*). This area is facing many pressures with major habitat loss by deforestation due to logging, the growing human population density and infrastructure development, agricultural extension, forest fires, and tourism development (*Lang, 2001*; *Meyfroidt & Lambin, 2009*). Therefore, further studies are urgently needed to assess and manage the biodiversity and elaborate the adequate conservation efforts before more undescribed species are lost.

Overall, our study reinforces the idea of the global importance of Indochina as the principal evolutionary hotspot for the autochthonous herpetofaunal diversity, as well as a key area facilitating dispersals between East Asia, Indo-Burma and Sundaland. Further studies on phylogeny and the diversification patterns of different animal groups endemic to Asia on a broad geographic scale might provide key insights into the role of complex paleogeography and paleoclimate history as the drivers of speciation forming the extant Asian biodiversity.

## ACKNOWLEDGEMENTS

The authors are grateful to Andrey N. Kuznetsov (JRVTTC) and Thai Van Nguyen (SVW) for supporting our study. We thank Vladislav A. Gorin, Evgeniya N. Solovyeva, Roman A. Nazarov, Sabira S. Idiatullina, The Anh Nguyen, Hieu Minh Pham, Huy Xuan Ngoc Nguyen, Bang Van Tran, Eduard A. Galoyan, and Anna B. Vassilieva for their support during the fieldwork and assistance in the lab or with data analysis. We are also grateful to Liudmila B. Salamakha for preparation of drawings for this paper. We are deeply bgrateful to Leonid A. Neimark, Indraneil Das, Guek Hock Ping aka Kurt Orion, Huy X. N. Nguyen, and Mali Naiduangchan for providing photos used in the present paper. We are grateful to Tasos Limnios for his advices on Ancient and Modern Greek grammar and lexicon. Furthermore, we thank the following collection curators, who gave access to specimens under their care and helped us while we visited their respective institutions: Jens Vindum and Alan Leviton (CAS); Yuezhao Wang, Xiaomao Zeng, Jiatang Li and Ermi Zhao (CIB); Ding Li (DL); Alan Resetar (FMNH); Nicolas Vidal and Annemarie Ohler (MNHN); Giuliano Doria (MNSG); Hmar Tlawmte Lalremsanga (MNZU); Colin McCarthy and Patrick Campbell (NHMUK); Silke Schweiger and Georg Gassner (NMW); Esther Dondorp, Pim Arntzen and Ronald de Ruiter (RMNH); Gunther Köhler and

Linda Acker (SMF); George Zug, Kenneth Tighe and Ronald Heyer (USNM); Dennis Rödder and Wolfgang Böhme (ZFMK); Oliver Rödel and Frank Tillack (ZMB); Jakob Hallermann (ZMH); Frank Glaw and Michael Franzen (ZSM); and Valentina F. Orlova and Roman A. Nazarov (ZMMU). Alan Leviton (CAS) is thanked for inviting GV to CAS. Ding Liee, Jiatang Li and Zening Chen are thanked for their support of the work of GV in China. We want to thank Anastasio Zographos (Montmorency, France) for his expertise in classical Greek grammar. The authors thank the academic editor of this paper, Patrick David (MNHN) and an anonymous reviewer for their useful comments which improved the earlier draft of the manuscript.

### Funding

This work was supported by the Russian Science Foundation to Nikolay A. Poyarkov (RSF grant No. 19-14-00050; specimen collection, molecular, phylogenetic and morphological analyses, data analysis); by Thailand Science Research and Innovation Fund and the University of Phayao (FF65-UoE003; specimen collection) to Chatmongkon Suwannapoom; and by a grant of the Ministry of Science and Higher Education of the Russian Federation (No. 075-15-2021-1069) and state theme of Zoological Institute RAS 1021051302397-6 (to Nikolai Orlov). The funders had no role in study design, data collection and analysis, decision to publish, or preparation of the manuscript.

### Grant Disclosures

The following grant information was disclosed by the authors:
Russian Science Foundation to Nikolay A. Poyarkov (RSF): 19-14-00050.
Thailand Science Research and Innovation Fund and the University of Phayao: FF65-UoE003.
Ministry of Science and Higher Education of the Russian Federation: 075-15-2021-1069.
Zoological Institute RAS: 1021051302397-6.

### Competing Interests

Nikolay A. Poyarkov serves as an Academic Editor for PeerJ. The other authors declare that they have no competing interests.

### Author Contributions

- Nikolay A Poyarkov conceived and designed the experiments, performed the experiments, analyzed the data, prepared figures and/or tables, authored or reviewed drafts of the paper, and approved the final draft.
- Tan Van Nguyen conceived and designed the experiments, performed the experiments, analyzed the data, prepared figures and/or tables, authored or reviewed drafts of the paper, and approved the final draft.
- Parinya Pawangkhanant performed the experiments, analyzed the data, prepared figures and/or tables, authored or reviewed drafts of the paper, and approved the final draft.

- Platon V Yushchenko performed the experiments, analyzed the data, prepared figures and/or tables, authored or reviewed drafts of the paper, and approved the final draft.
- Peter Brakels performed the experiments, analyzed the data, prepared figures and/or tables, authored or reviewed drafts of the paper, and approved the final draft.
- Linh Hoang Nguyen performed the experiments, analyzed the data, prepared figures and/or tables, authored or reviewed drafts of the paper, and approved the final draft.
- Hung Ngoc Nguyen performed the experiments, analyzed the data, prepared figures and/or tables, authored or reviewed drafts of the paper, and approved the final draft.
- Chatmongkon Suwannapoom conceived and designed the experiments, performed the experiments, analyzed the data, authored or reviewed drafts of the paper, and approved the final draft.
- Nikolai Orlov conceived and designed the experiments, performed the experiments, authored or reviewed drafts of the paper, and approved the final draft.
- Gernot Vogel conceived and designed the experiments, performed the experiments, analyzed the data, prepared figures and/or tables, authored or reviewed drafts of the paper, and approved the final draft.

## Animal Ethics

The following information was supplied relating to ethical approvals (*i.e.*, approving body and any reference numbers):

The Institutional Ethical Committee of Animal Experimentation of the University of Phayao

## Field Study Permissions

The following information was supplied relating to field study approvals (*i.e.*, approving body and any reference numbers):

No field studies were carried out specifically for this work; tissue samples and specimens stored in museum collections were used in this study. However, some specimens stored in the mentioned collections were collected by the coauthors of this manuscript or with their participation during numerous field trips in a time frame over 15 years.

Institutions and authorities from Thailand, Laos and Vietnam who granted the fieldwork permissions are as follows:

- The Institute of Animals for Scientific Purpose Development (IAD), Bangkok, Thailand;

- The Biotechnology and Ecology Institute Ministry of Science and Technology, Lao PDR;

- The Department of Forestry, Ministry of Agriculture and Rural Development of Vietnam;

- Forest Protection Departments of the Peoples' Committee of Gia Lai Province, Vietnam.

## DNA Deposition

The following information was supplied regarding the deposition of DNA sequences:

The sequences obtained in this study are available at GenBank: MZ712215–MZ712343.

## Data Availability

The raw morphological data are summarized in Supplemental Files. The raw data has been supplied as voucher specimens and tissue samples stored in the following herpetological collections:

1) AUP: School of Agriculture and Natural Resources, University of Phayao, Phayao, Thailand;

2) BNHS: Bombay Natural History Society, Mumbai, India;

3) CAS: California Academy of Sciences Museum, California, USA;

4) CHS: Song Huang's private collection, College of Life Sciences, Anhui Normal University, Wuhu, Anhui, China;

5) CIB: Chengdu Institute of Biology, Chengdu, China;

6) DL: Ding Lee's private collection, Chengdu, China;

7) DTU: Duy Tan University, Da Nang, Vietnam;

8) FMNH: Field Museum of Natural History, Chicago, USA;

9) GP: Guo Peng's private collection, College of Life Science and Food Engineering, Yibin University, Yibin, China;

10) KIZ: Kunming Institute of Zoology, Chinese Academy of Sciences, Kunming, Yunnan, China;

11) MNHN: Muséum National d'Histoire Naturelle, Paris, France;

12) LSUHC: La Sierra University Herpetological Collection, Riverside, California, USA;

13) MSNG: Museo Civico di Storia Naturale "Giacomo Doria," Genova, Liguria, Italy;

14) MZB: Museum Zoologicum Bogoriense, Juanda 3, Kebun Raya, Bogor, Java, Indonesia;

15) MZMU: Departmental Museum of Zoology, Mizoram University, Mizoram, India;

16) NHMUK (formerly BMNH): The Natural History Museum, London, UK;

17) NMNS: National Museum of Natural Science, Taichung, Taiwan;

18) NMW: Naturhistorisches Museum Wien, Vienna, Austria;

19) QSMI: Queen Saovabha Memorial Institute, Thai Red Cross Society, Bangkok, Thailand;

20) RMNH: Naturalis-Nationaal Natuurhistorisch Museum [formerly Rijksmuseum van Natuurlijke Historie], Leiden, the Netherlands (includes MHNPB & ZMA);

21) SIEZC: Southern Institute of Ecology, Ho Chi Minh City, Vietnam;

22) SMF: Naturmuseum Senckenberg, Frankfurt am Main, Germany;

23) UNS; University of Science, Ho Chi Minh City, Vietnam;

24) USNM: National Museum of Natural History [formerly United States National Museum], Smithsonian Institution, Washington, District of Columbia, USA;

25) YPX: Field number for tissue samples stored in KIZ;

26) ZFMK: Zoologisches Forschungsmuseum Alexander Koenig, Bonn, Germany;

27) ZMB: Zoologisches Museum für Naturkunde der Humboldt-Universität zu Berlin, Berlin, Germany;

28) ZMH: Zoologisches Institut und Museum, Universität Hamburg, Hamburg, Germany;

29) ZMMU: Zoological Museum of Moscow University, Moscow, Russia;

30) ZSM: Zoologische Staatssammlung, München, Germany.

## New Species Registration

The following information was supplied regarding the registration of a newly described species:

Publication LSID:

urn:lsid:zoobank.org:pub:192CDD83-E08C-40B1-92EB-3DB2C3E63CFA

Subgenus name: *Spondylodipsas*

urn:lsid:zoobank.org:act:3FE7563C-2BFE-4BA4-A084-1A66E3D9B706

Subspecies name: *Pareas carinatus tenasserimicus*

urn:lsid:zoobank.org:act:11F7F6BA-4733-41FB-8E2D-405DCA5743E5

Subspecies name: *Pareas berdmorei truongsonicus*

urn:lsid:zoobank.org:act:3E45EE5B-8DD5-4FB1-814A-76DC2B821E29

Species name: *Pareas kuznetsovorum*

urn:lsid:zoobank.org:act:1CD26CB3-F3E9-4370-B501-6F678851C9FB

Species name: *Pareas abros*

urn:lsid:zoobank.org:act:85CA3212-E8D4-48D1-8ED2-DC8CB183E7E9

## Supplemental Information

Supplemental information for this article can be found online at http://dx.doi.org/10.7717/peerj.12713#supplemental-information.

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
