# Peer review of "An integrative taxonomic revision of slug-eating snakes (Squamata: Pareidae: Pareineae) reveals unprecedented diversity in Indochina"

_PeerJ, doi:10.7717/peerj.12713_

## Round 0.1 · original submission · Minor Revisions

I received two reviewer reports for your paper, both of which agree to a great extent that the study is thorough and is a significant contribution for the understanding of this poorly studied group of snakes. Please attend to all of their concerns and provide a point by point rebuttal.

Though I am not recommending specific taxonomic action to be taken, please take into careful consideration the perspectives of the reviewers when considering the taxonomy. I agree with one of the reviewers that the Results and Discussion are not done as well as the Introduction and Materials and Methods sections. So please improve R & D sections to improve the overall quality of your work.

I am awaiting the third report, which is late. If a receive it in a timely manner, I will pass it on. Irrespective of the recommendation by the third reviewer, my overall decision will remain the same but you can attend to any constructive criticism suggested by that reviewer also, at your discretion.

Once the necessary revisionary work is done, I anticipate a great looking paper.

·

Basic reporting

1. – This paper is a major contribution to the difficult systematics of the snake family Pareidae.
It is generally clearly written in professional English (American) language.
2. – Structure of the paper:
- The Introduction is very good and informative.
- The core of the paper is usually very good. However, the problem of the syntypes of P. berdmorei should be addressed.
- The Discussion is very weak. Most parts repeat what has been written in the introduction to the results. The last part, on Conservation, is excellent and would make a better conclusion than the current one.
- The conclusion is really the weakest point of the paper.
3. – The Discussion and Conclusions should be entirely rewritten for 1) reducing the “summary” of your results, 2) avoiding repetitions, 3) providing a true discussion of your results, and 4) present a true conclusion.
These two chapters are really the weak points of your paper, event if some parts are excellent.
4. – The Literature section is very complete and with relevant references
There are mistyping that will be easily corrected.
5. – The structure conforms to PeerJ standards but some parts have to be revised.
6. – All figures are relevant, of high quality, including the phylogenetic trees. Their caption is excellent.
7. - Raw data are supplied and may be accessed online.

Experimental design

1. – Original primary research within Scope of the journal: yes, without doubt.
2. – Research question well defined, relevant & meaningful: yes, all criteria.
This paper addresses with success (but see below) an important taxonomic problem on a large group of Asian snakes
3. – Rigorous investigation performed to a high technical & ethical standard: yes, highly professional
The Materials and Methods section is complete, well explained and extensive. It shows that the authors perfectly know their subject and state-of-the-art methods.
4. – Methods described with sufficient detail & information to replicate: perfect.

Validity of the findings

Well, here, results, namely the descriptions of new taxa or the resurrections of synonyms, are generally excellent. I agree with most of the results but those linked with the division of Pareas berdmorei into three subspecies. How is it possible to describe a subspecies based on one or two specimens?
I am fully aware of authors’ explanations about the choices of their taxonomic concepts and I understand their rationale for dividing two species into subspecies.
In my opinion, the taxon Pareas carinatus tenasserimicus is a full species, whatever may be its genetic distance with P. c. carinatus.
In contrast, I am not agreeing with the division of Pareas berdmorei into three subspecies:
One, P. b. unicolor, occupies a very disjoint range.
One, P. b. annamiticus (what an ugly name!), is barely distinguishable from P. b. berdomrei. Furthermore, it is described based on two specimens, of which one is even referred to as P. berdmorei cf. annamiticus.
I urge the authors to consider again their taxonomy of P. berdomrei and to not take such conclusions based on so reduced a material. It would be better to state that the phylogeny shows that three lineages are visible on the phylogenetic tree but, based on too few specimens, no taxonomic decision can be taken now.
Of course, I accept the validity of other species and I fully agree on the division of P. carinatus as presented in this paper.

Additional comments

You produced a major work on the systematics of Pareidae.
All my corrections, comments and suggestions appear directly on the PDF, aligned at right except in one table. As a summary:
1) Correct the mistyping and unclear sentences.
2) Please stop writing lengths of specimens at the nearest 0.01. This is very meaningless, as the error in measuring a snake is, at best, 0.5 mm.
2) Re-write both the discussion and conclusion; both may be shortened and better structured. This is a mandatory reviewer’s request.
3) Consider again of the validity of the two subspecies of P. berdmorei, as indicated in the PDF. However, I will not make a case if you retain your subspecies based on one or two specimens. This is just meaningless at present time.
4) Address the problem of the syntypes of P. berdmorei and made valid your designations of lectotypes… they miss one of the article of the Code; see the PDF.

All other comments are in the PDF.

Reviewer 2 ·

Basic reporting

No comment.

Experimental design

No comment.

Validity of the findings

No comments.

Additional comments

This manuscript is a high-quality research with a comprehensive review on taxonomic status and species count of slug-eating snakes. The authors clarified the definition of all the genera, subgenera and some species with deep genetic divergence, and further published new taxonomic units.

Most of the comments I list below are just editing problems which could be solved easily with only minor revisions. The only major comment I would argue is the choice of split both Asthenodipsas and Pareas into two subgenera? Or to treat all these clades as independent genus? Nevertheless, I also respect the final decision to remain the clades as subgenera, which helps “enhance taxonomic stability”.

Conclusively, this manuscript is undoubtedly a milestone of research on Pareidae snakes and is with a high value to be published.

Minor comments:

Abstract could be shrunk; the current version is too lengthy (more than 430 words). At the end of the abstract, it could be better to provide a summary of current taxa count of this family.

Line 135. Do you mean Das (2012; 2018) made incorrect identifications? Or do you mean that Das proposed these misidentifications cases in these two references? I think it should be declared more clearly in this sentence.

Line 196. The abbreviation of specimens from National Museum of Natural Science should be “NMNS” instead of “NMNH”. These mistakes also occurred several times in the tables and appendixes; but they are current in Line 2705 and Supplementary file 1.

Lines 2681–2721. I suspect that this section is used to declare the position or the link of open data, not the deposition position of specimens. Furthermore, this information repeated Supplementary file 1, the abbreviation of museums.

Table 2. Currently this table represent the raw data of measurements and counts of each specimen. However, this information is too detailed with redundancy. I suggest to move this table to supporting table, and could be replaced by a new table describing the ranges, means, and variation of these characters.

Supporting files require some rearrangements and organization. I would suggest to put all supplementary Tables and Figures into a single PDF file instead of numerous independent files. Supplemental file 1 (the museum abbreviations) and Supplementary file 2 (the specimen list) could also be incorporated. If sequences have been uploaded to GenBank, Supplementary file 3 (the sequences) could be deleted. All the other collection permissions could be combined as another file. Finally you will have only two or three supplementary files, and this could largely reduce the trouble for a reader to search and switch among different files.

Table S1. Just a gentle and very minor remind that Taiwan is not a part of China. We realized that this political issue is still under debate; but currently it does not belong to the latter. Specimen ID of collection from Taiwan should be NMNS instead of NMNH.

Table S3. Tables do not need vertical lines. I think this might be one of the reasons to combine all these supporting tables into a single file; you could try to unify the format of these tables.

Table S5. Please replace NMNH as NMNS.

Table S12. Please check the format; there are a few unexpected lines under the range of TL, Tal/TL, and SC of P. b. annamiticus.

---

## Round 0.2 · Minor Revisions

Please revise the manuscript, addressing the changes suggested by the reviewer.

·

Basic reporting

1. – This paper is a major contribution to the difficult systematics of the snake family Pareidae.
It is clearly written in professional English (American) language.
2. – Structure of the paper:
- The Introduction is very good and informative.
- The core of the paper is very good throughout.
- The Discussion is now very good and suitable.
- The Conclusion, with its emphasis on conservation, is excellent.
3. – The Literature section is very complete and with relevant references
There are mistyping that will be easily corrected.
5. – The structure conforms to PeerJ standards.
6. – All figures are relevant, of high quality, including the phylogenetic trees. Their caption is excellent.
7. - Raw data are supplied and may be accessed online.

Experimental design

1. – Original primary research within Scope of the journal: yes, without doubt.
2. – Research question well defined, relevant & meaningful: yes, all criteria.
This paper addresses with success an important taxonomic problem on a large group of Asian snakes
3. – Rigorous investigation performed to a high technical & ethical standard: yes, highly professional
The Materials and Methods section is complete, well explained and extensive. It shows that the authors perfectly know their subject and state-of-the-art methods.
4. – Methods described with sufficient detail & information to replicate: perfect.

Validity of the findings

I agree with the validity of the resurrected or described taxa, either at species or, after having read the authors’ rebuttal response, at subspecies levels. In fact, the authors recognized subspecies mainly based on molecular results while subspecies are usually based on morphological criteria. Anyway, both methods are valid.
I just requested the authors to add a sentence regarding the wide distance between the ranges of P. berdmorei berdmorei and P. b. unicolor as shown on Fig. 1.

Additional comments

You produced a major work on the systematics of Pareidae.
All my corrections, comments and suggestions appear directly in the Word file and, from p. 110 onwards (tables, captions…) on the PDF, aligned at right.

I have only one requirement: please correct the mistyping.
All other comments are in the Word and PDF files

---

## Round 0.3 · accepted · Accept

The concerns of the reviewers are now addressed to an appreciable degree. The manuscript now meets the editorial conditions for acceptance. I congratulate the authors on this impressive body of work and wish them the best in their future research work.